

**Diversity of intact polar lipids in the oxygen minimum zone of the Eastern Tropical North Pacific:**
**Biogeochemical implications of non-phosphorus lipids**
Florence Schubotz [1*], Sitan Xie [1,¶], Julius S. Lipp [1], Kai-Uwe Hinrichs [1], Stuart G. Wakeham[2]
[1]MARUM and Department of Geosciences, University of Bremen, 28359 Bremen, Germany
[2]Skidaway Institute of Oceanography, Savannah, GA 31411, USA
[¶]Current address: Wai Gao Qiao Free Trade Zone, 200131 Shanghai, China
[*]Corresponding author. MARUM, University of Bremen, Leobener Str. 13, Room 1070, 28359 Bremen,
Germany. Tel: +49-421-218-65724. Fax: +49-421-218-65715. E-mail: schubotz@uni-bremen.de
**Keywords:** intact polar lipids, phospholipids, glycolipids, betaine lipids, ether lipids, oxylipins,
phospholipid substitution, oxygen minimum zone



**Abstract**
Intact polar lipids (IPLs) are the main building blocks of cellular membranes and contain
chemotaxonomic, ecophysiologic and metabolic information, which makes them valuable biomarkers in
microbial ecology and biogeochemistry.    This study investigates the IPL distribution in suspended
particulate matter (SPM) in the water column of the Eastern Tropical North Pacific Ocean (ETNP), an
area characterized by one of the most extensive open ocean oxygen minimum zones (OMZ) in the world
with strong gradients of nutrients, temperature and redox conditions.    A wide structural variety in polar
lipid head group composition and core structures exists along physical and geochemical gradients within
the OMZ.    Our goal is to use this structural diversity in IPLs to evaluate the microbial ecology and
ecophysiological adaptations that affect organisms inhabiting the OMZ in the context of biogeochemical
cycles.    Diacylglycerol phospholipids are present at all depths, but exhibit highest relative abundance
and compositional variety (including mixed acyl/ether core structures) in the upper and core OMZ where
prokaryotic biomass was enriched.    Surface ocean SPM is dominated by diacylglycerol glycolipids that
are typical lipid components of photosynthetic membranes.    These and other glycolipids with varying
core structures composed of ceramides and hydroxylated fatty acids are also detected with varying relative
abundances in the OMZ and deep oxycline, signifying additional non-phototrophic bacterial sources for
these lipids.    Similarly, betaine lipids (with none or multiple hydroxylations in the core structures) that
are typically assigned to microalgae are found throughout the water column down to the deep oxycline
but do not show a depth-related trend in relative abundance.    Archaeal IPLs comprised of glycosidic and
mixed glycosidic-phosphatidic glycerol dibiphytanyl glycerol tetraethers (GDGTs) are most abundant in
the upper OMZ where nitrate maxima point to ammonium oxidation, but increase in relative abundance



in the core OMZ and deep oxycline.    The presence of abundant non-phosphorus lipids within the OMZ
suggests that the indigenous microbes might be phosphorus limited at phosphate concentrations of 1 to
3.5 μM.    It remains unclear if the detected amino and glycolipids indeed function as substitutes for
phospholipid in these oxygen-depleted environments as microbial sources for many of these lipids still
remain unknown.



## 1. Introduction

Oxygen Minimum Zones (OMZ, first coined by Richards, 1965) are permanently oxygen-deficient regions in the ocean defined by $O_2$ concentrations <20 µM. They primarily occur in areas where coastal or open ocean upwelling of cold, nutrient-rich waters drive elevated levels of primary production and subsequent respiration of organic matter exported out of productive surface waters consumes oxygen faster than it is replaced by ventilation or by mid-depth lateral injections of oxygenated water. OMZs are generally considered as "dead zones" in which low oxygen levels cause habitat compression, whereby species intolerant to low levels of oxygen are restricted to oxygenated surface waters (Keeling et al., 2010; Rush et al., 2012). But even these low levels of oxygen permit vertical migration of some zooplankton taxa into hypoxic waters (e.g., Wishner et al., 2013), although metabolic rates are greatly suppressed (e.g., Seibel, 2011). Oxygen depletion also stimulates diverse microbial life capable of utilizing alternative electron acceptors for respiration under microaerobic conditions (e.g., Ulloa et al., 2012; Tiano et al., 2014; Carolan et al., 2015; Kalvelage et al., 2015; Duret et al., 2015). Important prokaryote-mediated remineralization processes within OMZs include denitrification and the anaerobic oxidation of ammonium (anammox), which together may account for 30-50% of the total nitrogen loss from the ocean to the atmosphere (Gruber, 2008; Lam and Kuypers, 2011). Modern day OMZs comprise ~8% of global ocean volume (Karstensen et al., 2008; Paulmier and Ruiz-Pino, 2009; Lam and Kuypers, 2011), but any expansion in the coming decades as a consequence of global warming and increased stratification (Stramma et al., 2008; Keeling et al., 2010) would have profound effects on marine ecology, oceanic productivity, global carbon and nitrogen cycles, the biological pump and sequestration of carbon (Karstensen et al., 2008; Stramma et al., 2010; Wright et al., 2012). A better understanding of the effect





of low-$O_2$ on marine biogeochemistry and microbial ecology is thus warranted.

The Eastern Tropical North Pacific Ocean (ETNP) off the west coast of Mexico and Central America

hosts one of the largest OMZs in the open ocean, extending halfway across the Pacific Ocean and
comprising ~41% of global OMZs (Lavín and Fiedler, 2006; Fiedler and Talley, 2006; Paulmier and Ruiz-
Pino, 2009).    By comparison, the OMZs of the Eastern Tropical South Pacific Ocean (ETSP) off Peru
and Chile and in the Arabian Sea are ~14% and ~8%, respectively, of global OMZs.    In the ETNP, a
sharp permanent pycnocline develops where warm, saline surface waters lie on top of a shallow
thermocline, producing a highly stratified water column.    Moderate primary production, dominated by
picoplankton, depends on oceanic upwelling and wind mixing of coastal waters but is generally limited
by lack of micronutrient dissolved iron (Franck et al., 2005; Pennington et al., 2006).    Remineralization,
~70% of which is microbially mediated (Cavan et al., 2017), of particulate organic carbon exported out of
surface waters consumes oxygen at rates that cannot be balanced by ventilation across the pycnocline and
by sluggish lateral circulation, leading to $O_2$ levels as low as 0.1 µM at depths between ~100 and ~800 m.
Abundances of micro- (Olson and Daly, 2013) and macro-zooplankton (Wishner et al., 2013; Williams et
al., 2014) that are high in surface waters are reduced in the OMZ, and those macrozooplankton that are
diel vertical migrators survive in the OMZ with reduced metabolic rates (Maas et al., 2014; Cass and Daly,
2015).    Microbial abundances and activities for both heterotrophic and chemoautotrophic metabolisms
are high in both surface waters and within the OMZ, but again with reduced metabolic rates in the OMZ
(Podlaska et al., 2012).    A strong nutricline suggests microbial nitrogen cycling involving co-occurring
nitrification, denitrification and anammox (Rush et al., 2012; Podlaska et al., 2012), perhaps contributing
up to 45% of the global pelagic denitrification (Codispoti and Richards, 1976).    Microbial communities



are mainly comprised of proteobacteria, with increasing contributions of crenarchaea in deeper waters.
Yet, on average ca. 50% of the prokaryotic communities within the OMZ of the ETNP remained
uncharacterized (Podlaska et al., 2012).

Intact polar lipids (IPLs) that are the main building blocks of cellular membranes may be used to

characterize abundance and physiology of aquatic microorganisms from all three domains of life
(Schubotz et al., 2009; Van Mooy et al., 2009; Pitcher et al., 2011; Popendorf, et al., 2011a).    IPL
distributions have been documented in surface waters of the Eastern Subtropical South Pacific (Van Mooy
and Fredricks, 2010), the Western North Atlantic Ocean (Van Mooy et al., 2006; Popendorf, Lomas, et al.,
2011a), the Mediterranean Sea (Popendorf, et al., 2011b), North Sea (Brandsma et al., 2012), lakes (Bale
et al., 2016) and throughout the water column of stratified water bodies (Ertefai et al., 2008; Schubotz et
al., 2009; Wakeham et al., 2012).    Surface waters are typically dominated by nine different IPL classes.
Three diacylglycerol glycolipids, monoglycosyl (1G-), diglycosyl (2G-) and sulfoquinovosyl
diacylglycerol (SQ-DAG), are established photosynthetic markers as they are the main IPLs in all
thylakoid membranes, including those of cyanobacteria (Siegenthaler et al., 1998).    Generally abundant
are also three classes of betaine lipids, diacylglyceryl homoserine (DGTS), hydroxymethyl-trimethyl-*ß*-
alanine (DGTA) and carboxy-*N*-hydroxymethyl-choline (DGCC), which are widely distributed in lower
plants and green algae (Dembitsky, 1996) and are thus typically assigned to eukaryotic algae in the ocean
(Popendorf, et al., 2011a).    Three commonly detected phospholipids are diacylglycerol phosphatidyl
choline (PC), phosphatidyl ethanolamine (PE), and phosphatidyl glycerol (PG), all of which have mixed
eukaryotic or bacterial sources in the upper water column (Sohlenkamp et al., 2003; Popendorf, et al.,
2011a).    These microbial source assignments have been broadly confirmed by isotope labeling studies





(Popendorf, et al., 2011a).   Deeper in the water column of stratified water bodies the IPL distribution
becomes more diverse and other phospholipids such as phosphatidyl (*N*)-methylethanolamine (PME),
phosphatidyl (*N,N*)-dimethylethanolamine (PDME) and diphosphatidyl glycerol (DPG) become more
abundant; these IPLs occur in a number of bacteria that may be present at these depths (cf. Schubotz et
al., 2009; Wakeham et al., 2012).   Several lipids with currently unknown bacterial sources have also been
detected, including glycosidic ceramides or dietherglycerol phospholipids (Schubotz et al., 2009;
Wakeham et al., 2012).   Archaeal IPLs commonly found in the oceanic water column are glycosidic
glycerol dialkyl glycerol tetraethers (GDGT) or mixed phospho-glyco head groups (Pitcher et al., 2011;
Zhu et al., 2016).   Abundances of archaeal GDGTs vary considerably with depth, but are typically
elevated in zones of water column oxygen depletion (e.g., Black Sea, Wakeham et al., 2007; Schubotz et
al., 2009; Cariaco Basin, Wakeham et al., 2012; off Cape Blanc, NE Atlantic, Basse et al., 2014; ETNP,
Xie et al., 2014; ETSP, Sollai et al., 2015), especially where ammonium oxidizing thaumarchaea are
abundant (Pitcher et al., 2011) or at greater depths where Marine Group II euryarchaea have been detected
(Lincoln et al., 2014).

In addition to their use to phylogenetically classify major microbial groups, IPLs can be applied as

metabolic and physiologic markers.   Many organisms remodel their IPL composition when faced with
environmental stressors such as changes in pH, salinity, temperature or availability of nutrients (Zhang
and Rock, 2008; Van Mooy et al., 2009; Turich and Freeman, 2011; Meador et al., 2014; Carini et al.,
2015; Elling et al., 2015; Hurley et al., 2016).   Notably here, replacing phospholipids with non-
phosphorus containing substitute lipids is an important mechanism when facing nutrient limitation in
oligotrophic surface waters where phosphate concentrations reach nanomolar levels.   Cyanobacteria



replace PG-DAG with SQ-DAG (Benning et al., 1993; Van Mooy et al., 2006), microalgae and some
bacteria replace PC-DAG with DGTS (Geiger et al., 1999; Van Mooy et al., 2009; Popendorf, et al., 2011b)
due to their similar ionic charge at physiological pH.    Recently, it was also shown that marine
heterotrophic bacteria replace PE-DAG with either 1G-DAG or DGTS (Carini et al., 2015; Sebastian et
al., 2016; Yao et al., 2015).    Such a dynamic response of membrane lipid composition to nutrient
limitation has also been demonstrated for mixed planktonic communities in the environment (Van Mooy
and Fredricks, 2010; Popendorf et al., 2011b).

In this study we use a complementary approach to study the microbial populations inhabiting the

OMZ of the ETNP via the analysis of eukaryotic, bacterial and archaeal IPLs in SPM.    This study is an
extension of Xi et al. (2014), which focused on the detailed distribution of core and intact polar archaeal
and bacterial tetraether lipids at two of stations (station 1 and 8) described here.    There are distinct
biogeochemical zonations within the water column of the ETNP based on IPL distributions which must
reflect the localized ecology.    Abundant non-phosphorus (substitute) lipids within the core of the OMZ
might result from phosphorus limitation of the source microorganisms.    Overall our results should
provide deeper insight into the biogeochemical cycles and functioning of OMZs throughout the World
Ocean.

**2. Methods**
*2.1 Sample collection and CTD data*

Suspended particulate matter (SPM) samples were collected at four stations (distance to shore:

400~600 km;Fig 1) along a northwest-southeast transect (Station 1: 13°N,105°W; Station 2: 12° 14’N,



101° 13.74' W; Station 5: 10° 41.41' N,96° 56.6' W; and Station 8: 9°N,90°W) in the ETNP during the
R/V *Seward Johnson* cruise in November 2007 (R/V *Seward Johnson* Cruise Scientists, 2007).    Station
1 in the Tehuantepec Bowl is an area of relatively low primary productivity (e.g., 0.05 mg Chl-*a*/m$^2$;
(Fiedler and Talley, 2006; Pennington et al., 2006) whereas Station 8 in the Costa Rica Dome is moderately
productive (1 mg Chl-*a*/m$^2$).    All stations are characterized by a strong thermocline/pycnocline/oxycline
(at 20-50 m depths depending on location) and a profound and thick OMZ (down to ~2 μM $O_2$ between
~300-800 m depth).    Station 1 is a reoccupation of the Vertical Transport and Exchange (VERTEX) II
and II site that was intensely studied in the early 1980's (Martin et al., 1987), including several reports on
organic biogeochemistry there (Lee and Cronin, 1984; Wakeham and Canuel, 1988; Wakeham1987, 1989).

Seawater was filtered *in-situ* using submersible pumps (McLane Research Laboratories WTS-142

filtration systems) deployed on the conducting cable of the CTD/rosette (Seabird 3+ temperature sensor,
Seabird 9+ digital quartz pressure sensor, Seabird 4C conductivity sensor, Seabird 43 oxygen sensor, C-
Point chlorophyll fluorescence sensor, Wetlabs CST-721DR 25 cm path length transmissometer).
Volumes ranged between 130 and 1800 L.    Temperature, conductivity, fluorescence and dissolved
oxygen were measured during pump deployments and again during recovery; pump depths (4 pumps per
cast) were monitored from the CTD depth during pumping.    Pumps were fitted with two-tier 142 mm
diameter filter holders: a 53 μm mesh Nitex "prefiltration" screen to remove most eukaryotes and marine
snow aggregates and a double-stacked tier of ashed glass fiber filters (142 mm Gelman type A/E, nominal
pore size 0.7 μm).    It is likely that smaller cells (diameter 0.2-0.7 μm) were not retained on the filter and
thus the reported IPL concentrations represent minimum values.    After filtration, samples were wrapped
in pre-combusted foil and stored frozen at -20°C until extraction.

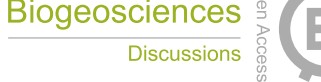


*2.2 Elemental, pigment and nutrient analysis*

Particulate organic carbon (POC) and total particulate nitrogen (TN) were measured on 14 mm-diameter subsamples of each GFF prior to lipid extraction; therefore, POC and TN concentrations reported here are only for <53 μm material.   The plugs were acidified in HCl vapor in a desiccator for 12 hours to remove inorganic carbon.   Elemental analysis was performed with a ThermoFinnigan Flash EA Series 1112 interfaced to a ThermoFinnigan Delta V isotope ratio mass spectrometer at the Skidaway Institute Scientific Stable Isotope Laboratory (SISSIL).   Organic carbon and nitrogen contents were calibrated against internal laboratory chitin powder standards which in turn had previously been cross-calibrated against USGS 40 and 41 international standards.

Two sets of pigment analyses were conducted.   Chlorophyll-*a* (Chl-*a*) and pheopigment concentrations were measured on-board the ship (Olson and Daly, 2013).   Seawater samples (100 – 500 ml) from CTD casts were filtered onto Whatman GF/F filters which were immediately extracted with 90% acetone.   Fluorescence was measure with a Turner Designs 10AU fluorometer and Chl-*a* concentrations were determined after Parsons et al (1984).   Post-cruise HPLC analysis of pigments in 100-500 ml seawater samples filtered onto Whatman GF/F filters were conducted at the College of Charleston Grice Marine Laboratory, Charleston, SC on a Hewlett Packard 1050 system (DiTullio and Geesey, 2002).

Seawater samples for nutrient analyses ($NO_2^-$, $NO_3^{-2}$, $NH_4^+$) were collected directly from Niskin bottles into acid-washed, 30-mL high-density polyethylene (HDP) bottles.   After three rinses, bottles were filled to the shoulder, sealed, and frozen (−20°C).   All frozen samples were transported to the Oceanic Nutrient Laboratory at USF for analysis using a Technicon Autoanalyzer II.



*2.3 Lipid extraction and analysis of intact polar lipids*


Lipids associated with the <53 µm SPM on the GFFs were Soxhlet-extracted using


dichloromethane:methanol (DCM:MeOH; 9:1 v/v) for 8 h (Wakeham et al., 2007). Extracted lipids were


partitioned into DCM against 5% NaCl solution and dried over $Na_2SO_4$. Total lipid extracts (TLEs) were


stored at -20°C.


An aliquot of the TLE was dissolved in DCM/Methanol (5:1 v/v) for analysis on a ThermoFinnigan


Surveyor high-performance liquid chromatography system coupled to a ThermoFinnigan LCQ DecaXP


Plus ion-trap mass spectrometer via electrospray interface (HPLC-ESI-IT-MS[n]) using conditions


described previously (Sturt et al., 2004).   Ten µL of an aliquot spiked with $C_{19}$-PC as internal standard


was injected onto a LiChrosphere Diol-100 column (150 × 2.1 mm, 5 µm, Alltech, Germany) equipped


with a guard column of the same packing material.   These data were used to obtain IPL concentrations.


Sum formulas and IPL structures were assigned based on exact masses in the MS1 and MS-MS


experiments during analysis of an aliquot of the TLE on a Bruker maXis Plus ultra-high resolution


quadrupole time-of-flight mass spectrometer (Q-TOF) with an ESI source coupled to a Dionex Ultimate


3000RS UHPLC.   Separation of IPLs was achieved using a Waters Acquity UPLC BEH Amide column


as described in (Wörmer et al., 2013).   Selected samples were measured in positive and negative


ionization modes with automated data-dependent fragmentation of base peak ions.   Acyl moieties of


glycolipids and aminolipids were identified via HPLC-IT-ESI-MS[2] experiments in positive ionization


mode, and of phospholipids in negative ionization mode.   Details of mass spectral interpretation, and


identification of fatty acids moieties are described in Sturt et al. (2004) and Schubotz et al. (2009).   For




all analyses, response factors of individual IPLs relative to the injection standard $C_{19}$-PC could be
determined for all major and some minor IPLs using commercially available standards: 1G-DAG, 2G-
DAG, SQ-DAG, 1G-CER, DGTS, PG-DAG, DPG, PE-DAG. PME-DAG, PDME-DAG, PC-DAG, PC-
DEG (Avanti Polar Lipids, USA) and 1G-PG-GDGT (Matreya LLC, USA). Some ions assigned to either
PE-AEG and PC-AEG could not be quantified individually due to co-elution of these compounds and
were thus quantified as one group using the mean response factor of PE- and PC-DAG.    For compound
classes for which no standards were available, such as PI-DAG, OL and the unknown aminolipids AL-I
and AL-II the relative response could not corrected for and values may thus be off by a factor of 0.2 to
1.4, which is the maximum range of response factors observed for the standards.

*2.4 Statistical analysis*

Nonmetric multidimensional scaling (NMDS) analysis was used to illustrate the relationships

among objects hidden in a complex data matrix (Rabinowitz, 1975) and was performed in the free software
R (version 3.4.3, www.r-project.org/) with *metaMDS* (vegan library, version 2.4-6) as described by
Wakeham et al. (2012).    The datasets of relative lipid distribution and variations in chain length and
unsaturation were standardized by Hellinger transformation using the function *decostand*, while for all
other variables (environmental parameters, microbial groups) absolute numbers were used.    The
compositional dissimilarity was calculated by Euclidean distance measure.    The resulting plot shows the
distribution of lipids and sampling depths.    Microbial groups and geochemical parameters were overlaid
by function *envfit*. Lower stress is related to high quality of solution, and normally stress $\leq 0.1$ is a
guideline for good quality results (Rabinowitz, 1975).    Non-parametric Spearman Rank Order





Correlation analysis was performed on combined data of environmental variables and IPL ratios and IPL
relative abundances of all four stations using SigmaPlot 11.0 (Systat Software Inc., San Jose, USA).

**3. Results**
*3.1 Biogeochemical setting*
All along the transect, the thin mixed layer (upper ~20 m) was warm, ~25 − 28 °C, with oxygen
concentrations approaching air saturation at ~200 μM (Fig. 1b,c). The thermocline was abrupt at ~20-
50 m, where temperatures dropped to ~15 − 18 °C and oxygen decreased to ~20 μM. Temperatures
stabilized by ~250 − 300 m depth at ~10 − 12 °C and oxygen levels were <2 μM; especially at Station 8
there were spatially and temporally variable oxygen intrusions into the upper portion of the OMZ. By
~600 − 800 m depth, a deep oxycline was observed where oxygen concentrations began to rise again to
~40 μM at temperatures of ~4 °C by 1250 m. Thus at the time of our cruise and for the purpose of this
discussion, the water column of the ETNP could be roughly compartmentalized into four horizons based
on oxygen content: an oxic epipelagic zone down to the thermocline (0 − 50 m; 200 μM > $O_2$ > 20 μM);
an upper OMZ (Station 1 and 8: 50 – 300 m, Station 5: 50 – 350 m, Station 2: 50 – 200 m; 20 μM > $O_2$ >
2 μM); the core OMZ (Station 1 and 8: 300 – 800 m, Station 5: 350 – 600 m Station 2: 200 – 600 m; $O_2$
< 2 μM); and a deep oxycline (Station 1 and 8 ≥ 800 m, Station 2 and 5 ≥ 600 m; $O_2$ > 2 μM) of rising $O_2$
levels (Fig. 1a).
Chl-$\alpha$ was highest in surface waters with maximum values of 1.8 μg/L at 10 m at station 5, were
between 0.2 and 0.7 μg/L at station 2, 5 and 8 and decreased to values close to zero below 100 m at all
stations (Fig. 1d; see also Fiedler and Talley, 2006, and Pennington et al., 2006, for additional results from



previous surveys). HPLC analysis of accessory pigments showed that picoplankton, primarily
*Prochlorococcus* (indicated by divinyl chlorophyll *α*), were an important component of the
photoautotrophic community, along with diatoms (fucoxanthin), especially *Rhizosolenia* at the deep
fluorescence maximum at stations 1 and 5 but *Chaetoceros* at station 8, and prymnesiophytes
(19'hexanoyloxyfucoxanthin and 19'butanoyloxyfucoxanthin; Suppl. Table 1). High phaeopigment
abundances (up to 90% of [Chl-*α* + phaeopigments]) attested to algal senescence or grazing by macro-
(Wishner et al., 2013; Williams et al. 2014) and micro-zooplankton (Olson and Daly, 2013) above and into
the oxycline.    Whereas the primary maxima in transmissivity (Beam C) corresponded with the peak Chl-
*α* concentrations and fluorescence maxima, secondary transmissivity maxima were observed between 300
and 400 m at stations 1, 5, and 8 indicating elevated particle abundances in the core of the OMZ (Fig. 1e).
Significant nitrite ($NO_2^-$) maxima were observed in the OMZ at all stations coinciding with nitrate
($NO_3^{2-}$) deficits (Fig. 2a,b).    Ammonium ($NH_4^+$) concentrations remained rather constant through the
water column (Fig. 2c).    Phosphate ($PO_4^{3-}$; Fig. 2d) and total dissolved nitrogen (TDN; not shown) were
low (respectively, < 0.5 and < 3 µM) in the upper 20 m of the oxic zone, but concentrations increase in
the OMZ; both were again low at the deep oxycline at station 1 (<1 µM for $PO_4^{3-}$ and<10 µM for TDN).
In contrast high $PO_4^{3-}$ (up to 3.4 µM) and high TDN (up to 44.5 µM) were observed in the deep OMZ at
stations 2, 5 and 8 (Fig. 2d).    N:P ratios were lower than the Redfield ratio (16) at all sites and depths
(Fig. 2e); N:P minima were observed in surface waters (2.6 to 10 in the upper 20 m) and at ~500 m within
the core OMZ and the deep oxycline at station 1 (<9).
POC and TN concentrations (< 53 µm material) were highest in the euphotic zone (POC: 20 –100
µg/L; TN: 4 – 15 µg/L), rapidly dropping to 5 µg/L and 1 µg/L below the upper OMZ, respectively (Fig.



3a; Suppl. Fig. 2).   There were slight secondary maxima for POC (~10 µg/L) and TN (~2 µg/L) within
the core of the OMZ that might reflect elevated microbial biomass there (see below).   Concentrations
again dropped in the deep oxycline, ≤3 µg/L and ≤0.5 µg/L for POC and TN, respectively.
IPL concentrations between 250 and 1500 ng/L were measured in the oxic epipelagic zone, and
abruptly decreased more than 10-fold (to <20 ng/L) in the upper OMZ, following the decrease of $O_2$ levels
(Fig. 3b). Secondary maxima in IPL concentrations (15 - 40 ng/L) within the OMZ at all stations roughly
coincided with elevated numbers of prokaryotes (Fig. 3d). IPL/POC ratios decreased with increasing depth
(Fig. 3c) and track trends of POC, TN and IPL concentrations.

*3.2 Changes in IPL composition with water column depth in the ETNP*
In total, 24 IPL classes were identified in the ETNP (Fig. 4).   Based on their head group composition
these were grouped into glycolipids, phospholipids or aminolipids.   Figure 5 shows changes in the
relative abundance of non-isoprenoidal (i.e. non-archaeal) glycolipids, phospholipids and aminolipids
along the transect as well as select substitute lipid ratios (cf. Van Mooy et al., 2006; Popendorf, et al.,
2011b; Carini et al., 2015).   Relative abundances of phospholipids (as percentage of total measured IPLs)
were highest in the core OMZ between 400 and 600 m at all sites, where they comprise up to 45-76% at
stations 1, 2 and 5 and between 12 and 61% at station 8.   Lower phospholipid abundances were observed
within the upper OMZ and oxic zone at all stations (between 4 and 55%) and in the deep oxycline at
station 8 (<1%).   Aminolipid content was highest in SPM from the upper 55 m at station 5 and 8 (10 to
25%), the core OMZ at station 8 (15 to 34%) and the deep oxycline at station 1 (17%).   Lower aminolipid
contents (2 to 11%) were observed in the oxic zone and the core OMZ at stations 1 and 2, the upper OMZ





at station 5 (0 to 11%) and the deep oxycline at station 8 (<2%).   Glycolipid abundance was >9% at all
depths, with highest abundance (average 54%, max. 82%) within the upper OMZ and oxic zone at all
stations and the deep oxycline at station 8.   Values down to 9% were observed within the core OMZ.

The phospholipid substitute ratio SQ-DAG to PG is based on the observation that cyanobacteria

biosynthesize SQ-DAG preferentially over PG when phosphorus limited (Benning, 1993; Van Mooy et
al., 2006, 2009) and is here for the purpose of this discussion extended to other bacteria and eukarya that
are probable sources of IPLs in subsurface waters.   At the ETNP this ratio ranged between 1 and 10
within the upper 100-200 m along the transect and is <1 deeper into the OMZ.   The ratio of DGTS to PC
is reflective of the algal response to phosphorus limitation since it was observed that microalgae and some
bacteria substitute PC-DAG with DGTS when phosphate concentrations are low (Van Mooy et al., 2009;
Zavaleta-Pastor et al., 2010).   At the ETNP this ratio did not show consistent trends and ranges between
0.4 and 2.4 at most depths, but with notable spikes (>30) within the upper core OMZ at station 2 and 8, in
the oxic zone at station 5 and in the lower portion of the core OMZ at station 8.   Similarly, the ratio of
1G-DAG to PE, which has been recently proposed to reflect the response of heterotrophic bacteria to
phosphorus limitation (Carini et al., 2015) did not show any consistent trend but generally ranges between
0.2 and 6 at most depths except for highly elevated values within the upper OMZ at station 2, 5 and 8 with
ratios up to 800 and within the deep oxycline at station 8, where 1G-DAG:PE ratios range between 650
and 950 (Fig. 5).

*3.2.1 Major lipids*

Major lipids are defined here as those IPL compound classes that comprised more than 10% of total





IPLs at more than one depth at the four stations. Eleven major IPL classes were identified. Three are
assigned to an archaeal origin: 1G-GDGT, 2G-GDGT and HPH-GDGT (Suppl. Fig. 1; Fig. 4).   A
previous study on archaeal lipid distributions in these same ETNP samples (Xie et al., 2014) reported on
only the two glycosidic archaeal IPLs (1G-GDGT and 2G-GDGT).   In that study the TLE had been
separated into fractions and only the glycosidic fractions had been analyzed for GDGTs.   Subsequent re-
examination of the original LCMS data indicate that HPH-GDGT were indeed present in an unanalyzed
fraction (fraction 3 in Xie et al., 2014).   The remaining eight major IPLs were assigned to either a
bacterial or eukaryotic origin and were three glycolipids (1G-DAG, 2G-DAG, SQ-DAG), four
phospholipids (PG-DAG, PE-DAG, PC-DAG, PE+PC-AEG) and one aminolipid (DGTS).   All major
lipid classes were found at all four stations and most of them occur at all depths, but with varying relative
abundances (see also Suppl. Table 2).
*Archaeal lipids*: Relative archaeal IPL abundances generally increased with depth from non-
detectable in surface waters to >50% of total IPLs at station 8 (bottom of core OMZ and deep oxycline),
maximally 30% at station 1 and 2 (bottom of upper OMZ, core OMZ and deep oxycline) and generally
<10% at station 5 (Fig. 4).   At station 1 and 2, 1G-GDGT and 2G-GDGT were most abundant with
variable amounts of HPH-GDGTs at different depths, while 1G-GDGT and HPH-GDGT dominated
archaeal IPLs at station 5 and 8 at most depths.   In terms of the distribution of glycosidic IPL-GDGTs
these results corroborate the absolute values reported by (Xie et al., 2014) where 1G-GDGT was more
abundant than 2G-GDGT at station 8 when compared to station 1.
*Diacylglycerol lipids*: The oxic epipelagic zone and the upper OMZ were dominated at all sites by
the three diacylglycerol glycolipids, 1G-DAG, 2G-DAG and SQ-DAG, except for station 8, where the



sum of the three glycolipids amounted to ≤50% (Fig. 4).    In the core OMZ and deep oxycline, relative
amounts of 2G-DAG and SQ-DAG decrease, with maximal abundances of 4% and 12%, respectively. 1G-
DAG amounts were lowest in the core OMZ at all stations, but increased again up to 47% of total IPL in
the deep oxycline.    Diacylglycerol phospholipids, PE-, PG- and PC-DAG were the second most abundant
compound class group at almost all stations and depths.    Average abundances of PE- and PG-DAG
became highest within the upper and core OMZ, comprising over 50% in the core OMZ at station 1, >30%
at stations 2 and 5, and 16% at station 8.    PC-DAG did not exhibit a depth related trend and had average
abundances of 5% at stations 1, 2, 8 and 3% at station 5.    The third most abundant diacylglycerol class
was the betaine lipid DGTS, which was present throughout the water column with average abundances of
7% at station 1, 2 and 8, and 5% at station 5.

All of the major diacylglycerol lipids showed changes in average chain lengths and double bond

numbers with depth (Fig. 6, Suppl. Table 3). The three glycolipids and PC-DAG exhibited a decrease in
average length of up to three carbons and a decrease in the number of double bonds of up to 2 at the top
of the upper OMZ and within the core OMZ compared to the oxic zone and the deep oxycline. Average
chain length for the phospholipids PE- and PG-DAG and the betaine lipid DGTS showed an inverse profile
to this, both increasing up to two carbons from the upper OMZ to the core OMZ in the average chain
length.    Changes in the number of double bonds were not as pronounced for PG-DAG and DGTS, but
they had on average 1 to 2 double bonds more in surface waters than in deeper waters, while the number
of double bonds increased on average with depth for PE-DAG.

*Acyl-ether glycerol lipids*: Mixed ether-ester glycerol core structures were observed with either PE

or PC head groups at all stations and all depths (except for the deep oxycline at station 8) with average




abundances ranging between 4 and 8%.

*3.2.2 Minor lipids*

Minor lipids are defined as those IPL compound classes that comprised less than 10% of total IPLs

at more than one depth of the four stations.    In total 13 minor IPL classes were identified, five of which
were glycolipids, four phospholipids and four aminolipids.    All types of minor lipids could be detected
at all four sites except for OH-DGTS which was not detected at Station 1.    For some of these minor lipids
water column stratification within the distinct zones could be observed, but some were detected at all
depths (Fig. 4).

*Diacylglycerol lipids:* Two diacylglycerol glycolipids, 1G-OH-DAG and 3G-DAG, were detected

mainly within the oxic zone and the upper OMZ, comprising between 2 to 15% of all minor lipids on
average (0.1 to 0.6% of total IPLs), but also selectively reappeared within the core OMZ and deep oxycline.
1G-OH-DAG showed highest relative abundances at station 5, comprising up to 40% of all minor lipids.
Four phospholipids with diacylglycerol core structures were identified with the following head groups:
diphosphatidylglycerol (DPG), phosphatidyl (*N*)-methylethanolamine (PME), phosphatidyl (*N,N*)-
dimethylethanolamine (PDME) and phosphatidyl inositol (PI). DPG, PME-DAG and PDME-DAG had
highest relative abundances (respectively 65, 56 and 35%) within the upper and core OMZ, but were also
present within the oxic zone at all stations and the deep oxycline at station 1, 2 and 5.    PI-DAG was most
prominent in the oxic zone and the upper OMZ (up to 25%), but was also present in the core OMZ and
the deep oxycline, except for station 8.    Three types of aminolipids were observed as minor lipids in the
ETNP.    OH-DGTS with up to three hydroxyl-groups attached to the fatty acyl side chains (Suppl. Fig. 3)



was mainly observed at station 8 at almost all depths with an average relative abundance of 23% among
the minor lipids and was selectively detected at station 2 and 5 within the oxic zone and upper OMZ.
Two other aminolipids had an undefined head group that exhibited fragmentation patterns characteristic
of betaine lipids, but without established betaine head group fragments (Suppl. Fig. 4b, c).    The
tentatively assigned sum formula for the head group of the first unknown aminolipid (AL-I) at ca. 6.7
minutes LC retention time was $C_8H_{17}NO_3$ and for the second unknown aminolipid (AL-II) at 10.5 minutes
was $C_7H_{15}NO_3$.    The head group sum formula for AL-II matches that of DGCC, however, the diagnostic
head group fragment of m/z 252 was not detected, and furthermore, AL-II did not elute at the expected
earlier retention time for DGCC.    AL-I and AL-II were detected at almost all depths at all four stations,
with average abundances of 1 to 6% for AL-I and comparably higher relative abundances of AL-II ranging
from 16 to 36%.
*Acyl-ether glycerol lipid*: One minor compound that eluted slightly earlier than SQ-DAG with a
similar fragmentation pattern as SQ-DAG but with exact masses of the parent ion and MS-MS fragments
in both positive and negative ion mode pointing to a mixed acyl-ether glycerol core lipid structure (Suppl.
Fig. 4d, e).    We therefore tentatively assigned this lipid to SQ-AEG, this IPL was observed at all four
stations at almost all depths with highest relative abundances of 5 to 60% within the oxic zone.
*Sphingolipids*: Two types of sphingolipids were identified, monoglycosyl ceramide (1G-CER), and
hydroxylated monoglycosyl ceramide (1G-OH-CER) with up to two hydroxyl groups attached to the
hydrophobic side chains (Suppl Fig. 3e).    Both of these lipid classes were observed at all depths at
stations 1, 2, and 5 at average relative abundances between 3 and 8%, but were not detected in the deeper
part of the core OMZ and the deep oxycline at station 8.





*Ornithine lipids*: Ornithine lipids were detected in trace amounts (<4%) in the core OMZ of station
2 and 5.

*3.2.3 Relationships between environmental parameters and lipid distribution*
Spearman Rank Order Correlation was used to evaluate relationships between relative lipid
abundance of lipid classes and environmental parameters (Table 1).   Glycolipids 2G- and SQ-DAG
showed highly significant (p<0.001) and positive correlations with depth, fluorescence, POC, TN,
temperature and Chl-$\alpha$, significant positive correlations were also observed with oxygen.   Both
glycolipids showed highly significant and negative correlations with phosphate and nitrate, and these
overall trends were mirrored in the SQ-DAG:PG-DAG ratio.   Total glycolipids (GL) and 1G-DAG only
showed correlations with a few environmental parameters and total GL were only significantly positively
correlated with oxygen.   Most aminolipids and phospholipids did not show significant correlations with
environmental parameters and most other correlations were neither strongly positive nor negative.
Relative abundances of total aminolipids and aminolipid (AL) to phospholipid (PL) ratios correlated
positively with ammonium.   AL:PL also correlated positively with oxygen.   Relative abundance of total
phospholipids and most individual phospholipids (PG-, PE-, PME-, and PDME-DAG) correlated
negatively with oxygen.   The only phospholipid that significantly correlated with phosphate was PDME,
however, the positive correlation is not very strong ($r^2$<0.4).
NMDS analysis revealed that all samples from the oxic zone had a negative loading on the NMDS2
axis along with environmental variables such as oxygen, fluorescence, TN, POC and Chl-$\alpha$.   IPLs with
a strong negative loading on the NMDS2 axis (<-0.2) were 1G-OH-DAG, SQ-AEG, 2G-DAG, SQ-DAG,



PI-DAG and OH-DGTS.    Most samples from the core OMZ and deep oxycline had a positive loading on
the NMDS2 axis, together with depth, phosphate and nitrate.    IPLs that showed a strong positive loading
on the NMDS2 axis (>0.2) were PDME-DAG, 2G-GDGT, DPG, PME-DAG and HPH-GDGT.    Almost
all environmental variables had low $p$-values (<0.001), indicating highly significant fitted vectors with the
exception of temperature, salinity, ammonium and nitrate.    Highest goodness of fit statistic was observed
with oxygen ($r^2$=0.54), followed by phosphate ($r^2$=0.48) and then fluorescence ($r^2$=0.46).

**4. Discussion**

The study area in the ETNP is characterized by moderate primary productivity in surface waters that
is coupled with intense microbial degradation of particulate organic matter exported to the thermocline
and restricted midwater oxygen replenishment produces the strong, shallow (~20 m deep) oxycline and a
~500 m thick OMZ with dissolved oxygen concentrations of <2 μM, not unlike other oceanic OMZs (e.g.,
coastal upwelling off Peru and Namibia, Arabian Sea, Cariaco Basin and Black Sea; e.g. Ulloa et al.,
2012).    Oxygen gradients in the ETNP thus create stratified microhabitats for metazoans and microbes.
Micro-grazers are critical for trophic transfer and remineralization (Sherr and Sherr, 1994) in systems like
the ETNP that are dominated by picoplankton; indeed, micro-grazers in the eastern equatorial Pacific have
been reported previously to consume in excess of 100% of phytoplankton production (Landry et al., 2011).
During our ETNP expedition, Olsen and Daly (2013) found that micro-grazing removed 33 and 108% of
surface primary production in the upper mixed layer.    Peak macrozooplankton biomass was located at
the thermocline, near the upper boundary of the OMZ, where food resources would be most available, but
a secondary biomass peak of different zooplankton assemblage was present at the deep oxycline once $O_2$



concentrations rose to ~2 µM (Wishner et al., 2013; cf. Wishner et al., 2008 for a comparable Arabian Sea
investigation).    Carbon and nitrogen stable isotopes in ETNP SPM (splits of the same SPM filters used
for IPL analyses) and zooplankton suggest that shallow-water, plankton-derived particulate organic carbon
is the primary food source for zooplankton in the mixed layer, upper oxycline and core OMZ, whereas
deep POC, some of which might have been produced by microbes in the OMZ, is important for deep
oxycline zooplankton (Williams et al., 2014).

Microbial community structure and activities in the ETNP as part of our expedition were investigated

via CARD-FISH by Podlaska et al (2012).    Cell numbers of total prokaryotes that are highest in the
euphotic layer decreased with depth at the thermocline but rose again within the core OMZ (Fig. 2 of
Podlaska et al., 2012), these are observations typical for other oxygen-deficient regions such as the
upwelling area off the coast of Namibia (Woebken et al., 2007) and anoxic basins, e.g., the Black Sea and
Cariaco Basin (Taylor et al., 2001; Lin et al., 2006; Wakeham et al., 2007, 2012).    Bacteria dominate the
prokaryotic community at all stations, but archaeal abundance can be as high as 50% and 26% at the
bottom of the OMZ at stations 2 and 5, respectively.    Heterotrophic activity, measured by uptake of
leucine, was prevalent in and above the thermocline/upper oxycline where reactive organic matter is most
available. Elevated rates of chemoautotrophy, measured by dark dissolved inorganic carbon (DIC)
assimilation, were observed at several depths in the OMZ and in the lower oxycline.    Dark DIC
assimilation correlated with total prokaryote abundances.    Mid-water microbial chemoautotrophy was
further indicated by stable carbon isotope values for POC in the upper and core OMZ that are depleted by
2 to 6‰ at the upper oxycline and within the OMZ compared to $\delta^{13}C$ values of -24 to -21‰ for surface
water plankton (Podlaska et al., 2009).    Transfer of chemoautotrophically-fixed carbon into zooplankton

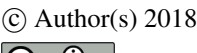



food webs is also evident (Williams et al., 2014). Nitrifying bacteria (nitrite-oxidizers as Nso1225-
positive cells) constituted 3-7% of total DAPI-positive prokaryotes, with surface water peaks where nitrate
was not detectable and in the upper OMZ where ammonium was depleted but nitrate and nitrate were high.
Sulfate-reducing bacteria (SRB358-positive cells) were detected between 17 and 34% of total prokaryotes
in the upper OMZ and deep oxycline where they might be associated with anoxic microzones within
particle aggregates even at low dissolved oxygen concentrations (Woebken et al., 2007; Carolan et al.,
2015). Similarly, anoxic microzones may be responsible for observed abundances of Planctomycetes
(Pla46-positive cells; up to 24% to total prokaryotes), and anammox bacteria (Amx368-positive cells; <1%
of prokaryote numbers; Podlaska et al., 2012) and ladderane lipids in the OMZ correspond with secondary
nitrite maxima. Nitrate deficits point to nitrate reduction as a source for nitrite in the OMZ. Archaeal
cell abundances peak at the start of the upper OMZ at all stations (reaching maximal values of 37% of
total prokaryotes at station 2), within the core OMZ at station 2 (up to 54% of total detected cells at station
2) and within the deep oxycline at station 5 and 8 (around 25%; Fig. 2e). These peaks in archaeal cells
are further corroborated by maxima in 1G- and 2G-GDGT abundances at 120 m and 725 m at station 1
and at 200 m and 550 m at station 8 reported by (Xie et al., 2014). Crenarchaeota and thaumarchaeota
(as Cren537-positive cells) represented ~20% of prokaryotes throughout the water column, generally
being highest in the lower OMZ and deep oxycline, and at stations 2 and 5 just above the secondary Chl-
*a* maxima at ~75 m. Euryarchaeota (Eury806-positive cells) were 16-20% of total prokaryotes,
especially in waters above the OMZ, and generally correlating with ammonium concentrations and
heterotrophic potential.

Total IPL concentrations that were over 50 times higher in the surface waters than at deeper depths





coincide with high Chl-$\alpha$ concentrations reflecting the importance of eukaryotic rather than microbial
sources to the IPL pool above the thermocline.    Below the thermocline, IPL concentrations generally
track trends observed in the cell abundances, with elevated IPL concentrations in the upper and core OMZ
whenever nitrite concentrations are elevated.    The rapid decrease in IPL concentrations below ~100 m
probably results from a combination of a dearth of potential source organisms and the decomposition of
detrital lipids (Harvey et al., 1986; Matos and Pham-Thi, 2009) associated with particulate matter sinking
from above. Substantial IPL concentration decreases below the euphotic zone have been observed
elsewhere (Van Mooy et al., 2006; Schubotz et al., 2009; Van Mooy and Fredricks, 2010; Popendorf et al.,
2011b; Wakeham et al., 2012).    Despite these low absolute concentrations of IPL, the diversity of
molecular compositions and significant changes in relative abundances of IPLs reflect a complex
eukaryotic and prokaryotic community structure throughout the water column of the ETNP.

*4.1 Sources for IPLs in the water column of the ETNP*
Distinct changes in IPL relative abundances and core lipid compositions coincide with the
biogeochemical zonation of the OMZ at all four stations, although many of the IPL were detected at
multiple depths.    Potential sources and possible physiological roles for the observed IPLs in the different
zones are discussed below.

*4.1.1 Oxic zone*
The glycosyldiacylglycerides that dominate the IPL composition in surface waters, 1G-DAG, 2G-
DAG and SQ-DAG, are major constituents of photosynthetic thylakoid and chloroplast membranes of



plants (Poincelot 1973, Mackender and Leech, 1974; Nishihara et al., 1980), eukaryotic algae (Araki et
al., 1991; Thompson, 1996) and cyanobacteria (Wada and Murata, 1998; Siegenthaler, 1998).    They are
commonly the most abundant IPLs in oceanic surface waters (Van Mooy et al., 2006; Schubotz et al.,
2009; Van Mooy and Fredricks, 2010; Popendorf et al., 2011b; Wakeham et al., 2012), where they are
assigned to photosynthetic algae or cyanobacteria.    The acyl groups of the glycolipids can give further
indications about their biologic sources. 1G- and 2G-DAG are dominated by $C_{16}$ and $C_{18}$ fatty acids in the
euphotic zone with zero to 4 double bonds (Suppl. Table 1, Fig. 6).    Many different combinations of
polyunsaturated fatty acids (PUFA) are observed, such as $C_{16:4}/C_{18:3}$, $C_{16:4}/C_{18:4}$, $C_{18:3}/C_{16:2}$, $C_{18:4}/C_{14:0}$ and
$C_{18:5}/C_{14:0}$, which are characteristic for marine phytoplankton (Brett and Müller-Navarra, 1997; Okuyama
et al., 1993).    SQ-DAG in the ETNP do not contain these PUFA, but instead have predominantly
combinations of $C_{14:0}$, $C_{16:0}$, and $C_{16:1}$ fatty acids, resulting in shorter chain lengths and a lower average
number of double bonds (0.5 to 1) than the other glycolipids in surface waters (Fig. 6).    This is consistent
with SQ-DAG being primarily derived from marine cyanobacteria that mainly have saturated and
monounsaturated $C_{14}$ and $C_{16}$ fatty acids (e.g., Siegenthaler, 1998).    Cyanobacteria are therefore likely
the primary source organisms for all three glycolipids in the euphotic zone and upper OMZ of the ETNP
as they were abundant from the surface waters into the upper OMZ (as indicated by divinyl chlorophyll $\alpha$,
a diagnostic pigment for *Prochlorococcus* cyanobacteria, Suppl. Table 1; see also Goericke et al., 2000;
Ma et al., 2009), notably at the secondary fluorescence maxima that were observed just below the
thermocline, especially at stations 1 and 8.    The PUFA fatty acids in 1G-DAG and 2G-DAG additionally
indicate mixtures of eukaryotic algae as source for these lipids.    The presence of eukaryotic algae, such
as diatoms (characteristic pigment: 19'hexanoyloxyfucoxanthin) and Prymnesiophytes (characteristic



pigment: 19'butanoyloxyfucoxanthin; Suppl. Table 1) albeit not as abundant as cyanobacteria, is also
indicated by the detection of PC-DAG with fatty acyl combinations of $C_{22:6}$ and $C_{20:5}$ long chain PUFA
and $C_{16:0}$ fatty acids (Suppl. Table 3).    These long chain PUFAs are common in many eukaryotic
phytoplankton (Brett and Müller-Navarra, 1997; Okuyama et al., 1993).    Stable carbon isotope labeling
experiments in the North Atlantic have confirmed the importance of a phytoplankton source for PC in
surface waters (Popendorf et al., 2011a).    Betaine lipids such as DGTS, which are also diagnostic
eukaryotic algal markers (Dembitsky, 1996; Popendorf et al., 2011a), are present in surface waters of the
ETNP in abundances similar to those of PC.    Major acyl moieties of betaine lipids were $C_{14}$, $C_{16}$, $C_{18}$ and
$C_{20}$ with multiple unsaturations (on average 1.5 to 3 double bonds).

Eukaryotic phytoplankton and cyanobacteria are assumed to be a major source for PG-DAG in the

euphotic zone as it is the only phospholipid in cyanobacteria and thylakoid membranes (Wada and Murata,
1998).    Popendorf et al. (2011a) have shown that in surface waters of the Atlantic heterotrophic bacteria
seem to be a dominant source for PG-DAG, which is consistent with it being a major phospholipid in
bacterial membranes (Goldfine, 1984).    Similarly, PE-DAG is a common phospholipid in membranes of
bacteria (Oliver and Colwell, 1973; Goldfine, 1984) and is also sometimes present in low abundances in
eukaryotic algae (e.g., Dembitsky et al., 1996).    Both of these sources have been confirmed for PE-DAG
in surface waters of the Atlantic (Popendorf et al., 2011a).    We therefore suggest heterotrophic bacteria
to be the major source for PE and PG-DAG in the euphotic zone of the ETNP, with cyanobacteria being
responsible for PG-DAG and heterotrophic bacteria for PE-DAG.    Lower average number of double
bonds in PG and PE-DAG (<2) is consistent with a primarily bacterially-derived source of these lipids in
the upper water column of the ETNP (Fig. 6).

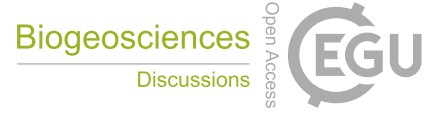

PE and PC also occurred with mixed acyl and ether lipids (AEG) in their core structures. PE-AEG
have been described in some sulfate-reducing bacteria (Rütters et al., 2001), which are, however, an
unlikely source in the oxic water column of the ETNP, unless in anoxic microzones of fecal pellets or
aggregates (e.g., Bianchi et al., 1992; Shanks and Reeder, 1993).    A recent study of surface waters of the
Southern Ocean and the eastern South Atlantic Ocean found increased abundances of 1-*O*-monoalkyl
glycerol ethers (MAGE), which are presumably breakdown products of PE- and PC-AEGs (Hernandez-
Sanchez et al., 2014).    These authors suggested aerobic bacteria to be a likely source for these lipids, but
cultured representatives are currently lacking to confirm this conclusion.    The relative abundance of
MAGE-based phospholipids, 4 to 7% of total IPLs, although low, highlights the overall significance of
these orphan lipids in the surface waters of the ETNP.    Similarly, some of the observed minor IPL
compounds have to date not been described in culture, including SQ-AEG.    Since SQ is a diagnostic
headgroup found in cyanobacteria, we suggest that cyanobacteria are a likely source for SQ-AEG in the
ETNP surface waters, although, again, these lipids have not been reported in cultured cyanobacteria.
3G-DAG, another minor IPL detected in the euphotic zone at all stations except for station 5 with up to
six double bonds and $C_{14}$, $C_{16}$ and $C_{18}$ fatty acid has been found in some plants (Hölzl and Dörmann, 2007)
and some anaerobic gram-positive bacteria (Exterkate and Veerkamp, 1969).    We thus propose that
eukaryotic algae are the probable source for this glycolipid in the euphotic zone of the ETNP.
Phospholipids detected in minor amounts in the euphotic zone include PI-DAG and DPG.    As these are
also minor components in several types of marine algae (Dembitsky, 1996) and bacteria (Morita et al.,
2010; Diervo et al., 1975; Mileykovskaya and Dowhan, 2009), mixed inputs are likely for these IPLs.
Likewise, bacteria in the oxic euphotic zone may be the origin of the low detected levels of *N*-methylated





phospholipids (Goldfine and Ellis, 1964).
1G-CER belongs to the class of sphingolipids consisting of a sphingosine backbone linked to a fatty
acid via an amide bond. 1G-CER was detected in the oxic zone as a minor component with relative
abundance less than 5% of IPL at all stations (Fig. 4). Glycosidic ceramides occur in eukaryotic algae
such as the coccolithophore *Emiliana huxleyi* (Vardi et al., 2009). We also detected 1G-OH-CER with
up to 2 hydroxylations in the core lipid structure (Suppl. Fig. 3). The presence of multiple-hydroxylated
sphingoid bases has been proposed as a marker for viral infection and cell death in at least some marine
phytoplankton, such as *Emiliania huxleyi* in the study of Vardi et al. (2009). We did not, however, find
mass spectral evidence for the presence of viral polyhydroxylated 1G-CER, as described by Vardi et al.
(2009) and therefore rather suggest that eukaryotic algal cells are potential sources for the G-CER (Lynch
and Dunn et al., 2004) in surface waters of the ETNP. Apart from the presence of hydroxylated
sphingolipids, a notable observation was the presence of hydroxylated glycolipids (1G-OH-DAG) and
aminolipids (OH-DGTS) with up to two hydroxyl-groups or one hydroxyl group combined with an expoxy
or keto function attached to the fatty acyl groups (Suppl. Fig. 3). The addition of hydroxyl groups or
general oxygenation of fatty acids in plants, algae and yeast has been identified as a defense mechanism
and response to oxidative stress (Kato et al., 1984; Andreou et al., 2009). Hydroxy fatty acids, for
example, are intermediates in oxidative degradation of fatty acids (Lehninger, 1970), and since they are
constituents of structural biopolymers of many microorganisms (Ratledge and Wilkinson, 1988), they are
present in marine particulate matter (e.g., Wakeham, 1999), derived from membrane constituents of Gram
negative bacteria, the most abundant bacteria in seawater (Rappé and Giovannoni, 2000).



*4.1.2 Upper OMZ*

Within the upper OMZ below the thermocline/oxycline, glycolipid abundance varied between 15 to 80% of total IPL, but notably SQ-DAG and 2G-DAG exhibited strong decreases in relative and absolute abundance below 125 m at all stations consistent with the decrease in phytoplankton biomass as seen in Chl-$\alpha$ profiles. Anoxygenic phototrophic bacteria might be a source of these glycolipids (Hölzl and Dörmann, 2007) in the upper OMZ, which overlapped with the base of the euphotic zone. Other anaerobic bacteria, including sulfate-reducing bacteria, are able to replace phospholipids with glycolipids, particularly 1G-DAG, when phosphate concentrations were <20 µM (Bosak et al., 2016). Sulfate-reducing bacteria (δ-proteobacteria) comprised up to 10% of the total bacterial communities in the upper OMZ (Podlaska et al., 2012). Other α-, β-, γ-, ε-proteobacteria were also abundant in the upper OMZ. Since aerobic representatives of these classes are capable of similar phospholipid substitutions (Geske et al., 2012; Carini et al., 2015; Sebastian et al., 2016; Yao et al., 2015), we infer that the presence of 1G-DAG could indicate P-limited proteobacteria within the upper OMZ, considering phosphate levels in the upper OMZ of the ETNP range between 2 to 2.5 µM. Notably, relative glycolipid abundance was very high within the upper OMZ, while phospholipid abundance was low (Figs. 2, 4), particularly at station 2 and 5 where glycolipids comprise >80% of total IPLs. Core lipid chain length and number of double bonds of all three glycolipids showed considerable variations within the upper OMZ (Fig. 6), indicating a change in source organisms compared to the oxic zone. Both PC-DAG and DGTS decreased in chain length and number of double bonds at most stations within the upper OMZ and became increasingly dominated by $C_{14}$, $C_{16}$ and $C_{18}$ saturated and monounsaturated fatty acids (Suppl. Table 3). This likely reflects a change from eukaryotic to bacterial sources for these IPLs. Similar to 1G-DAG, DGTS can



function as a substitute lipid under phosphorus limitation in some bacteria (Geiger et al., 1999; Sebastian
et al., 2016) and we thus suggest that phosphate concentrations around 2 μM induce such ecological
changes in the water column of the ETNP.   AL-I and AL-II are other non-phosphorus-containing IPL
that remain abundant within the minor IPLs in the upper OMZ.   Since we could not elucidate the structure
of theses lipids (Supp. Fig. 4) potential bacterial source(s) remain unclear.   Phospholipids PME- and
PDME-DAG and DPG increase in abundance within the upper OMZ and indicate an increased abundance
of bacteria at these depths, as has been observed in other stratified water bodies (Schubotz et al., 2009;
Wakeham et al., 2012).

G-CER and 1G-OH-CER were detected in the upper OMZ in similar relative amounts as in the oxic

zone but microbial sources for these IPLs in suboxic environments remain unclear.   Their abundant
presence in the anoxic zones of the stratified Black Sea (Schubotz et al., 2009) and Cariaco Basin
(Wakeham et al., 2012) has been previously assigned to as yet unidentified anaerobic bacteria.   The
oxygenated glycolipids and betaine lipids that were observed in the oxic zone are also present within the
upper OMZ at most of the stations and thus underline a likely bacterial source of these currently
unassigned IPLs and potentially signify oxidative stress or other acting defense mechanisms, (cf. Kato et
al., 1984; Andreou et al., 2009).

Archaeal IPLs with glycosidic headgroups and tetraether core structures (1G- and 2G-GDGT) were a

greater proportion of the overall IPL pool within the upper OMZ.   As shown by Xie et al. (2014) in an
earlier analysis of these same samples, absolute abundances of glycosidic GDGTs peak roughly at depths
where nitrite maxima are observed, consistent with the source archaea being ammonium oxidizers,
comparable to other OMZs around the world (e.g., Pitcher et al., 2011; Schouten et al., 2012).   At all



stations GDGTs with hexose-phosphate-hexose (HPH) headgroups were observed at depths of nitrate
maxima, but at deeper depths as well.    HPH headgroups are common among ammonium oxidizing t

haumarchaeota (Elling et al., 2017).    Other archaeal sources such as Marine Group II euryarchaeota

(Zhu et al., 2016; Lincoln et al., 2014), are possible for the observed glycosidic GDGTs, since Podlaska
et al. (2012) detected both crenarchaeota/thaumarchaota and euryarchaeota within the upper OMZ of the
ETNP.

*4.1.3 Core OMZ and deep oxycline*

IPL distributions in the core OMZ and at the deep oxycline of the ETNP were notably different from

the oxic zone and the upper OMZ.    In general, phospholipid abundance increased at all stations to over
50% (except for station 8) while glycolipid abundance decreased. Chain length and number of double
bonds were distinctly different within the core OMZ compared to the overlying water column (Fig. 6).
For instance, the PUFA that were observed in the euphotic zone and that are widespread in marine
phytoplankton (Brett and Müller-Navarra, 1997; Okuyama et al., 1993) became less abundant at the deeper
depths (Suppl. Table 3), indicating either the decline of sources or rapid degradation of these labile, highly
unsaturated compounds (De Baar et al., 1983; Prahl et al., 1984, Neal et al., 1986).    Degradation is the
more likely scenario since marine bacteria, including deep-sea species, are capable of biosynthesizing
PUFAs (DeLong and Yayanos,1986, Fang et al. 2003; Valentine and Valentine, 2004).    PE and PG-DAG
were the most abundant phospholipids in the core OMZ, followed by PC-DAG and PE- and PC-AEG.
We interpret the increase in phospholipid abundance as due to the increase in bacterial abundance within
the OMZ.    This is also reinforced by the increase of DPG, PME and PDME-DAG among the minor lipids



(Fig. 4).   Multiple bacterial sources are possible since PE, PG and DPG are common phospholipids in
membranes of most proteobacteria (Oliver and Colwell, 1973; Goldfine, 1984), and genes for the synthesis
of PME, PDME and PC are widespread among α-, γ- and some β-proteobacteria, all of which were
abundant within the core OMZ and deep oxycline (20 to 40% of the total bacterial population, Podlaska
et al., 2012).   Fatty acid combinations for the detected phospholipids included saturated $C_{14:0}$, $C_{15:0}$ and
$C_{16:0}$ and monounsaturated $C_{16:0}$ $C_{17}$ and $C_{18:0}$ (Suppl. Table 3).   The increased proportion of odd-chain
fatty acids further underpins a bacterial origin for these phospholipids.

The three glycolipids that dominate surface water IPLs were also present in the core OMZ and most

parts of the deep oxycline, although at greatly reduced concentrations.   Since chain length and number
of double bonds were distinct from the surface waters, with on average 1 to 2 carbon atoms shorter chain
lengths and 1 to 3 fewer double bonds at all sites (Fig. 6), we infer a microbial source for these glycolipids
within the OMZ.   In the core OMZ and deep oxycline SQ-DAG with combinations of fatty acids with
odd numbers of carbon atoms (e.g., $C_{15:0}/C_{16:0}$ and $C_{14:0}/C_{15:0}$) further support a bacterial source for this
IPL, despite its widespread attribution as cyanobacterial marker in environmental studies.   Indeed, SQ-,
1G- and 2G-DAG have been reported in some members of the Gram-positive Bacillus and Firmicutes
(Hölzl and Dörmann, 2007) and their presence in deeply buried Wadden Sea sediments has been ascribed
to an anaerobic bacterial source (Seidel et al., 2012). Unfortunately, Gram-positive bacteria were not
specifically targeted in previous phylogenetic characterizations of the OMZ of the ETNP (Podlaska et al.,
2012).   However, in the OMZ of the eastern tropical South Pacific Gram-positive bacteria such as
Actinobacteria accounted only for a negligible amount of total prokaryotic community (Stevens and Ulloa,
2008) and are thus likely not contributing significantly to the glycolipids in the core OMZ.



Aminolipids, DGTS and betaine lipid like AL-II, were observed in similar relative abundances in the
core OMZ and deep oxycline as in the overlying shallower water column.    The presence of DGTS has
so far only been reported in a few aerobic proteobacteria, and then only when grown under phosphorus
limitation (Benning et al., 1993; Geiger et al., 1999; Sebastian et al., 2016).    Consequently, potential
bacterial sources for the aminolipids in the core OMZ remain elusive, particularly since these regions are
not considered to be phosphorus limited with phosphate concentrations exceeding several micromolar (Fig.

2).

Bacterial sources for 1G-CER and 1G-OH-CER within the core OMZ and deep oxycline remain
similarly unresolved, but as suggested for anoxic water columns (Schubotz et al., 2009; Wakeham et al.,
2012), uncultured anaerobic bacteria are potential source organisms.    Ornithine lipids, also non-
phosphorus containing lipids but which are not known to play important roles in lipid substitutions (cf.
Geiger et al., 2010), were present in minor to trace amounts within the core OMZ and can be assigned to
Gram-negative bacteria.
For several of the major IPLs, such as 2G-DAG, PC-DAG and DGTS, the average chain length and
number of double bonds increased again to levels observed in surface waters within the deep oxycline
layer (Fig. 6).    PC-DAG and DGTS both contained long-chain PUFA, specifically in the case of DGTS
and 2G-DAG, $C_{16:4}$ and $C_{18:3}$.    This could thus reflect an exported fossil signal from the surface water,
however, since experimental studies of IPL degradation, including PC, have not provided evidence for
selective preservation of phospholipids (Logemann et al., 2011), we exclude the possibility that the
presence of PUFAs represents a fossil signal exported from the surface waters but rather propose a deep-
sea bacterial source (DeLong and Yayanos,1986, Fang et al. 2003; Valentine and Valentine, 2004), as





noted above.
Head group variations of GDGTs in the core OMZ/deep oxycline depth region were similar to their
distributions in the upper OMZ, with a notable increase of HPH-GDGTs with depth at stations 2 and 5.
Similar to the upper OMZ, archaeal sources for the detected IPL-GDGTs could be either euryarchaeota,
crenarchaeota or thaumarchaeota (Lincoln et al., 2014; Elling et al., 2017) as these phyla were detected
within the core OMZ and deep oxycline of the ETNP (Podlaska et al., 2012). Relative archaeal IPL
abundances within the core OMZ and deep oxycline vary between stations, but were highest at station 8
where they reach over 50% of the total microbial IPLs.    Elevated abundances of archaea had been
enumerated by quantification via CARD-FISH (Podlaska et al., 2012), with highest abundances (25 to 50%
of total DAPI-stained cells) at stations 2 and 5. It should be noted again, however, that >50% of the
microbial populations that were DAPI-positive cells remained uncharacterized by the CARD-FISH
approach used (Suppl. Fig. 2).

*4.2 Factors influencing IPL distribution*
Relative abundances of IPLs in the oxic zone of the ETNP were distinct from IPL distributions in
surface waters of other oceanic ocean regions where SQ-DAG and PC-DAG were typically the most
abundant compounds within the glycolipids and phospholipids, respectively (Van Mooy and Fredricks,
2010; Popendorf et al., 2011a,b).    Whereas SQ-DAG was among the most abundant IPL in the surface
waters of the ETNP (18-50%), PC-DAG was comparably minor (3-13%). This difference might result
from the highly compressed mixed layer of the ETNP compared to other locales, with consequent
differences in plankton ecology.    Alternatively, there could be differences in physiologic adaptations and



hence membrane lipid modifications between the ETNP and other regions, (e.g., Van Mooy et al., 2009).
In general, our study confirms the dominance of glycolipids as a common feature of surface ocean waters
in which IPL distributions have been determined, in particular the Black Sea (Schubotz et al., 2009), the
Eastern Subtropical South Pacific (Van Mooy and Fredricks, 2010), the Western North Atlantic
(Popendorf et al., 2011a), the Mediterranean Sea (Popendorf et al., 2011b) and the Cariaco Basin
(Wakeham et al., 2012). In addition, the present study highlights the potential importance of glycolipids
and other non-phosphorus lipids for bacteria in low oxygen environments as will be discussed in detail
below.

*4.2.1 Influence of environmental parameters on IPL distribution*
The NMDS analyses and Spearman Rank Order Correlations provide a better understanding of the
influence of environmental factors and the microbial community structure on the IPL composition in the
water column of the ETNP. NMDS analysis of normalized IPL composition and quantitative microbial
data (abundance of α, β, γ, ε-proteobacteria, sulfate-reducing bacteria δ-proteobacteria, planctomycetes,
crenarchaeota including thaumarchaote and euryachaeota) did not yield any high goodness of fit statistic
($r^2 < 0.3$; Suppl. Table 4). There are several potential reasons for this, the most likely being that IPL also
derive from eukaryotes in the oxic zone and secondly because many of the proteobacteria in fact also
biosynthesize similar IPL assemblages. Changes in bacterial community structure might not necessarily
result in significant variations in IPL composition. Rather than focusing solely on IPL source, we were
interested in deciphering whether environmental factors such as temperature, nutrient or oxygen
concentrations might affect IPL composition and thus compare any environmental impact with what has



been observed in culture studies and other natural settings.  Many of the major and minor glycolipids
were loaded negatively on the NMDS2 axis, as were oxygen, fluorescence, Chl-$\alpha$, POC and TN, with the
notable exception of 1G-DAG which had only a slightly negative loading on the NMDS-2 axis.  These
relationships (loadings) roughly reflect the vertical distribution of IPLs in the water column of the ETNP.
Glycolipids, particularly 2G-DAG and SQ-DAG, were most abundant in the oxic zone characterized by
high oxygen concentration and moderate primary productivity (high POC, TN and elevated Chl-$\alpha$ and
fluorescence).  Spearman Rank Order Correlations confirm these observations, including the lack of
significant correlations between 1G-DAG and depth or any other environmental parameter.  One
explanation for this is that 1G-DAG has diverse sources throughout the water column independent of any
environmental variable.  Phospholipids, and in particular PE-, PME- and PDME-DAG, become more
prevalent within the core OMZ where eukaryotic sources were minimal and non-photosynthetic bacteria
dominated the microbial communities and at deeper depths where nutrient concentrations ($NO_3^-$ and $PO_4^{3-}$)
were elevated due to organic matter remineralization, giving positive loadings of these environmental
parameters on the NDMS2 axis.  Notably only the minor phospholipids, DPG, PME-, and PDME-DAG,
showed a similar positive loading on the NDMS2 axis as did depth, $NO_3^-$ and $PO_4^{3-}$.  Most major
phospholipids and aminolipids did not correlate with any of the tested environmental variables, due to
their presence in equal relative abundances within all the different biogeochemical zones.  This is also
reflected in the lack of Spearman Rank Order Correlation for most of the major lipids with environmental
parameters (Table 1).  In contrast, most archaeal IPLs showed a positive loading on the NMDS2 axis,
consistent with the increasing importance of archaeal abundance with depth and at reduced oxygen
concentrations.






### 4.2.2 Links between substitute lipid ratios and nutrient concentrations

Because they have similar biochemical functions and the same ionic charge at physiological pH, SQ-DAG and DGTS are known as substitute lipids for PG-DAG and PC-DAG, respectively, when phosphorus is limiting (Benning et al., 1993; Van Mooy et al., 2009; Popendorf et al., 2011b). The elevated ratios of SQ-DAG:PG-DAG and DGTS:PC-DAG in the surface waters of phosphorus-limited Sargasso Sea (4 to 13) compared to phosphorus-replete South Pacific (3) suggests that phytoplankton synthesize phosphorus-free substitute lipids to maintain growth in response to phosphorus starvation (Van Mooy et al., 2009). Underlining this observation, relative abundance of phospholipids was positively correlated with phosphate concentration across the Mediterranean Sea (Popendorf et al., 2011b). Microcosm incubations of seawater from the Mediterranean Sea supplemented with phosphate and ammonium confirmed that changes in substitute lipid ratios were partly caused by a physiological response to nutrients (Popendorf et al., 2011b). However, neither of these substitute lipid ratios was significantly correlated with abundance of phosphate in surface waters of the Eastern Subtropical South Pacific (Van Mooy and Fredericks, 2010), leading the authors to conclude that not only phosphate limitation but also algal community structure may impact these ratios.

To further explore the possibility that SQ-DAG and aminolipids (DGTS) in the OMZ of the ETNP might serve as substitute lipids, we performed a Spearman Rank Order Correlation of known substitute lipid ratios as well as total aminolipid (AL) to phospholipid (PL) and total glycolipid (GL) to PL ratios with nutrient concentrations and other environmental parameters. Only SQ-DAG:PG-DAG was significantly correlated with phosphate (-0.56, $p < 0.001$) but also correlated with other parameters, such



as depth (-0.76, p<0.001) and oxygen concentration (0.58, p<0.001).   These correlations reflect the
elevated SQ-DAG:PG-DAG ratios (>2) in the surface waters and upper OMZ (Fig. 4) and support the
notion that SQ-DAG functions as a substitute lipid in the ETNP where phosphate concentrations are <2
µM.   The other proposed substitute lipid ratios, DGTS:PC-DAG (Van Mooy et al., 2009) and
1G-DAG:PE-DAG (Carini et al., 2015), did not correlate with nutrient concentrations in the water column
of the ETNP but rather showed highly variable distributions.   Similarly, AL:PL ratios did not exhibit
strong relationships with any environmental parameter, and GL:PL ratios showed similar but less
pronounced trends as SQ-DAG:PG-DAG ratios.   Similar to the suggestion of Van Mooy and Fredericks
(2010) for the Eastern Subtropical South Pacific, the lack of a correlation of these substitute lipid ratios
with phosphorus in the ETNP might be due to changes in community composition and not to phosphorus
limitation, since phosphate concentrations increase within the core OMZ and the deep oxycline.
However, this interpretation stands in contrast to what is currently known from cultured representatives,
that replace their phospholipid content with glycolipids at phosphate concentrations <20 µM (e.g., Bosak
et al., 2016).   Accordingly, many of the organisms living within the OMZ may already be phosphorus
limited at micromolar concentrations of phosphate.

*4.2.3 Factors affecting structural diversity of the core lipid composition*

We observed a considerable diversity in both headgroup and core lipid types, from diacylglycerol

lipids with varying chain lengths and zero to multiple unsaturations, with or without hydroxylations to
mixed ether/ester glycerolipids, sphingolipids and ornithine lipids.   Changes in core lipid chain length or
unsaturations are often associated with temperature.   However, NMDS analysis did not yield any strong



correlations with temperature and chain length or number of double bonds of the major IPL classes ($r^2 <$
0.02, Suppl. Table 4), or with other environmental parameters (r2 < 0.3, Suppl. Table 4).   Instead, we
conclude that observed changes in chain length and unsaturations are most likely due to changing
biological sources.   For instance, long chain PUFAs in surface waters are mainly synthesized by
phytoplankton, while in deeper waters myriad bacterial sources are probable.   Likewise, hydroxylations
in the acyl side chains did not show any clear link to specific environmental factors, although, both
1G-OH-Cer and OH-DGTS showed negative loadings on the NMDS-2 axis indicating a higher abundance
of these compounds in oxic samples.   This is in accordance with the previous assumption that these lipids
play a role during oxidative stress and/or are involved in other defense mechanisms.   The occurrence of
mixed ether-acyl lipids in ocean waters has been reported previously in a wide number of oceanic settings
(Hernandez-Sanchez et al., 2014) where aerobic bacteria were suggested as source organisms.   In our
study we detected PE- and PC-AEG at all depths in the ETNP but with no noticeable correlation with
depth or oxygen concentrations (Fig. 7) and thus we suggest that various bacteria living in both the
oxygenated surface waters and in the suboxic OMZ are potential sources for these compounds in the water
column.
Ornithine lipids and sphingolipids play many functional roles in biological systems, confounding
identification of potential sources.   Ornithine lipids were strongly negatively loaded on the NMDS-1 axis,
but none of the measured environmental parameters could account for this negative loading (Fig. 7).
Therefore, it remains unclear what factor(s) ultimately determine their distribution.   Likewise, the
absence of significant correlations between the sphingolipid 1G-Cer, and any environmental parameter
lead us to conclude that the abundance of 1G-Cer reflects the diverse microbes inhabiting the changing



oxygen regime within the water column rather than any specific source organism.

**5. Conclusions**

Diverse intact polar lipids, including four classes of diacylglycerol glycolipids (with monoglycosyl,
diglycosyl, triglycosyl and sulfoquinovosyl head groups), seven diacylglycerol phospholipids (with
phosphatidyl glycerol, phosphatidyl ethanolamine, phosphatidyl choline, phosphatidyl (*N*)-
methylethanolamine, phosphatidyl (*N,N*)-dimethylethanolamine, diphosphatidyl glycerol and
phosphatidyl inositol head groups) and three diacylglycerol aminolipids (with homoserine and two
unidentified head groups) are present in the water column of the ETNP.    Mixed ester-ether glycerol lipids
with phosphatidyl ethanolamine, phosphatidyl choline and sulfoquinovosyl head groups as well as
glycosidic ceramides and ornithine lipids were detected throughout the water column.    A wide range of
archaeal GDGTs were most abundant within the OMZ.    This diversity in IPL compositions reflects the
dynamic nature of the biological community that inhabits the range of environments in the ETNP, with
oxygen as a primary determinant, from fully oxygenated surface waters to a strong oxygen minimum zone
at depth.    Highest concentrations of IPLs (250 – 1500 ng/L) in oxygenated surface waters zone reflect
the dominance of phototrophic eukaryotic and cyanobacterial sources above the OMZ, but secondary
peaks in IPL concentration (12 – 56 ng/L) within the core of the OMZ result from elevated abundances of
heterotrophic and chemoautotrophic bacteria and archaea under low oxygen conditions.    Glycolipids
derived from photoautotrophs generally accounted for more than 50% of total IPLs in the euphotic zone
(< 200 m, oxic and upper OMZ zones), while bacterial phospholipids were more predominant (avg. 40%)
in the OMZ and deep oxycline layers.    Depth-related variations in the dominant fatty acid compositions



for each IPL class show that specific biological source(s) for each IPL were distinct in each depth/oxygen-
content horizon.  Nevertheless, microbial sources for many of the detected lipids remain unclear and
therefore potentially unique ecophysiological adaptations these lipids may represent remain to be explored.
The presence of the glycolipid, monoglycosyl diacylglycerol (1G-DAG), and the betaine lipid,
diacylglyceryl homoserine (DGTS), with varying fatty acid compositions, within all zones indicates that
these canonical phototrophic markers may indeed be synthesized in different parts of the water column by
a much larger host of organisms (including non-phototrophs) than previously thought.  Since 1G-DAG
and DGTS are known to be biosynthesized by a variety of bacteria only under phosphorus limitation, we
suggest that they might serve as substitute lipids for the microorganisms in the OMZ.  Since lipid
substitutions have been observed in bacterial cultures at phosphate concentrations < 20 μM, conditions
that are met in the OMZ of the ETNP and other oceanic systems that are generally not considered to be
phosphorus limited, perhaps the paradigm of substitute lipids needs to be re-evaluated.

**Author contribution**
SGW collected the samples. SGW, FS and KUH designed the study. SX and FS measured and processed
the data. JSL and FS performed statistical analyses. FS, SX and SGW wrote the paper with input from
KUH and JSL.

**Competing interests**
The authors declare that they have no conflict of interest.



**Acknowledgments**
We are grateful to the captain and the crew of R/V *Seward Johnson*, to K. Daly and K. Wishner as co-
chief scientists, and to the U.S. National Science Foundation for supporting the cruise. H. Albrecht, B.
Olsen and S. Habtes helped with PM sampling. We thank K. Fanning and R. Masserini (University of
South Florida) for providing their nutrient results; C. Flagg (Stony Brook) processed CTD hydrographic
data; Jay Brandes and Mary Richards (Skidaway Institute) conducted the POC and TN analyses; B. Olson
and K. Daly (University of South Florida) provided ship-board Chl-*a* analyses; and G. DeTullio (College
of Charleston) conducted HPLC analyses of pigments.    Lab supplies and analytical infrastructure for
lipid analyses was funded by the Deutsche Forschungsgemeinschaft (DFG, Germany) through the Cluster
of Excellence/Research Center MARUM. The UPLC-QTOF instrument was granted by the DFG,
Germany through Grants Inst 144/300-1. S. Xie was funded by the China Scholarship Council, F. Schubotz
by the Zentrale Forschungsförderung of the University of Bremen, and U.S. National Science Foundation
grant OCE-0550654 to S. G. Wakeham supported this project. SGW also acknowledges a Fellowship from
the Hanse-Wissenschaftskolleg (Hanse Institute for Advanced Studies) in Delmenhorst, Germany.

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





**Tables**

**Table 1.** Spearman Rank Order Correlation coefficients (r) for data combined from all four stations. Only

significant correlations, where 0.05 (highly significant p < 0.001, in bold), are presented.

| | Glycolipids | | | | | | Aminolipids | | | | Phospholipids | | | | | |
|---|---|---|---|---|---|---|---|---|---|---|---|---|---|---|---|---|
| | GL % | 1G % | 2G % | SQ % | GL:PL | SQ:PG | AL % | DGTS % | AL:PL | DGTS:PC | PL % | PC % | PG % | PE % | PME % | PDME % |
| Depth | - | 0.32 | **-0.7** | - | -0.41 | **-0.76** | | | | | | | | - | -0.46 | -0.52 |
| Fluorescence | | | **0.63** | **0.67** | | **0.65** | | | | | | | | | | |
| POC | | | **0.61** | **0.6** | | **0.6** | | | | | | | | | | |
| TN | | | **0.66** | **0.62** | | **0.63** | | | | | | | | | | |
| Oxygen | **0.57** | 0.3 | **0.48** | 0.35 | **0.55** | **0.58** | | | 0.36 | | **0.49** | - | 0.38 | 0.33 | | |
| Temperature | 0.3 | | **0.52** | **0.63** | 0.39 | **0.69** | | | | | | | | | | |
| Chl a | 0.35 | | **0.72** | **0.71** | 0.42 | 0.78 | | | | | | | | | | -0.33 |
| Phosphate | | | **0.62** | **0.53** | -0.4 | **-0.56** | | | | | | | | | | 0.36 |
| Nitrate | | | - | - | | -0.38 | | | | | | | | | | |
| Nitrite | - | 0.33 | **0.53** | **0.49** | | | | | | | | | | | 0.3 | |
| Ammonium | | | | | | | 0.41 | 0.42 | 0.35 | 0.4 | | | | | | |
| N:P | | | -0.3 | 0.32 | | | - | | | | | | 0.36 | | | |

Abbreviations: GL – glycolipids, 1G – monoglycosyl, 2G – diglycosyl, SQ – sulfoquinovosyl, PL – phospholipids, AL – aminolipids, DGTS – diacylglyceryl trimethyl homoserine, PC – phosphatidyl choline, PG – phosphatidyl glycerol, PE – phosphatidyl ethanolamine, PME – phosphatidyl methyl-ethanolamine, PDME – phosphatidyl dimethyl-ethanolamine



**Figures**

**Figure 1.** a) Map of ETNP with R/V *Seward Johnson* (November 2007) cruise sampling stations. Depth profiles of (b) oxygen, (c) temperature, (d) chlorophyll-$\alpha$ and (e) transmissivity along a northwest-southeast transect of the study area. Numbers across the top panels denote station, black dots are individual samples.

**Figure 2.** Section plots of major macronutrients along a northwest-southeast transect of the ENTP up to 1300 m water depth: (a) nitrate, (b) nitrite, (c) ammonium, (d) phosphate, (e) N:P (dissolved) is the sum of total dissolved nitrogen species (nitrate, nitrite and ammonium) over phosphate, and (f) C:N (SPM) is total carbon over total nitrogen of the solid phase collected by water filtration. Note that C:N is only analyzed for <53 μm particle fraction. Numbers across the top panels denote station, black dots are individual samples.

**Figure 3.** Absolute abundance of (a) particulate organic matter (POC) and (b) intact polar lipids (IPL) as well as (c) the ratio of their concentration are shown for stations 1, 2, 5 and 8. Note that POC and IPL/POC are only analyzed for the <53 μm particle fraction. Also shown are (d) absolute cell abundance and relative proportions of (e) archaeal cells and (d) unclassified cells at the same stations. Numbers across the top panels denote station, black dots are individual samples.

**Figure 4.** Relative abundance of (a) major and (b) minor IPLs at sampled depths of stations 1, 2, 5, and 8 in the ETNP. Major IPLs are defined as those comprising more than 10% of total IPLs (minor compounds



comprised less than 10%) at more than one depth horizon at the four stations. Also depicted are the
different geochemical zones in the water column.

**Figure 5.** Relative abundance of IPLs along a northwest-southeast transect from station 1 to 8 grouped by
headgroup: total non-archaeal (a) phospholipids, (b) aminolipids, and (c) glycolipids are shown as percent
of total IPLs. The ratios of non-phospholipids to phospholipids are shown for (a) SQ-DAG to PG-DAG,
(e) DGTS to PC-DAG, and (f) 1G-DAG to PE-DAG. Numbers across the top panels denote station, black
dots are individual samples.

**Figure 6.** Changes in average chain length (CL) and number of double bonds (DB) of major IPLs detected
at stations 1, 2, 5 and 8 in the ETNP.

**Figure 7.** Nonmetric multidimensional scaling (NMDS) ordination plot assessing the relationship between
IPL biomarkers, sampling depths and geochemical parameters in the ETNP (stress=0.125). Squares
represent the water depth of each sample and are color-coded according to the defined geochemical
zonation. Filled circles stand for lipid distribution of major IPLs and open circles for minor IPLs on the
ordination. Vector lines of geochemical parameters are weighted by their p-values with each NMDS axis.





fig01







fig02

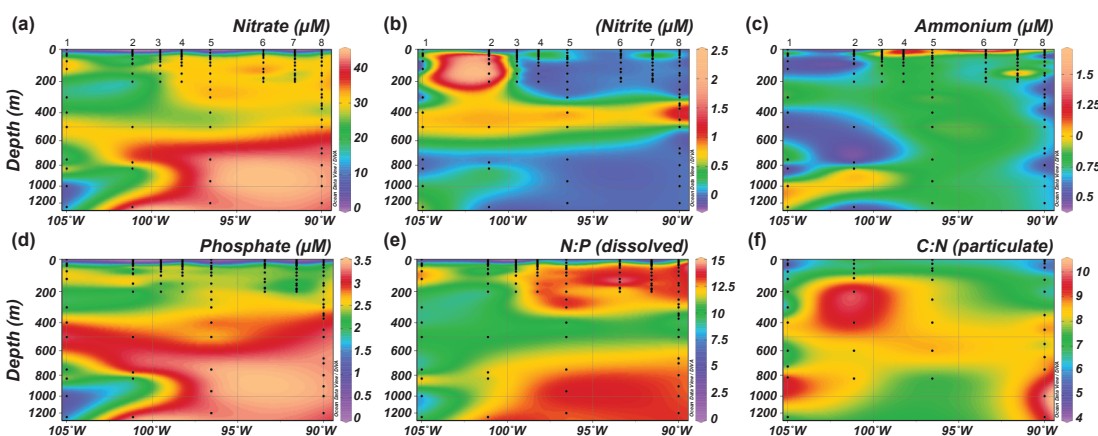





fig03

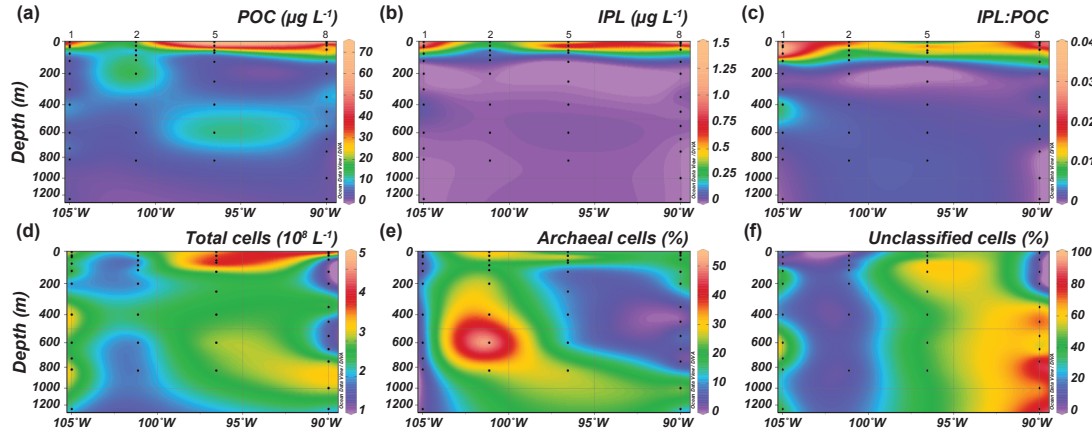





fig04





fig05

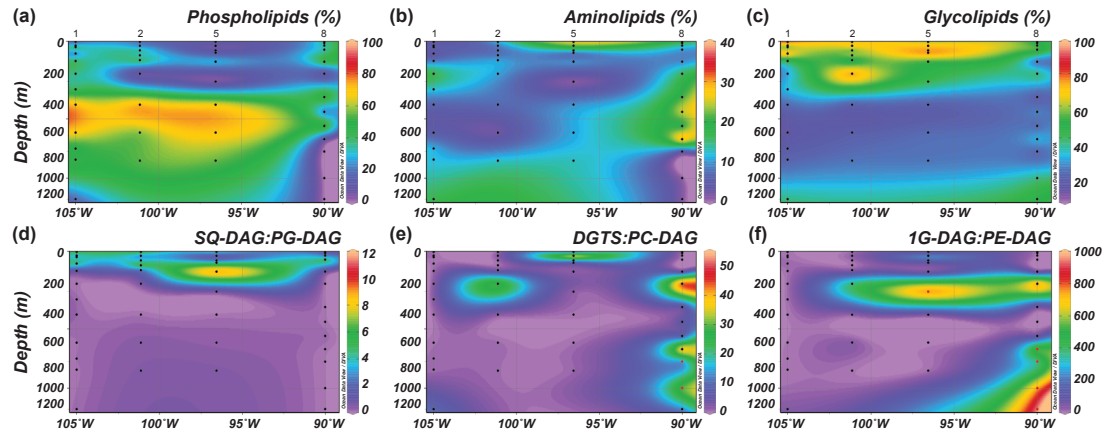



fig06

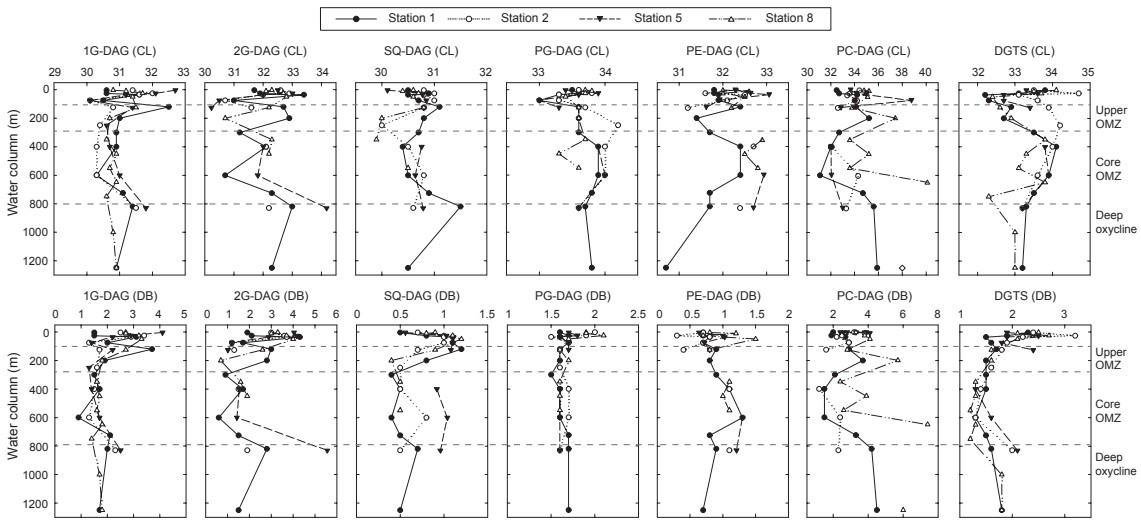

1234



fig07

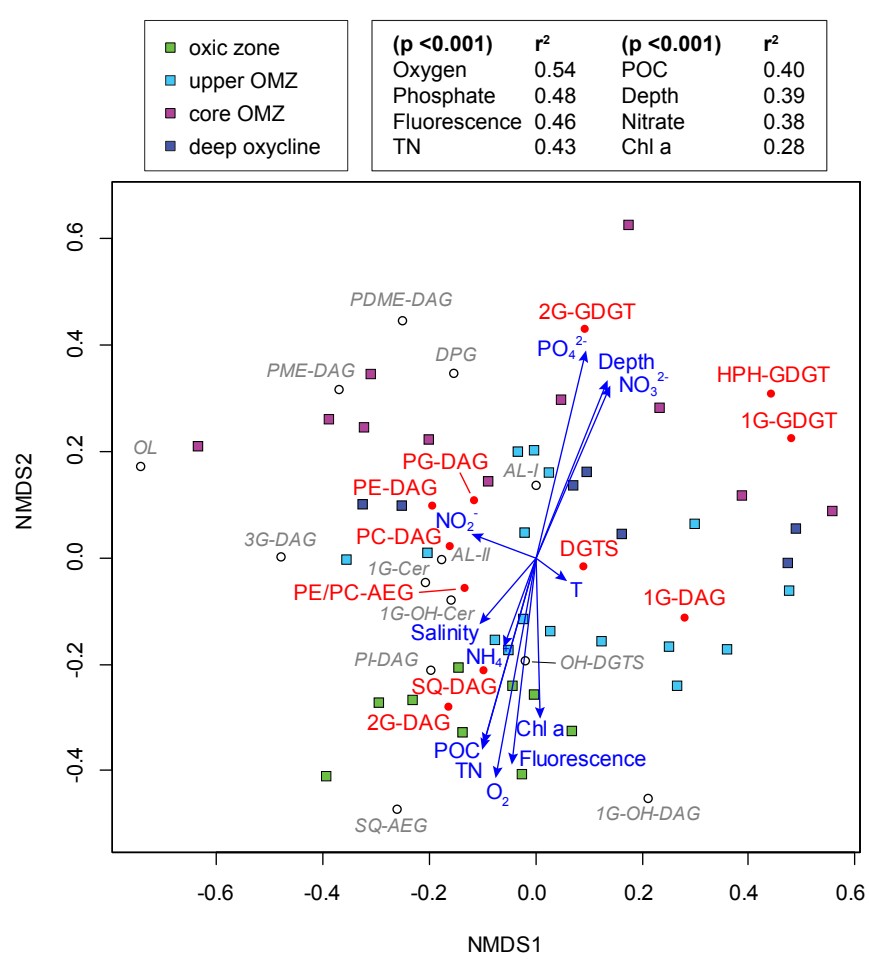