# Peer review of "Diversity of intact polar lipids in the oxygen minimum zone of the Eastern Tropical North Pacific: Biogeochemical implications of non-phosphorus lipids"

_Biogeosciences, 2018_

## Referee Comment (RC1) · Anonymous Referee #1 · 26 Mar 2018

This study aims to determine the intact lipids in suspended particles in the water column using samples collected from ocean oxygen minimum zones from the east/north pacific. OMZs are important marine ecosystems particularly with regard to oceanic N cycles. The comprehensive data presented in this study has significantly advanced our understanding of IPLs in this unique environment. This reviewer has no major concerns. Some specific comments are listed below for further improvement of the manuscript.

L93, it would be very useful to specifically discuss previous IPL studies on OMZ sam-

[Figure]

ples, e.g ETNP, ETSP, Arabian Sea etc in discussions.

L101 and throughout the MS, most previous studies have used MGDG, DGDG and SQDG to refer to mono- and di glycosyl- DAG and sulfoquinovosyl DAG, please change to these commonly used acronyms for the sake of consistency in literature.

L104-5 , l307-9; this may be a little misleading since DGTS has been found in a wide range of marine heterotrophic bacteria.

L116, please refer to recent study of Hunter et al., AEM doi: 10.1128/AEM.02034-17 for novel diglycosylceramides found in Thalassiosira.

L214-215, the authors have referred to previous studies for mass spectral interpretation and IPL assignments. It would be very useful to summarize and synthesise these information in a table (or in the supplementary information) and to detail the criterial for IPL identification. Presumably IPL assignment is based on comparing to retention time of standards (where applicable) and characteristic MS/MS patterns, representative characteristic ions or characteristic neutral loss. How has DGTS but not DGTA been conclusively assigned in this study? Has DGTA been found in any samples?

L223, for unknown aminolipids AL1 AL2, do the authors have any hypothesis of their structures based on MSn fragmentation patterns (suppl fig 4)? What are the possible amino head group structures? Have accurate ms of AL1, AL2 been obtained?

The authors mentioned CSRD FISH data in supplementary dataset but did not mention how this was done in the materials and methods.

My general impression for discussion is that it can be shortened significantly.

It is a pity that no microbial diversity data were obtained in this study as one would like to see the correlation between specific microbial groups and IPLs, which may provide clues for the origin of these lipids, particularly w.r.t. to AL1 and AL2.

Section 4.1.1 two recent papers (Carini et al, Sebastian et al) have shown marine

heterotrophic bacteria are also abundant in MGDG. These need to be discussed here in line with these new evidence.

---

## Referee Comment (RC2) · N.J. Bale (Referee) · 30 Mar 2018

General comments

This study examines the intact polar lipid (IPL) distribution in suspended particulate matter (SPM) from four stations in the oxygen minimum zone (OMZ) of the Eastern Tropical North Pacific (ETNP). It aims to link the IPL distribution of different water column zones with the microorganisms found there and to examine the ecophysical adaptions to the different zones of the OMZ. This is an extensive data IPL set and authors have theorized which groups of microorganisms are responsible for which IPLs. The strength, as I see it, in this study is the examination of how the IPL distribution changes across the different zones of the water column due to the changes in the biogeochemical environment. Indeed, due to the generic nature of many of these IPLs it is only possible to put forward tentative assignments of their sources, whereas examining changes in the lipid composition with changing environmental parameters provides more solid information. Overall, I recommend this article for publication with the following edits and with suitable responses to my questions on the analytical methods.

Specific comments

I have two specific comments.

My first specific comment relates to the extraction and analysis of these samples. The manuscript states that the samples were collected in 2007 and (presumably soon after) that they were extracted using Soxhlet-extraction with DCM:MeOH for 8 hours. From an IPL perspective this seems a "harsh" extraction method that has potential to destroy certain IPLs, resulting in a IPL distribution unrepresentative of that in nature. Could the authors comment on whether this would be their preferred method of extraction for IPLs or whether IPL analysis was not the original reason for the chosen extraction method? Indeed the first author has described utilizing the much gentler modified 'Bligh and Dyer' extraction method in other publications relating to IPLs. The authors also describe two different analytical methods used to quantify and to identify structures. Can you state whether analysis occurred soon after extraction in both cases? If not, how were the extracts stored and for how long? If the two analyses were carried out at different times, did extract storage introduce changes in the lipid distribution? I noted that the reference for the second LC-MS method applied (Wörmer et al., 2013), was published 6 years after the samples were collected.

Based on the authors' replies, suitable discussion of these issues should be included in the method section.

My second specific comment relates to the length and depth of the discussion. This is a subject I find very interesting and yet I felt rather weighed down in information at certain points. I feel it would aid the reader if this was shortened and made more succinct.

Technical corrections

Line 52 - change to "the subsequent"

Line 70 - change to "(ENTP), situated off the..."

Line 93 - change to "(IPL) are the main building blocks of cellular membrane and may"

Line 98 - change to "the North Sea"

Line 100 - you could also include in your reference list here the Western English Channel (White et al., 2015).

"The combined effects of seasonal community succession and adaptive algal physiology on lipid profiles of coastal phytoplankton in the Western English Channel. D.A.White , C.E.Widdicombe, P.J. Somerfield, R.L. Airs, G.A. Tarran, J.L. Maud, A. Atkinson. Marine Chemistry 177 (2015) 638–652)."

Line 129 - Remove 'here' to read "Notably, replacing..."

Line 141 - change to "extension of that of Xie"

Line 142 - change to "at two stations described here (station 1 and 8)"

Line 159 - Should this be "VERTEX I and II" ?

Line 160 The Martin et al. (1987) reference is missing from your reference list.

Line 177 - should define GFF at first use.

Line 223 - insert "response could not be corrected for"

Line 350 - The term amino lipid and betaine lipid seem to be used interchangeably

throughout the manuscript. Could this be defined at one point?

Lines 434-439 - This introductory sentence is too long to read and needs to be broken up or shortened.

Line 436 - replenishment that produces

Line 454 - "Podlaska et al. (2012)"

Line 482 - "by Xie et al. (2014)". I have noticed this citation format error in more places. Please change throughout.

Line 489 - "coinciding with high Chl-a concentrations, reflecting"

Line 489 - What do you mean eukaryotic rather than microbial? Eukaryotes can be microbes and microbes can be eukaryotic. Do you mean eukaryotic rather than prokaryotic? But also in your results section, 3.1, you state that Prochlorococcus (not eukaryotes) were an important component of the photoautotrophic community. Hence I think the correct thing would be to say "photoautotrophic rather than heterotrophic". Is this not what you wanted to say here? Check you have this correct throughout the manuscript.

Line 497 - pluralize "IPLs"

Lines 540 - 551 - This section is confusing because you contradict each statement. You state that Eukaryotic phytoplankton and cyanos are assumed to be a major source of PG-DAG. Yet you then state that heterotrophic bacteria can also be a dominant source. Maybe using conjoining words "however" and "although" would make this section flow nicely.

Line 547 - change to "we therefore suggest that"

Line 547 - remove heterotrophic bacteria. Cyanobacteria are not heterotrophic bacteria.

Line 576 - insert "abundance of less"

Line 586 - change to "fatty acid" or "acyl" rather than "fatty acyl"

Line 601 - change to "are <20"

Line 637-638 - remove tab within word "thaumarchaeota"

Line 648 - insert commas "zone, and that….1993), became"

Line 667 - replace "microbial" with "bacterial"

Line 679 - remove "shallower" as it is redundant.

Lines 694 - 670 - Please make this long sentence shorter or break it into two. You repeat the same words "exported, fossil and signal" twice.

Line 767 - insert "waters of the phosphorus-limited"

Line 768 - insert "to the phosphorus-replete"

Line 770 - insert "observation, the relative abundance"

Line 808 - change to "a myriad of bacterial sources"

Table 1 - Make the columns wider so that the cell contents all lie on one line.

Table 1 caption - should this read "where $p < 0.05$"

Figures 1,2,3 and 5 - Can you indicate the four water column zones on these figures. Perhaps with lines that join the specific depths at which the regions are defined (as was done in figures 4 and 6).

Figure 2 - unnecessary brackets around nitrite in panel b

---

## Referee Comment (RC3) · Anonymous Referee #3 · 23 Apr 2018

This paper provides a very detailed account of the IPL composition in the waters of the oxygen-deficient zone (ODZ) of the Eastern Tropical North Pacific. It adds to the growing inventory of IPL data. The authors claim that it contributes to our understanding of these systems. I am not entirely convinced. There are also a substantial number of issues related to the analytical methodology that need to be resolved before this paper can be published.

The general aim of the paper is "to evaluate the microbial ecology and ecophysiological adaptations that affect organisms inhabiting the OMZ in the context of biogeochemical

cycles" (line 30-32). However, when you read the remaining part of the abstract this does not materializes. It ends with a rather vague statement about potential phosphorous limitation, which in the light of the analytical constraints (see below) may be even weaker. I strongly suggest to remove these kinds of claims from the text and just focus on what the paper is about: an inventory of IPLs in the water column of this region.

The authors do report absolute IPL concentrations (Fig. 3b) which show an order of magnitude decrease over the first 100 m of the water column. This is in line what would be expected since this is the zone where primary production is taking place (as is also revealed by the pigment concentrations) and the data would allow to discriminate IPLs produced by phytoplankton in the photic zone from IPLs (produced in much lower concentration) by prokaryotes residing below the photic zone and within the ODZ. However, the focus in the paper is too much on relative abundances of IPLs (e.g. Fig. 5) for unknown reasons. The discussion should be more focused on the zone where IPLs are primarily produced (i.e. the photic zone) vs. the remainder of the water column that may be influenced by IPLs in descending particles (i.e. produced in the upper water column) and additional production by prokaryotes. In this discussion, it should be taken into account that PUFA-containing IPLs may degrade faster than other IPLs. Now, the total inventory of IPLs is too much discussed in terms of relative abundance in relationship with nutrient profiles and other environmental parameters over the whole water column, which is too simplistic.

The paper also suffers from too much detail. It is very hard to follow because every tiny IPL detected is described without a clear environmental implication. The authors should formulate specific research questions (i.e. not understanding the "functioning of OMZs throughout the world ocean" by studying IPLs) and address these. Not every minor IPL detected has to be described!

There are also a number of issue related to the analytical methodology of analyzing the IPLs. Adequate answers should be provided on all issues since this may seriously affect the interpretation of the data.

1) Filtration. The authors used 0.7 micrometer glass fiber filters for filtration. The limitations of the use of this filter size has extensively been discussed elsewhere and the authors acknowledge the limitations of their approach by admitting that they might be missing smaller cells. However, they should also mention that the pore size will decrease during the filtration process and thus will recover a (variable) part of this material. More importantly, they should stress that this does not only lead to "minimal values" for IPL concentrations but that it may also affect the distribution of IPLs that they report (as percentages). Furthermore, they used a prefiltration device to "remove most eukaryotes" (line 169). It is hard to believe that this will remove most of the algae; if so this would also strongly affect their interpretations.

2) Extraction. The IPLs have been Soxhlet extracted (i.e. boiling DCM/methanol for 8 h). Although this is a common method for extraction of less labile lipids, for IPLs it is rather unusual since these are very labile and the commonly applied method for this type of work is Bligh Dyer extraction at room temperature and controlled pH conditions. The authors should present data to show that their extraction method does not alter the IPL distribution (i.e. their main target of study) due to the fact that some IPLs are more labile than others (e.g. in the ratio of phospho vs. non-phospho IPLs, which plays an important role in the discussion).

3) Analysis. The experimental description indicates that the IPLs have initially been analyzed by HPLC-ESI-IT-MSn using the same system as described by Schubotz et al. (2009). In the meanwhile, this group has developed improved methods of analysis of IPLs (e.g. Wormer et al., 2013) and the question arises why these "old" results are still used since the samples were also re-analyzed using these new methods. It does not become clear how IPLs can be quantified with one method and identified by an-other method (lines 206-207) using an entirely different separation system. One issue that should also be addressed is the timeline of all these analyses. Once extracted, IPLs cannot be stored for a substantial time without significant alteration/degradation. Schubotz et al. (2009) in their very much related work of IPLs in the Black Sea stated

"Three years after primary qualitative analysis the samples were spiked with C16-PAF (1-O-hexadecyl-2-acetoyl-sn-glycero-3-phosphocholine) as injection standard and re-run for quantification. Slight changes in the relative distribution of IPLs were observed within the two runs. In particular the differences were identified as a selective loss of the glycolipids Gly-DAG, Gly-Cer, Gly-GDGT and 2Gly-GDGT (data not shown). We interpret this loss as a sign of selective degradation of glycolipids during storage." So, an important question is how much time evolved between these two analyses and can the results still be compared?

4) Quantitation. It does not become clear from the experimental description if the commercially available standards were run with the HPLC-ESI-IT-MSn system that was used for quantitation. If so, the results (response factors) should be reported. If not, there is a serious problem since response factors should be determined on the same system. The whole procedure of quantitation should be made clear. An "injection standard" is mentioned but it remains unclear what was done exactly. Why didn't the authors use an internal standard that was added to the extract? Such a standard would at least have been exposed to the same conditions as the IPLs of the samples (e.g. during storage). The authors should also report the analytical errors of their determinations. The data they now report (e.g. Table S2) are extremely accurate (e.g. a relative abundance of 16.68 % SQ-DAG in station 1 at 35 m). I would expect that the analytical error is 10-20%, so the reported data are far too accurate unless the analytical error is extremely low. This also holds for many of the other data: the reported accuracy of absolute pigment concentrations is also (far) too high and so is the data reported in Table S3 (if the SD is larger than the figure itself is odd to report three or four significant numbers).

Additional comments:

Line 40: It is useless to compare relative trends in IPLs with absolute trends of environmental parameters. To this end, absolute concentrations (like you have for pigments) need to be used.

Lines 68-69: ......but not provided in this paper, so remove this sentence.

Lines 117-119: strange sentence

Lines 119-124: I think this overview should be limited to papers describing intact IPLs in the water column. For example, the data presented by Lincoln et al., 2014 are not really solid IPL data. Turich and Freeman, 2001 and Hurley et al., 2016 present only core lipid data.

Line 125: It is not discussed how IPLs can be used as taxonomic indicators. This seems pretty relevant information for this type of study

Line 139: in what way is this approach "complementary"?

Line 146-147: here the authors promise that we should learn a lot (deeper insight into biogeochemical cycling, functioning of OMZs throughout the world ocean) but this is grossly overstated.

Lines 152-152: data on coordinates of sampling stations is incomplete.

Line 166: provide details on sampling volume

Line 186: provide details on pore size of filter

Line 200: storage at -20 degrees C is not sufficient to prevent alteration; IPL extracts should be stored at -80 degrees C and even then, distributions may change. How long were the extracts stored before analysis for IPLs?

Line 210: different column than in other analysis. Why?

Lines 217-219: provide more details on these standards. What are the acyl moieties of these standards? How are the response factors affected by unsaturations in the acyl moieties? What was the time between the arrival of these standards in the lab and their measurement? How were they stored? It is known that these standards degrade over time upon storage. How often were these standards run? Before each batch of

analyses? How did the response factors vary over time? Answers to these questions are essential for getting a feel for the confidence we can have in the reported relative abundances of the IPLs.

Line 224-224: why would the unknown response factors be in the range of the measured standards? This is not a scientifically acceptable statement in this way. Just say that they are unknown and what you have assumed to be able to calculate a concentration.

Lines 258-262: provide references to indicate that pigments can be used to reveal this information even at the species level (e.g. Rhizosolenia).

Line 267: secondary maxima are not revealed in Fig. 1e.

Lines 268-269: Not evident from Figs 2a-b.

Lines 273-276: So, the whole system is NOT P-limited!

Line 282: How were absolute concentrations obtained?

Line 284: Secondary maxima are not observable in Fig. 3d.

Line 289-291: It would be logical to introduce first all IPL classes observed. Why are absolute concentrations of IPL classes not described and is the manuscript concentrated on relative abundances?

Line 292: "select substitute lipid ratios" is not introduced. It should be introduced in the discussion not in the result since it is an interpretation of the data presented.

Line 293: "total IPLs"? Does this now include archaeal IPLs or not (see line 291)?

Lines 303-317: This section should be moved to the discussion. See also earlier comments on the distinction between the photic zone and remainder of the water column.

Line 319 and Fig. 4: The whole distinction between major and minor IPLs is rather artificial. It becomes especially confusing when minor IPLs are normalized on their

sum which is a variable part of the IPLs as a whole. It is entirely unclear why this is done other than for "stamp collection" purposes.

Line 332: it would be much more helpful to report absolute concentrations. In that way it can be directly compared with the abundances of archaeal cells. Now, it is normalized to something where it is not at all related to and which varies by more than an order of magnitude (Fig. 3b).

Line 352; formally this statement is incorrect: the chain length was not measured but the number of carbon atoms in the acyl chains. One cannot discriminate between branched and straight chains. The number of double bonds was also not determined since one cannot discriminate between a double bond and a ring. Needs adjustment. Fig. 6 does not really show a lot of useful information since the variation observed is not extensive. I would consider to drop this figure.

Line 366-367: this definition and the one at line 320-321 does not exclude that some groups are both minor and major lipids. Anyhow, this distinction is extremely confusing as mentioned before.

Lines 365-406: only describe IPLs that are useful in the discussion.

Line 383: One cannot make the distinction between an OH group and an epoxy group with the methods applied. Can the authors exclude that these components are formed upon storage of the extract?

Line 408: the authors should make clear why it would be useful to perform statistical correlations between environmental variables and relative abundance of IPLs. This remains entirely unclear. I suggest to skip this entire section.

Line 433: The discussion is extremely lengthy (19 pages of text) and should be focused on the important observations taking into account all the comments made so far. It should be cut by 50% or so. It is, therefore, not useful to provide detailed comments and I will restrict them to more general comments.

Lines 434-487: General overview which is not connected to IPL dataset at all. Requires substantial shortening.

Lines 490-492: This statement needs to be proven by showing some kind of correlation.

Lines 492-497: First time export of IPLs is mentioned; this should be introduced in the introduction since it is important for the reader to understand that IPLs at depth comprise a mixture of exported and newly produced IPLs.

Lines 497-499: Bold statement that is not (yet?) backed up by any data. Does not belong here.

Lines 502-505: Repetitious.

Lines 508-592: Very lengthy discussion assessing nothing really new: the IPLs in the photic zone derive from algae, cyanobactera, and heterotrophic bacteria. This was to be expected and has been shown previously.

Lines 595-641: What I miss here is an answer to the intriguing question: how much of the IPLs detected in this zone can be derived from settling from the photic zone.

Lines 631-641: Again, nothing new here. Have the authors evidence for the presence of specific GDGTs derived from Thaumarchaeota (i.e. crenarchaeol)? It would be fair to refer to the original literature for the detection of HPH GDGTs in Thaumarchaeota.

Line 648-653: It is highly unrealistic to suppose that PUFA-containing IPLs would be produced in-situ in the ODZ. It seems the authors agree but the text is really confusing.

Line 655: "due to the increase in bacterial abundance"? I guess bacterial abundance is still highest in the photic zone. The authors seem to forget that they are looking at relative abundances but when they would calculate absolute abundance a completely different picture emerges.

Lines 659: these genes are much more widespread in the bacterial kingdom.

Line 662: "chain" is incorrect

Line 674: the comparison of the IPL dataset with the FISH dataset is underdeveloped in this manuscript.

Line 693: for PUFAs I would make a clear distinction between C20 and C22 PUFAs and the C18 ones, otherwise the text will become confusing.

Lines 702-703: This strongly depends on the core of the GDGT IPLs detected. Crenarchaeol has only been detected so far in cultures of Thaumarchaeota. Suggestions that it may derive from euryarchaeota or crenarchaeota are only based on very indirect evidence and quoting these references (and not many other literature) in this context is only confusing the issue. In fact, one way to shorten this paper is to take out all the data related to isoprenoidal lipids. Part of this data has been published before (Xie et al., 2014) and the data reported here do not provide any new insight.

Lines 712-825: Extremely confusing title. We just had a very extensive description of how species composition could influence IPL distribution. This section seems to start all over again (lots of repetition). The statistical data treatment is doubtful as explained before. With all the major changes in environmental parameters (light level, oxygen concentration) and its consequences for species composition it is impossible to link changes in nutrient concentrations to differences in IPL distribution. Such studies should be done first with microbial cultures and then, perhaps, these data can be used to interpret environmental datasets like this one.

Figures 1: Explain how (software; kriging method) plots b-e were produced. The figure suggests (but the caption does not explain) that only at stations 1, 2, 5 and 8 full CTD casts have been obtained. Stations 1-8 are almost 3000 km apart. Is it statistically significant to perform interpolation between these stations for the deep (>200 m) waters? One can observe all kind of changes (as shown by colour changes) that are hard to understand from having only 4 data points over 3000 km. Specify the depth scale; it does not seem to be linear but it is not specified in the caption. Most of these

comments also apply to Figs. 2,3, and 5.

Table S3: The authors cannot report the number of double bonds; they determined the DBE number but they cannot discriminate between a ring or a double bond as far as I can tell. The table should be carefully checked; there are instanced where the FA combination says 18:4/18:4 but the number of bonds in nine. It is also not clear why sometimes they report the FA combination and sometimes not even for the same type of IPL. They should also specify where the relative abundance is normalized upon. It would be more convenient for the reader when these values are followed by a plus/minus sign and then the SD.

---

## Author Comment (AC1) · 4 Jun 2018

Response to Referee #1

This study aims to determine the intact lipids in suspended particles in the water column using samples collected from ocean oxygen minimum zones from the east/north pacific. OMZs are important marine ecosystems particularly with regard to oceanic N cycles. The comprehensive data presented in this study has significantly advanced our understanding of IPLs in this unique environment. This reviewer has no major concerns. Some specific comments are listed below for further improvement of the manuscript.

We thank Referee #1 for the positive comments.

L93, it would be very useful to specifically discuss previous IPL studies on OMZ samples, e.g ETNP, ETSP, Arabian Sea etc in discussions.

Most of the previous IPL studies performed in other OMZ regions have either focused on very specific IPLs for specific processes (e.g., ladderane lipids and HPH-crenarchaeol for ammonium oxidation in the Arabian Sea) or only discussed surface water IPL distribution of phototrophic organisms (in the ETSP). We therefore consider this study to be the first comprehensive study of IPLs in OMZs. Nevertheless, we made sure to refer to previous IPL studies in the discussion where appropriate, e.g. lines 564-565, lines 667-668.

L101 and throughout the MS, most previous studies have used MGDG, DGDG and SQDG to refer to mono- and di glycosyl- DAG and sulfoquinovosyl DAG, please change to these commonly used acronyms for the sake of consistency in literature.

We are aware that we use different abbreviations for these glycolipids than are often used in the literature. However, since we want to be consistent with our nomenclature, i.e. we are also calling mono- and diglycosyl GDGT 1G-GDGT and 2G-GDGT, we opted to stick to the currently used acronyms 1G-DAG and 2G-DAG. This nomenclature is also relevant when describing head groups with different core lipid structures, i.e. SQ-DAG and SQ-AEG.

We would also like to point out that there is plenty of literature in glycolipid research, particularly bacterial glycolipid research, where besides MGDG and DGDG (which are typically the acronyms for the specific thylakoid lipids monogalactosyl and digalactosyl diacylglycerol), the other sugars are referenced as Glc-DG, GlcGal-DG, depending on sugar type (Glc for glucose, Gal for galactose, etc.). We are therefore not that unique with our chosen nomenclature, which specifically highlights that we do not know the types of sugars (which surely change with the source organisms).

Since we are defining the used acronyms in the text (we also now added a footnote for clarification, page 6), we do not think this to be very problematic.

L104-5 , l307-9; this may be a little misleading since DGTS has been found in a wide range of marine heterotrophic bacteria.

We noted here that DGTS has also been detected in marine heterotrophic bacteria in phosphate-limited environments (line 105-106).

L116, please refer to recent study of Hunter et al., AEM doi: 10.1128/AEM.02034-17 for novel diglycosylceramides found in Thalassiosira.

Done.

L214-215, the authors have referred to previous studies for mass spectral interpretation and IPL assignments. It would be very useful to summarize and synthesise these information in a table (or in the supplementary information) and to detail the criterial for IPL identification. Presumably IPL assignment is based on comparing to retention time of standards (where applicable) and characteristic MS/MS patterns, representative characteristic ions or characteristic neutral loss. How has DGTS but not DGTA been conclusively assigned in this study? Has DGTA been found in any samples?

There are on average between 600 and 800 compounds that are identified and quantified in each sample. Since IPL identification is quite complex, it will be difficult to provide all the necessary information in a comprehensive table that will explain each lipid identification. In spite of this, upon the reviewers' request, we included a table in the Suppl. Material (Suppl. Table 3) that provides examples how mass spectral assignment of lipids was conducted. Furthermore, we clarified in the text how we identified DGTS over DGTA (lines 220-223).

L223, for unknown aminolipids AL1 AL2, do the authors have any hypothesis of their structures based on MSn fragmentation patterns (suppl fig 4)? What are the possible amino head group structures? Have accurate ms of AL1, AL2 been obtained?

Unfortunately, we have not more insights or hypotheses on the headgroup structures of AL-I and AL-II other than the ones provided in the text and figure caption. We have accurate masses of AL-I and AL-II fragments up to the third decimal point, which is why we provide sum formulas for the potential headgroup fragments (see Suppl. Fig. 4), but unfortunately we cannot provide any further insights on their structures.

The authors mentioned CSRD FISH data in supplementary dataset but did not mention how this was done in the materials and methods.

This data is from Podlaska et al. (2012), the methods for the CARD-FISH analyses are also described in this paper. We made a reference to this paper in the respective figure caption.

My general impression for discussion is that it can be shortened significantly.

We revised and shortened the discussion substantially from the originally 19 (Word) pages to 13 pages.

It is a pity that no microbial diversity data were obtained in this study as one would like to see the correlation between specific microbial groups and IPLs, which may provide clues for the origin of these lipids, particularly w.r.t. to AL1 and AL2.

We agree that having microbial diversity data would have greatly improved this manuscript and we will ensure that such data will be available for future IPL studies. We did, however, try to correlate the IPL data with the available CARD-FISH data, but unfortunately did not see significant correlations (see section 4.2.2).

Section 4.1.1 two recent papers (Carini et al, Sebastian et al) have shown marine heterotrophic bacteria are also abundant in MGDG. These need to be discussed here in line with these new evidence.

We now added a sentence stating the potential for heterotrophic bacteria to be sources for these glycolipids (line 482) and then refer to the acyl side chains to further delineate if bacteria are indeed potential sources or not.

---

## Author Comment (AC2) · 4 Jun 2018

Referee N.J. Bale

General comments

This study examines the intact polar lipid (IPL) distribution in suspended particulate matter (SPM) from four stations in the oxygen minimum zone (OMZ) of the Eastern Tropical North Pacific (ETNP). It aims to link the IPL distribution of different water column zones with the microorganisms found there and to examine the ecophysical adaptions to the different zones of the OMZ. This is an extensive data IPL set and authors have theorized which groups of microorganisms are responsible for which IPLs. The strength, as I see it, in this study is the examination of how the IPL distribution changes across the different zones of the water column due to the changes in the bio- geochemical environment. Indeed, due to the generic nature of many of these IPLs it is only possible to put forward tentative assignments of their sources, whereas examining changes in the lipid composition with changing environmental parameters provides more solid information. Overall, I recommend this article for publication with the following edits and with suitable responses to my questions on the analytical methods.

We thank Dr. Bale for these constructive comments.

Specific comments

I have two specific comments.

My first specific comment relates to the extraction and analysis of these samples. The manuscript states that the samples were collected in 2007 and (presumably soon after) that they were extracted using Soxhlet-extraction with DCM:MeOH for 8 hours. From an IPL perspective this seems a "harsh" extraction method that has potential to destroy certain IPLs, resulting in a IPL distribution unrepresentative of that in nature. Could the authors comment on whether this would be their preferred method of extraction for IPLs or whether IPL analysis was not the original reason for the chosen extraction method? Indeed the first author has described utilizing the much gentler modified 'Bligh and Dyer' extraction method in other publications relating to IPLs. The authors also describe two different analytical methods used to quantify and to identify structures. Can you state whether analysis occurred soon after extraction in both cases? If not, how were the extracts stored and for how long? If the two analyses were carried out at different times, did extract storage introduce changes in the lipid distribution? I noted that the reference for the second LC-MS method applied (Wörmer et al., 2013), was published 6 years after the samples were collected. Based on the authors' replies, suitable discussion of these issues should be included in the method section.

We have not performed a direct comparison of the Soxhlet extraction technique with the more common B&D ultrasonication technique using the same sample material, therefore we cannot comment on this with exclusive certainty. Nevertheless, we are not very worried about losing significant proportions of IPLs during Soxhlet extraction due to the following observations: (1) tests using microwave extraction showed optimal IPL yields at 70°C, indicating that IPLs may be more thermally stable than thought, (2) we detect presumably more labile compounds such as HPH-GDGTs in similar abundance using Soxhlet extraction as have been reported from other OMZ zones, indicating that these (presumably) more labile IPLs are not preferentially destroyed during Soxhlet extraction.

With respect to analyzing the samples at different time points: the first analysis using the quantitative data on the LC-ion trap-MS were performed in 2010 and 2011, while the QTOF samples were analyzed 4 years later in 2015. During this time the samples were stored in dry condition at -20°C. Again, we cannot state with absolute certainty that the IPL distribution has not been affected over time, however, based on the following lines of evidence, which were accumulated over more than a decade worth of experience in IPL analysis we are again not too worried about this: (1) we typically

analyze IPL standards every two months (together with our samples) that we store at -20°C over several years. So far these IPL standards have showed no selective degradation of compounds over time, thus indicating that relative IPL abundances will not be affected by storage at -20°C, (2) re-analysis of total lipid extracts that contained an internal standard within a 1.5-year timeframe gave similar absolute concentrations of different IPL species with different headgroups (this is unpublished data). This again indicates that there very likely are no significant selective changes in IPL abundance occurring over year-long storage of IPLs, at least none that would significantly affect the already existing uncertainties in IPL quantification.

Since also reviewer #3 expressed similar concerns, we added some explanatory sentences in the methods section with respect to these issues (see section 2.3).

My second specific comment relates to the length and depth of the discussion. This is a subject I find very interesting and yet I felt rather weighed down in information at certain points. I feel it would aid the reader if this was shortened and made more succinct.

We revised and shortened the discussion substantially from the originally 19 (Word) pages to 13 pages.

Technical corrections

Line 52 - change to "the subsequent"

Done.

Line 70 - change to "(ENTP), situated off the. . ."

Done.

Line 93 - change to "(IPL) are the main building blocks of cellular membrane and may"

Done.

Line 98 - change to "the North Sea"

Done.

Line 100 - you could also include in your reference list here the Western English Channel (White et al., 2015).

 "The combined effects of seasonal community succession and adaptive algal physiology on lipid profiles of coastal phytoplankton in the Western English Channel. D.A.White , C.E.Widdicombe, P.J. Somerfield, R.L. Airs, G.A. Tarran, J.L. Maud, A. Atkinson. Marine Chemistry 177 (2015) 638–652)."

Done.

Line 129 - Remove 'here' to read "Notably, replacing. . ."

Done.

Line 141 - change to "extension of that of Xie"

Done.

Line 142 - change to "at two stations described here (station 1 and 8)"

Done.

Line 159 - Should this be "VERTEX I and II" ?

Yes, this has been corrected.

Line 160 The Martin et al. (1987) reference is missing from your reference list.

Is now added.

Line 177 - should define GFF at first use.

Done.

Line 223 - insert "response could not be corrected for"

Done.

Line 350 - The term amino lipid and betaine lipid seem to be used interchangeably throughout the manuscript. Could this be defined at one point?

This should not be the case, because aminolipids refer to the sum of betaine lipids, ornithine lipids and the unknown ALI and ALII (see also legend in Fig. 4) and betaine lipids are just betaine lipids, i.e. in our case DGTS and OH-DGTS. Since ornithine lipids and the unknown AL are often negligibly low in abundance the bulk of aminolipids are indeed dominated by betaine lipids. We checked the manuscript to make sure this distinction is clear and that these terms were not used interchangeably.

Lines 434-439 - This introductory sentence is too long to read and needs to be broken up or shortened.

Done.

Line 436 - replenishment that produces

Done.

Line 454 - "Podlaska et al. (2012)"

Done.

Line 482 - "by Xie et al. (2014)". I have noticed this citation format error in more places. Please change throughout.

Done.

Line 489 - "coinciding with high Chl-a concentrations, reflecting"

Done.

Line 489 - What do you mean eukaryotic rather than microbial? Eukaryotes can be microbes and microbes can be eukaryotic. Do you mean eukaryotic rather than prokaryotic? But also in your results section, 3.1, you state that Prochlorococcus (not eukaryotes) were an important component of the photoautotrophic community. Hence I think the correct thing would be to say "photoautotrophic rather than heterotrophic". Is this not what you wanted to say here? Check you have this correct throughout the manuscript.

We agree that term prokaryote should be used here, rather than microbe. We checked for consistency throughout the manuscript.

Line 497 - pluralize "IPLs"

Done.

Lines 540 - 551 - This section is confusing because you contradict each statement. You state that Eukaryotic phytoplankton and cyanos are assumed to be a major source of PG-DAG. Yet you then state that heterotrophic bacteria can also be a dominant source. Maybe using conjoining words "however" and "although" would make this section flow nicely.

We revised this section for clarity.

Line 547 - change to "we therefore suggest that"

Done.

Line 547 - remove heterotrophic bacteria. Cyanobacteria are not heterotrophic bacteria.

Done.

Line 576 - insert "abundance of less"

Done.

Line 586 - change to "fatty acid" or "acyl" rather than "fatty acyl"

Done.

Line 601 - change to "are <20"

Done.

Line 637-638 - remove tab within word "thaumarchaeota"

Done.

Line 648 - insert commas "zone, and that. . ..1993), became"

Done.

Line 667 - replace "microbial" with "bacterial"

Done.

Line 679 - remove "shallower" as it is redundant.

Done.

Lines 694 - 670 - Please make this long sentence shorter or break it into two. You repeat the same words "exported, fossil and signal" twice.

This entire section has been revised and shortened.

Line 767 - insert "waters of the phosphorus-limited"

This entire section has been revised and shortened.

Line 768 - insert "to the phosphorus-replete"

This entire section has been revised and shortened.

Line 770 - insert "observation, the relative abundance"

This entire section has been revised and shortened.

Line 808 - change to "a myriad of bacterial sources"

This entire section has been revised and shortened.

Table 1 - Make the columns wider so that the cell contents all lie on one line. Table 1 caption - should this read "where $p < 0.05$"

The table has been revised accordingly.

Figures 1,2,3 and 5 - Can you indicate the four water column zones on these figures. Perhaps with lines that join the specific depths at which the regions are defined (as was done in figures 4 and 6).

Figures 1, 2, 3 and 5 have been completely revised according to reviewer #3 request and the water column zones are now indicated.

Figure 2 - unnecessary brackets around nitrite in panel b

Figures 2 has been completely revised

---

## Author Comment (AC3) · 4 Jun 2018

Referee #3

This paper provides a very detailed account of the IPL composition in the waters of the oxygen-deficient zone (ODZ) of the Eastern Tropical North Pacific. It adds to the growing inventory of IPL data. The authors claim that it contributes to our understanding of these systems. I am not entirely convinced. There are also a substantial number of issues related to the analytical methodology that need to be resolved before this paper can be published.

The general aim of the paper is "to evaluate the microbial ecology and ecophysiological adaptations that affect organisms inhabiting the OMZ in the context of biogeochemical cycles" (line 30-32). However, when you read the remaining part of the abstract this does not materializes. It ends with a rather vague statement about potential phosphorous limitation, which in the light of the analytical constraints (see below) may be even weaker. I strongly suggest to remove these kinds of claims from the text and just focus on what the paper is about: an inventory of IPLs in the water column of this region.

We thank the reviewer for the constructive comments. We believe that this study contributes to our understanding of the microbial ecology of the system, as (1) a large proportion of the microbial communities in OMZ currently remain grossly uncharacterized (cf. Podlaska et al., 2012) and IPL analysis can at least help evaluate bacterial vs archaeal sources, and (2) we report on a variety of changes in the lipid composition that relate to changes in biogeochemical zones and may represent ecophysiologic stress adaptations. We thus believe that our statements in lines 30-32 are justified. Nevertheless, we slightly modified other instances in the text where our interpretations might have been too bold (e.g., lines 468-470). While we agree that this study may not be able to fully explain the observed IPL distributions, we still provide ideas of what their functional roles or biological sources may be. Thus, this study lays the groundwork for future investigations that may probe the suggested sources and functions of the lipids we report.

In terms of trustworthiness of the presented data due to analytical issues: Even if there may be some issues with respect to providing absolute numbers, the detection of these 'unexpected' lipids in the ODZ cannot be dismissed and remains to be explained. Secondly, we do not believe that the presented data is prone to more bias than is usually the case for any lipid-based quantification. Finally, this is not the first and only study that observes non-phosphorus lipids in anoxic environments, therefore we are encouraged that our data is credible.

While revising the manuscript, we considered to not make bold claims that we cannot substantiate and aimed at making our main conclusions and take home messages clearer. Furthermore, we addressed potential analytical biases (see section 2.3)

The authors do report absolute IPL concentrations (Fig. 3b) which show an order of magnitude decrease over the first 100 m of the water column. This is in line what would be expected since this is the zone where primary production is taking place (as is also revealed by the pigment concentrations) and the data would allow to discriminate IPLs produced by phytoplankton in the photic zone from IPLs (produced in much lower concentration) by prokaryotes residing below the photic zone and within the ODZ. However, the focus in the paper is too much on relative abundances of IPLs (e.g. Fig. 5) for unknown reasons. The discussion should be more focused on the zone where IPLs are primarily produced (i.e. the photic zone) vs. the remainder of the water column that may be influenced by IPLs in descending particles (i.e. produced in the upper water column) and additional production by prokaryotes. In this discussion, it should be taken into account that PUFA-containing IPLs may degrade faster than other IPLs. Now, the total inventory of IPLs is too much discussed in terms of relative abundance in relationship with nutrient profiles and other environmental parameters over the whole water column, which is too simplistic.

The main focus of the paper lies in observing shifts in community composition by looking at relative changes in IPL composition and we are furthermore putting this into context to the different biogeochemical zones. For this reporting the absolute concentration is only of secondary importance.

It is true that quantitatively IPLs are most abundant in the photic zone and that most of the export of IPL therefore likely occurs from the photic zone (see also Kharbush et al., 2016, OGC 100, 29-41 for more details on this). However, this observation is not very relevant for this paper, because here we focus on the suspended particulate material (SPM, i.e. the in situ living microbiota) and are not looking at exported material as we are filtering out larger particles of marine snow.

Since we see shifts in the relative IPL composition (both in the head group and core lipids) within the different zones we are fairly confident in stating that IPLs are produced throughout the entire water column by different microbial communities and not primarily in the photic zone as stated by reviewer #3.

We would like to highlight here again the importance of microbial processes in ODZ (as introduced in our introduction) and that many important global element cycles, such as the nitrogen cycle are affected by microbes living BELOW the photic zone. Therefore, we see no reason why in this study we should solely focus on the photic zone. This would be completely beside the point we want to make with this study.

In addition, higher absolute abundances do not equal a higher significance: Since small sized prokaryotic picoplankton may not produce as abundant biomass as phytoplankton does, diagnostic IPLs that may be indicative of either changes in the microbial community composition or their adaptations to environmental conditions may only be present in small amounts. For instance, ladderane lipids that are diagnostic for anammox bacteria are only present in very low amounts in the ETNP water column (pg L-1; Rush et al., 2012, OGC 53: 80-87), but this does not mean they are insignificant as these organisms are responsible for large scale denitrification in the global ocean.

In summary, we disagree with reviewer #3 that our approach is too simplistic, instead we would argue that only looking at absolute concentrations and consequently interpreting IPLs in the deeper water column to be mainly derived from exported material out of the photic zone is too simplistic.

The paper also suffers from too much detail. It is very hard to follow because every tiny IPL detected is described without a clear environmental implication. The authors should formulate specific research questions (i.e. not understanding the "functioning of OMZs throughout the world ocean" by studying IPLs) and address these. Not every minor IPL detected has to be described!

The manuscript has now been substantially revised, the discussion has been shortened by ca. 6 pages and we made sure to better formulate our research questions (lines 135-143) and the summary of the main results (lines 708 and 731).

There are also a number of issue related to the analytical methodology of analyzing the IPLs. Adequate answers should be provided on all issues since this may seriously affect the interpretation of the data.

We have revised the manuscript accordingly and addresses potential issues with respect to the analytical methodology (see section 2.3).

1) Filtration. The authors used 0.7 micrometer glass fiber filters for filtration. The limitations of the use of this filter size has extensively been discussed elsewhere and the authors acknowledge the limitations of their approach by admitting that they might be missing smaller cells. However, they should also mention that the pore size will decrease during the filtration process and thus will recover a (variable) part of this material. More importantly, they should stress that this does not only lead to "minimal values" for IPL concentrations but that it may also affect the distribution of IPLs that they report (as percentages). Furthermore, they used a prefiltration device to "remove most eukaryotes"

(line 169). It is hard to believe that this will remove most of the algae; if so this would also strongly affect their interpretations.

We added a sentence in the methods section to address variable material recovery due to shrinking filter pore size (lines 165 and 166). We revised the sentence on pre-filtrations and changed 'most eukaryotes' to 'larger eukaryotes'.

2) Extraction. The IPLs have been Soxhlet extracted (i.e. boiling DCM/methanol for 8 h). Although this is a common method for extraction of less labile lipids, for IPLs it is rather unusual since these are very labile and the commonly applied method for this type of work is Bligh Dyer extraction at room temperature and controlled pH conditions. The authors should present data to show that their extraction method does not alter the IPL distribution (i.e. their main target of study) due to the fact that some IPLs are more labile than others (e.g. in the ratio of phospho vs. non-phospho IPLs, which plays an important role in the discussion).

We would like to refer here to our comments to the second reviewer Dr. Bale as she raised similar concerns. As stated above (and now also in the manuscript, lines 195-197), Lengger et al. (2012, OGC 47:34-40) compared the typical Bligh and Dyer extraction with Soxhlet and ACE extraction. While overall the IPL yields were best with the Bligh and Dyer extraction, Soxhlet extraction did yield also more presumably labile phospho-IPLs (such as the HPH-GDGT). We are encouraged that we did not greatly discriminate against these allegedly more labile compounds as we abundantly detect HPH-GDGT in our samples. Having said this, our own (unpublished) experiences in comparing various extraction techniques often tell us one thing: extraction efficiencies often predominantly dependent on sample type and even the same extraction techniques may yield different yields in different environments. Extraction efficiencies thus affect any lipid study and are simply part of the known quantification biases that are associated with lipid-based studies (as is also the case for any quantitative gene-based technique for that matter).

We have raised awareness of the 'extraction efficiency issue' in the methods section so that the reader is aware that this problem exists (lines 195-197).

3) Analysis. The experimental description indicates that the IPLs have initially been analyzed by HPLC-ESI-IT-MSn using the same system as described by Schubotz et al. (2009). In the meanwhile, this group has developed improved methods of analysis of IPLs (e.g. Wormer et al., 2013) and the question arises why these "old" results are still used since the samples were also re-analyzed using these new methods. It does not become clear how IPLs can be quantified with one method and identified by an- other method (lines 206-207) using an entirely different separation system. One issue that should also be addressed is the timeline of all these analyses. Once extracted, IPLs cannot be stored for a substantial time without significant alteration/degradation. Schubotz et al. (2009) in their very much related work of IPLs in the Black Sea stated

"Three years after primary qualitative analysis the samples were spiked with C16-PAF (1-O-hexadecyl-2-acetoyl-sn-glycero-3-phosphocholine) as injection standard and re- run for quantification. Slight changes in the relative distribution of IPLs were observed within the two runs. In particular the differences were identified as a selective loss of the glycolipids Gly-DAG, Gly-Cer, Gly-GDGT and 2Gly-GDGT (data not shown). We interpret this loss as a sign of selective degradation of glycolipids during storage." So, an important question is how much time evolved between these two analyses and can the results still be compared?

We apologize that we were not very clear in describing the sequence of IPL analysis in this study. We have now substantially revised the methods section 2.3 to clarify when which samples were run and why. Unfortunately, for our second IPL runs in 2015 we are not able to provide absolute quantitative numbers since in between 2010 and 2011 the samples were used for multiple other analyses (published

in Zhu et al., 2016, EM 18: 4324-4336) where overall total lipid extract losses could not be accounted for anymore.

With respect to the selective degradation of IPLs, we would also like to refer to our comments made to reviewer #2 and in addition we would like to state that we did not observe a similar selective loss of glycolipids between the runs that were performed in 2010/2011 and 2015.

4) Quantitation. It does not become clear from the experimental description if the commercially available standards were run with the HPLC-ESI-IT-MSn system that was used for quantitation. If so, the results (response factors) should be reported. If not, there is a serious problem since response factors should be determined on the same system. The whole procedure of quantitation should be made clear. An "injection standard" is mentioned but it remains unclear what was done exactly. Why didn't the authors use an internal standard that was added to the extract? Such a standard would at least have been exposed to the same conditions as the IPLs of the samples (e.g. during storage). The authors should also report the analytical errors of their determinations. The data they now report (e.g. Table S2) are extremely accurate (e.g. a relative abundance of 16.68 % SQ-DAG in station 1 at 35 m). I would expect that the analytical error is 10-20%, so the reported data are far too accurate unless the analytical error is extremely low. This also holds for many of the other data: the reported accuracy of absolute pigment concentrations is also (far) too high and so is the data reported in Table S3 (if the SD is larger than the figure itself is odd to report three or four significant numbers).

Again, we apologize that our quantitative approach was not clearly formulated. We have now revised section 2.3 extensively. We can assure reviewer #3 that the response factors were determined for the same instrument that the samples were run on – anything else would be unacceptable. For quantification of IPLs response factors have to be newly determined every time samples are being run, because not only do they change from instrument to instrument but also over time for the same instrument due to re-tuning after every cleaning. We therefore determined response factors for the 2010 and 2011 runs that were performed on the ion trap system and for the 2015 run on the HPLC-QTOF-MS.

Since response factors are so variable and not only change from instrument to instrument but also within months, we are apprehensive in providing the absolute response factors as this will not be of value to anyone. Instead, we are providing the range of standard variation (line 243) and we have also now provided detailed information on the used standards for response factor correction for the different instruments in a table in the supplementary material (Suppl. Table 2).

Additional comments:

Line 40: It is useless to compare relative trends in IPLs with absolute trends of environmental parameters. To this end, absolute concentrations (like you have for pigments) need to be used.

As already stated above, this study focused on relative abundances of IPLs in order to track (relative) changes in microbial communities and their ecophysiologic adaptations. Solely looking at changes in absolute abundances would not provide this level of information, therefore we disagree with the reviewer that looking at relative trends is useless.

Lines 68-69: . . .. . .but not provided in this paper, so remove this sentence.

This sentence is a general statement and we don't see the need to remove it just because we cannot provide the answer for this in our study.

Lines 117-119: strange sentence

The sentence was revised.

Lines 119-124: I think this overview should be limited to papers describing intact IPLs in the water column. For example, the data presented by Lincoln et al., 2014 are not really solid IPL data. Turich and Freeman, 2001 and Hurley et al., 2016 present only core lipid data.

This section has been revised and references have been removed.

Line 125: It is not discussed how IPLs can be used as taxonomic indicators. This seems pretty relevant information for this type of study

We believe that we have extensively introduced the concept of using IPLs as taxonomic markers in the paragraph above this referenced sentence, e.g., glycolipids for phototrophic organisms, betaine lipids for lower plants and algae and specific phospholipids for either phototrophic organisms or bacteria.

Line 139: in what way is this approach "complementary"?

This whole paragraph has been revised (lines 135-143).

Line 146-147: here the authors promise that we should learn a lot (deeper insight into biogeochemical cycling, functioning of OMZs throughout the world ocean) but this is grossly overstated.

This sentence has been revised according to the statements we can actually make (line 142-143).

Lines 152-152: data on coordinates of sampling stations is incomplete.

Revised for completeness (lines 148-149).

Line 166: provide details on sampling volume

Pore water volumes are now listed in Supp. Table 1

Line 186: provide details on pore size of filter

Done.

Line 200: storage at -20 degrees C is not sufficient to prevent alteration; IPL extracts should be stored at -80 degrees C and even then, distributions may change. How long were the extracts stored before analysis for IPLs?

We wonder on which study the reviewer bases this statement. To our knowledge no literature data exists that tested IPL labilities at -20 vs -80°C. We have acquired knowledge on IPL analyses and stabilities for more than 15 years and based on our experience (re-analysis of the same samples and standards, some of which have been stored at -20°C and -80°C), IPLs are not as labile as they were originally assumed to be.

Since both reviewers #2 and #3 have expressed concerns on the storage and analysis of IPLs we addressed these issue in the revised manuscript. (lines 202-204).

Line 210: different column than in other analysis. Why?

Because the method has improved (as the reviewer has already noted in comment 3 above regarding separation of compounds, peak shape, etc.) and this is the current state of the art of analysis of IPLs and we wanted to analyze the samples with the best available method. This has been clarified in the revised methods section 2.3.

Lines 217-219: provide more details on these standards. What are the acyl moieties of these standards? How are the response factors affected by unsaturations in the acyl moieties? What was the time between the arrival of these standards in the lab and their measurement? How were they stored? It is known that these standards degrade over time upon storage. How often were these standards run? Before each batch of analyses? How did the response factors vary over time? Answers to these questions are essential for getting a feel for the confidence we can have in the reported relative abundances of the IPLs.

We now included a Table in the Suppl. Material (Suppl. Table 2) that contains all the detailed information on the standards that were used. With respect to the effect of the acyl moieties, yes there are differences in the response according to the chain length, but the effect of acyl chain length does

not affect the response factor as much as the head group does (see also Popendorf et al., 2012, Lipids, 48: 185.195 for details on this). We hereby would also like to point out that it is impossible to have a standard for every single analyte (with different chain lengths and/or unsaturation/ring) that exist in a sample, considering that we are identifying and quantifying 600 to 800 compounds in each sample. Firstly, many of these compounds are not commercially available and secondly it would be utopian to prepare the missing standards in the lab due to the required biomass and necessary time. Due to these mainly practical reasons, it is generally accepted in the IPL community that response factors are merely corrected for the different head group classes.

We cannot stress enough that IPLs are not as labile as reviewers #2 and #3 point out. It is unclear what the reviewer means with "it is known"? We are not aware of a reference supporting this statement. We run standards every time we run samples, firstly to get an understanding of the performance of our system over time and secondly to use them to correct for response factors. The standards change over time as the instrument changes over time, this lies in the nature of every instrument and this is why we run standards to correct for these natural instrumental fluctuations (as should be done for any instrument and method).

Line 224-224: why would the unknown response factors be in the range of the measured standards? This is not a scientifically acceptable statement in this way. Just say that they are unknown and what you have assumed to be able to calculate a concentration.

The sentence has been revised.

Lines 258-262: provide references to indicate that pigments can be used to reveal this information even at the species level (e.g. Rhizosolenia).

Pigment assignments were done according to DiTullio and Geesey, 2002 (the reference is provided in the manuscript) as is usually the case for pigments in POM samples.

Line 267: secondary maxima are not revealed in Fig. 1e.

Between 300 to 400 m there are lighter orange shadings compared to the red colors above and below. Based on the reviewers comments we exchanged the section plots with XY-plots. Hopefully now the secondary maxima can be better seen.

Lines 268-269: Not evident from Figs 2a-b.

Based on the reviewers comments we exchanged the section plots with XY-plots. Hopefully now the described trends are better visualized.

Lines 273-276: So, the whole system is NOT P-limited!

Yes, and no, as this is exactly the point we want to make: From a common stand-point of oceanographers the whole system may not be P-limited (nutrient limitations are based on growth stimulation with nutrient amendment experiments and micromolar phosphate concentrations are concentrations where addition of phosphate may not stimulate additional growth). However, when one looks at bacterial cultures, these are exactly the phosphate concentrations where bacterial cultures replace phospholipids with glycolipids and are indeed growth limited. We have revised the text to clarify this (lines 701-705).

Line 282: How were absolute concentrations obtained?

See methods section 2.3 for this information.

Line 284: Secondary maxima are not observable in Fig. 3d.

Based on the reviewers comments we exchanged the section plots with XY-plots. Hopefully now the secondary maxima can be better seen.

Line 289-291: It would be logical to introduce first all IPL classes observed. Why are absolute concentrations of IPL classes not described and is the manuscript concentrated on relative abundances?

We opt to describe all the IPLs in detail later in the text because we find it logical to first provide a general overview of the grouping of glycolipids, phospholipids and aminolipids with depth. Based on the reviewers' suggestion we now mention the diversity of different IPL classes first and refer to Figure 4. Absolute concentrations of IPLs are described (lines 300-304). The reasons why we then focus more in detail on the relative abundances is explained above.

Line 292: "select substitute lipid ratios" is not introduced. It should be introduced in the discussion not in the result since it is an interpretation of the data presented.

We already introduced the (by now well-established) concept of substitute lipids in the introduction and we rather view the substitute lipid ratios as result that should be at least mentioned in the results section. We further discuss substitute-lipid rations in the discussion section 4.2.3.

Line 293: "total IPLs"? Does this now include archaeal IPLs or not (see line 291)?

We revised the sentence to make it clear that total IPLs excludes archaeal IPLs (line 312 and 316).

Lines 303-317: This section should be moved to the discussion. See also earlier comments on the distinction between the photic zone and remainder of the water column.

This section has been significantly revised and shortened.

Line 319 and Fig. 4: The whole distinction between major and minor IPLs is rather artificial. It becomes especially confusing when minor IPLs are normalized on their sum which is a variable part of the IPLs as a whole. It is entirely unclear why this is done other than for "stamp collection" purposes.

As explained above we report relative abundances of IPLs as we want to get an understanding of how the IPL composition changes with changing geochemical zones and microbial community composition. For this, minor lipids can be just as important as the major lipids. One example are ladderane lipids, which are highly diagnostic but are only present in very low abundances in the natural environment. If we would have not made the distinction of minor lipids (with an extra plot showing their distribution in Fig. 4) changes in their relative abundance would have not been evident due to their low abundance.

To answer the reviewers question: the distinction between major and minor IPLs was primarily done for visualization purposes in order to identify potential depth-related trends. Furthermore, this distinction also makes it clear which IPLs were used to determine the absolute amounts (as only major IPLs were seen with the ion trap system, while minor IPLs were additionally detected with the QTOF system, see also revised section 2.3).

Line 332: it would be much more helpful to report absolute concentrations. In that way it can be directly compared with the abundances of archaeal cells. Now, it is normalized to something where it is not at all related to and which varies by more than an order of magnitude (Fig. 3b).

As explained above the purpose of this study was to get an understanding of how the IPL composition changes with changing geochemical zones and microbial community composition. For this we do not need to examine changes in absolute concentrations but rather in relative concentration. Furthermore, absolute archaeal concentrations are already reported in Xie et al. (2014).

Line 352; formally this statement is incorrect: the chain length was not measured but the number of carbon atoms in the acyl chains. One cannot discriminate between branched and straight chains. The number of double bonds was also not determined since one cannot discriminate between a double bond and a ring. Needs adjustment.

The reviewer is correct with this and we revised our definitions accordingly. Chain length was changed to number of carbon atoms in the hydrophobic chain and double bonds changed to double bond equivalents (DBE).

*Fig. 6 does not really show a lot of useful information since the variation observed is not extensive. I would consider to drop this figure.*

As described in the text (lines 353-360), we believe there can be significant variation observed in this figure and would therefore like to keep it as one of the main figures.

*Line 366-367: this definition and the one at line 320-321 does not exclude that some groups are both minor and major lipids. Anyhow, this distinction is extremely confusing as mentioned before.*

We agree that at some depths some of the major lipids may be minor (but not the other way around). However, as already mentioned above, this distinction was (a) done for visualization purposes and (b) to distinguish between the ion trap and QTOF runs and we would therefore like to stick to this distinction.

*Lines 365-406: only describe IPLs that are useful in the discussion.*

The discussion has been substantially streamlined and shortened.

*Line 383: One cannot make the distinction between an OH group and an epoxy group with the methods applied. Can the authors exclude that these components are formed upon storage of the extract?*

Based on the MS2 data we are quite confident that we can make this distinction. Whenever we are observing the loss of water in the MS2, we see this as an indication for the loss of a OH-group. Even if a ring opening and loss of the oxygen from the epoxy-group would occur in the MS2 (which we doubt because this would require the addition of a nucleotide), all of the observed MS2 fragments could not be explained (due to the missing double bond that is formed during the loss of a OH-group). Whenever we do not have mass spectral evidence for the loss of water we do state that it could also be an epoxy group (line 531).

We do not have evidence that these oxygenated fatty acids are formed during storage: (1) Firstly, we do not see that these lipids appear (or increase in abundance) in other lipid extracts that we have stored over several years at -20°C. (2) Secondly, we know from the literature that OH- and epoxy-lipids are common components of algae and other organisms under stress conditions, therefore it is not surprising that they would be present in these samples. (3) Finally, if they were formed during storage, why would they not be equally present in all samples and why would they only form in some IPLs and not in all fatty acids in similar proportions? Certainly, additional investigations need to be conducted to further identify their structures and potential sources.

*Line 408: the authors should make clear why it would be useful to perform statistical correlations between environmental variables and relative abundance of IPLs. This remains entirely unclear. I suggest to skip this entire section.*

We respectfully disagree with the reviewer. This is a concept that has been applied before in many environmental studies, including gene-based studies (e.g., Legendre and Gallagher, 2001, Oecologia 129: 271-280; Ramette 2007, FEMS Microbiol. Ecol. 62:142-160; Rossel et al., 2011 GCA 75:164-184). Reviewer #3 seems to have the misconception that changes in communities can only occur in absolute amounts but not relative to each other. In a simple example: if there is quantitatively more light one can expect a relatively higher abundance of phototrophic organisms (which in turn synthesize phototrophic lipids) compared to low-light areas where there will be relatively lower abundances in phototrophic lipids. This is a simple correlation between relative abundances of IPL (or organisms) to absolute concentrations of environmental parameters that can be statistically evaluated, and we therefore do not see the justification to skip this section as it is an essential part of the paper.

*Line 433: The discussion is extremely lengthy (19 pages of text) and should be focused on the important observations taking into account all the comments made so far. It should be cut by 50% or*

so. It is, therefore, not useful to provide detailed comments and I will restrict them to more general comments.

We substantially revised and shortened the discussion by one third from the originally 19 pages to 13 pages.

Lines 434-487: General overview which is not connected to IPL dataset at all. Requires substantial shortening.

Done.

Lines 490-492: This statement needs to be proven by showing some kind of correlation.

This obvious trend can now be clearly observed in Figure 2.

Lines 492-497: First time export of IPLs is mentioned; this should be introduced in the introduction since it is important for the reader to understand that IPLs at depth comprise a mixture of exported and newly produced IPLs.

We actually do not expect the reported IPL signal to represent exported matter from above since we are looking at suspended particulate organic matter and not larger aggregates of exported material from above. Therefore there is no need to introduce this concept in the introduction.

Lines 497-499: Bold statement that is not (yet?) backed up by any data. Does not belong here.

We do not think that this is a bold statement since it is indeed backed up by our data: as described in the results section we observe 24 different IPL classes that also exhibit changes in core lipid composition with depth. What else should this IPL diversity reflect? Nevertheless, we toned the statement down (lines 468-470).

Lines 502-505: Repetitious.

This section was revised.

Lines 508-592: Very lengthy discussion assessing nothing really new: the IPLs in the photic zone derive from algae, cyanobactera, and heterotrophic bacteria. This was to be expected and has been shown previously.

The section was revised and shortened.

Lines 595-641: What I miss here is an answer to the intriguing question: how much of the IPLs detected in this zone can be derived from settling from the photic zone.

Not that much, this is why we show the changes in core lipids (carbon atom number and DBE), this is also stated at several instances in the manuscript (lines, 541-545, lines 573-574, lines 588-591).

Lines 631-641: Again, nothing new here. Have the authors evidence for the presence of specific GDGTs derived from Thaumarchaeota (i.e. crenarchaeol)? It would be fair to refer to the original literature for the detection of HPH GDGTs in Thaumarchaeota.

Yes, as stated in Xie et al. (2014) we detected crenarchaeol and are now also showing the distribution of the core GDGT composition for the individual IP-GDGTs in Supp. Fig. 2. We revised this section according to the reviewers' suggestion (line 562) and are also now citing Schouten et al. (2008).

Line 648-653: It is highly unrealistic to suppose that PUFA-containing IPLs would be produced in-situ in the ODZ. It seems the authors agree but the text is really confusing.

We do not agree that this is highly unrealistic, PUFAs have been shown to be synthesized by marine bacteria (see references cited in the text, lines 582-586). PUFAs could thus very well be produced in situ. Section 4.1.3 has been substantially revised and hopefully it is less confusing now.

Line 655: "due to the increase in bacterial abundance"? I guess bacterial abundance is still highest in the photic zone. The authors seem to forget that they are looking at relative abundances but when they would calculate absolute abundance a completely different picture emerges.

As already explained above, we do not understand the reviewers concern that we cannot use relative abundances of IPL to describe changes in relative community composition and/adaptations to environmental conditions. We disagree that a completely different pictures would emerge when only reporting absolute abundances.

Generally, bacterial abundance is highest in surface waters (with some exceptions at stations 1 and 2, see Fig. 2). However, what the reviewer seems to miss is that relative bacterial abundance is not highest in the photic zone, instead biomass (and IPLs) derived from eukaryotic phytoplankton dominate. In the ODZ, where phytoplankton abundance decreases (or is absent) relative bacterial abundance increases. This is also why we have more IPLs that are derived from bacteria within the ODZ and below.

Lines 659: these genes are much more widespread in the bacterial kingdom.

Yes of course, but we are discussing the IPLs here only in light of what is known from the FISH analyses performed at these sites.

Line 662: "chain" is incorrect

The term was replaced.

Line 674: the comparison of the IPL dataset with the FISH dataset is underdeveloped in this manuscript.

Where possible and applicable we referred to the existing FISH dataset (lines 457, 552, 566, 578 and 637-639).

Line 693: for PUFAs I would make a clear distinction between C20 and C22 PUFAs and the C18 ones, otherwise the text will become confusing.

The distinction we made for this was to call the C20 and C22 PUFAs 'long-chain PUFAs', we tried to be consistent using this distinction.

Lines 702-703: This strongly depends on the core of the GDGT IPLs detected. Crenarchaeol has only been detected so far in cultures of Thaumarchaeota. Suggestions that it may derive from euryarchaeota or crenarchaeota are only based on very indirect evidence and quoting these references (and not many other literature) in this context is only confusing the issue. In fact, one way to shorten this paper is to take out all the data related to isoprenoidal lipids. Part of this data has been published before (Xie et al., 2014) and the data reported here do not provide any new insight.

We disagree with the reviewer that our isoprenoidal data does not provide any new insight as now we also report on the presence of the important thaumarchaeal marker HPH-GDGT, which was not done in the previous Xie et al. (2014) data set. We revised this section of the manuscript and are now also providing supplementary information on the ring distribution of the individual IP-GDGTs (Suppl. Fig. 2).

Lines 712-825: Extremely confusing title. We just had a very extensive description of how species composition could influence IPL distribution. This section seems to start all over again (lots of repetition). The statistical data treatment is doubtful as explained before. With all the major changes in environmental parameters (light level, oxygen concentration) and its consequences for species composition it is impossible to link changes in nutrient concentrations to differences in IPL distribution. Such studies should be done first with microbial cultures and then, perhaps, these data can be used to interpret environmental datasets like this one.

We substantially revised the discussion, including the headers and the order of the different sections. The purpose of the three sections is to provide an overall synthesis and summary of the above described IPL species. As explained above we do believe our statistical evaluation to be relevant and necessary as it does provide additional insight into the zonation of IPLs into different geochemical

zones. We hope that by shortening and streamlining the discussion our main points we want to make are now expressed in a clearer way.

Figures 1: Explain how (software; kriging method) plots b-e were produced. The figure suggests (but the caption does not explain) that only at stations 1, 2, 5 and 8 full CTD casts have been obtained. Stations 1-8 are almost 3000 km apart. Is it statistically significant to perform interpolation between these stations for the deep (>200 m) waters? One can observe all kind of changes (as shown by colour changes) that are hard to understand from having only 4 data points over 3000 km. Specify the depth scale; it does not seem to be linear but it is not specified in the caption. Most of these comments also apply to Figs. 2,3, and 5.

Based on the reviewers comments we exchanged the section plots with XY-plots.

Table S3: The authors cannot report the number of double bonds; they determined the DBE number but they cannot discriminate between a ring or a double bond as far as I can tell. The table should be carefully checked; there are instanced where the FA combination says 18:4/18:4 but the number of bonds in nine. It is also not clear why sometimes they report the FA combination and sometimes not even for the same type of IPL. They should also specify where the relative abundance is normalized upon. It would be more convenient for the reader when these values are followed by a plus/minus sign and then the SD.

We changed the terminology and call it now double bond equivalents (DBE) and also revised Suppl. Table 5 according to the reviewers' suggestion.

---

## Author Comment (AC4) · 4 Jun 2018

[revised manuscript text omitted]

fig02

[Figure]

fig03

[Figure]

fig04

[Figure]

fig05

[Figure]

fig06

[Figure]

---

## Author Response (AR2)

I have reviewed the revised manuscript. It has improved but there remain a number of important issues that still need further attention before this manuscript can be published.

The title of the manuscript still emphasizes the "biogeochemical implications" which are actually fairly minor (see their abstract). The title also claims that it only studies lipids in the OMZ but that is not true. The term "diversity" for lipids seems odd, it is a biological term that should not be used in terms of chemical composition.

We changed the title to "Intact polar lipids in the water column of the Eastern Tropical North Pacific: Abundance and structural variety of non-phosphorus lipids".

The authors have responded in an extensive way to my comments on the analytical methodology. It has become much more clear how the time-line of the analyses has been. They state that they have >10 years of experience with analysis of IPLs and I am not doubting their analytical capabilities at all. However, in the cases where they use rather unusual procedures (Soxhlet extraction of IPLs, re-analysis of IPL extracts after 7 years of storage at -20 degrees centigrade) they have to come up with evidence to back up their statements in the rebuttal and in the text (e.g. lines 195-197; "we believe"). Just the presence of labile IPLs does not mean that the distribution (i.e. relative percentage) has not changed (and it is the distribution that is the focus of the paper as explicitly mentioned in the rebuttal). The authors refer to Lengger et al. (2012) to support this statement. However, this paper uses only Bligh-Dyer extraction. Perhaps, they intended to refer to Lengger et al., Organic Geochemistry 47, 34 – 40 also published in 2012 but this paper shows that the extraction technique has a huge influence on the ratio between glyco- and phospholipids (see Figure 3 in that paper): Bligh Dyer extraction increases the relative amount of the phospholipid by a factor of ten or so. I therefore remain doubtful about the chosen methodology which in view of this is inappropriate.

We meant to cite Lengger et al., 2012 OGC, but mistakenly cited Lengger et al., 2012 GCA, we apologize for this mix-up. We agree with the reviewer that Bligh and Dyer extraction would be the choice method for IPL analysis, however, it is often the case that samples are re-analyzed after newer protocols have been developed and it is not possible anymore for us to perform a different extraction protocol for these samples.
We mention potential extraction issues due to the performed Soxhlet extraction in the revised methods section of the manuscript (lines 196-199) and hope that these now sufficiently address the reviewers concerns.

With respect to the storage issue: why don't the authors show a supplementary figure showing the composition of their IPL standard mixture over time? This would at least solve this issue convincingly with scientific data and would answer my questions is a reasonable way. With respect to the response factors of the standards: I do understand that these response factors are not useful for the community since they are specific for the instrument at that specific moment in time. However, it still would be very helpful to see the change in response factors over time for each individual standard. If the standards are so stable upon storage: why are they renewed every 3 years (which is only 40% of the time between extraction and analysis)?

We added a supplementary figure showing the fluctuations of relative and absolute responses of our different standard mixes over time (Suppl. Fig. 1). The response of each IPL varies over time as the machine is being tuned and cleaned, but there is no obvious indication for preferential degradation of any or all compounds, as absolute and relative responses go up and down (and up again) over the years.

We renew standards not primarily due to concerns of degradation, but mainly because they are either used up or new standards become commercially available. This is also why there is no fixed date when standards get renewed, but always ca. every 2 to 3 years.

A second major point that is not solved in this new version is the distinction between major and minor IPLs (see Figure 4). I reiterate my previous comment: "The whole distinction between major and minor IPLs is rather artificial. It becomes especially confusing when minor IPLs are normalized on their sum which is a variable part of the IPLs as a whole. It is entirely unclear why this is done other than for "stamp collection" purposes. ".

As we stated in the first response to the reviewer, we visualize the minor lipids, because also minor lipids can be environmentally relevant (cf. laderrane lipids). If we would not zoom into the minor lipids in figure 4 as we do, then changes with depth and zonation would not be visible.

Nevertheless, we agree with the reviewer that scaling the minor lipids up to their sum is arbitrary and may be misleading. Therefore, we now decided to scale the minor lipids to their actual relative abundance to the total IPLs. See revised figure 4.

The text has now been changed to define major and minor IPLs (Lines 309-310) but this definition is rather confusing with the figure because it shows many IPL classes that are <10%. The bar plots of the minor compounds shows really weird things; sometimes there is only one IPL, sometimes none, and the most important thing is that it is normalized on something that is extremely variable (i.e. the sum of "minor" IPLs that varies between 0-20%). So, it is entirely confusing way of plotting the data. If the authors want to focus on these minor IPLs, why don't they use the scale of the plots shown on the left of this figure (IPL relative abundance normalized on the sum of all IPLs) but use a scale that runs from 0-2% or so. That would be a much more fair way to present the data and would also eliminate some of the weird things (i.e. "minor" IPLs dominated by one IPL because in these cases the "minor" IPLs represent <1% of the total).

As the reviewer suggested, we now plotted the minor lipids that they amount to their actual relative abundance compared to total IPLs, see revised figure 4.

This issue directly relates to the problem that the authors keep on insisting in their idea that relative abundances are more important than absolute abundances. The problem they generate in that way is directly evident form their Figure 4. At station 8 archaeal IPLs represent 50-80% of the IPLs whereas this is much lower in the shallower waters. Fig. 2e shows that in the shallower waters there are a comparable or perhaps even higher number of archaeal cells. However, in the shallow waters the IPL concentration is much higher. When absolute concentrations of archaeal IPLs would be taken into consideration this discrepancy would likely not exist. Therefore, I strongly urge the authors to look at their data also in terms of absolute concentrations. Can't they combine their measured absolute total concentrations of IPLs in 2010/2011 (which they now hardly use) with the %IPL determined in 2015 to arrive at concentration profiles for individual IPLs? That would allow a much more meaningful ecological interpretation of their extensive dataset. Of course, percentage data also provides some insight but the basic conclusion from Fig. 4 is now: the photic zone contains predominantly IPLs derived from phytoplankton whereas the IPLs in the OMZ are primarily derived from bacteria and archaea. This is hardly surprising; do we really need all these analyses to arrive at this conclusion?

As stated in the manuscript, the 2010/2011 analyses did not consider HPH-GDGT; therefore, we refrain from reporting total GDGT data from the Xie et al., 2012 paper as these only consider 1G-GDGT and 2G-GDGT.
Unfortunately, the reviewer still seems to misunderstand the main point of the paper if he/she believes it is simply to show that phytoplankton dominate the surface waters while bacteria and archaea dominate the deeper layers. Instead, the main point of the paper is to report on the types of lipids that are found with the different geochemical and biological zones. To us it was surprising that in deeper zones (within the OMZ and below) we find IPLs that were previously not assumed to be typical for bacteria.
Again, we can only re-iterate what we mentioned previously: We do not agree that showing absolute concentrations would gain any more insight into our main point. Furthermore, there are several papers that have reported on absolute concentrations of archaeal lipids in oxygen minimum zones (Pitcher et al., 2011, Schouten et al., 2012), and we thus do not see how our reporting of this would be novel.

I also still believe that correlating environmental parameters to relative abundances of IPLs is much less useful than to absolute concentrations. In any case, the statistical data treatment is not really used in the discussion, so this section (Lines 406-429) can be easily eliminated.

We would like to keep the statistical evaluation of the data in the manuscript, otherwise we could not make statements about correlations of lipids with environmental parameters, which is surely of interest to many people in this field. Also, it is not true that we do not use the statistical data in the discussion, we do so in many instances in section 4.2.

The discussion has been shortened substantially but is still lengthy. It starts with a paragraph (lines 432-459) that only reiterates previous findings in this setting. Where these previous finding can be directly related to the IPL profiles they should be mentioned there and this paragraph can be skipped.

As this background information reports on previous findings we do believe that the discussion section is the appropriate placing of this relevant information.

There is also overlap in Lines 468-474. I am also puzzled by the fact that in section 4.1.3 IPLs from two completely different zones (core OMZ and deep oxycline, i.e. without and with oxygen) are discussed together.

We inserted a sentence in line 578 explaining why we discuss these two zones together because similar IPL distributions indicate similar biogeochemistries, even though oxygen is rising in the deep oxycline.

The discussion of the origin of the archaeal IPLs (Lines 558-570) is fairly limited. In the upper OMZ hardly any archaeal IPLs are detected (Figure 4) so this needs explanation.

As we report in line 458 (part of a paragraph the reviewer wanted us to delete) Podlaska et al., 2012 showed low archaeal abundances shallower than the upper OMZ, which matches our IPL data.

Also, the reason for linking them to Group II Euryarchaeota needs argumentation.

We revised the sentence where we linked the IPLs to Group II Euryarchaeota (line 572).

The conclusion (line 708-711) is simply wrong. Not only oxygen is the primary determinant but also light. Without light there would not be any production of IPLs by algae and cyanobacteria, quantitatively the most important IPLs in the water column of the studied area.

We now clarify in the text that not only oxygen, but also light drive IPL distribution (lines 265, 477, 657 and 715).

In summary, I strongly feel that this manuscript still needs quite some revision.

We again revised the methods section as well as the discussion and figure 4 according to the reviewers concerns. In addition, we added a supplemental figure showing the fluctuations of the IPL standard response factors over time.

Line 1: The title of the manuscript was changed

Line 33, 35, 143: 'oxygen minimum zone' was replaced with 'water column' to highlight that we are not only focusing on the OMZ.

Line 203: The methods section was revised to address the Soxhlet extraction issue.

Line 597: Sentence was rephrased according to the reviewer's request.

Line 603: Sentence was added according to the reviewer's request.

Lines 277, 687, 745 and 746: Next to high oxygen content also the presence of light in the surface waters is stressed here, according to the reviewer's request.

We modified Figure 4 and added Suppl. Figure 1 according to the reviewer's request.

[revised manuscript text omitted]

---

## Author Response (AR3)

**Point by point response to the reviews**

I am happy to note that the authors finally acknowledge in their rebuttal that the extraction technique used is not really appropriate for analyzing intact polar lipids (IPLs) as it may cause changes in the distribution, especially with respect to the abundance of the relatively labile phospholipids. They now state in their manuscript "Soxhlet extractions, rather than for example microwave assisted Bligh-Dyer extractions, were chosen at the time because it was the only feasible way to handle the double 142mm filters. Extraction protocol surely can affect IPL distributions; as shown by Lengger et al. (2012) for smaller sediment samples." (lines 196-197). This is a bit weird since the Bligh Dyer extraction technique method was already published in 1959 and it remains unclear why this method could not be used on GFF filters. A quick search reveals that these kind of extractions were already described in 1997 (Macnaughton et al., Journal of Microbiological Methods, 31 19-27) so "at that time" is inappropriate phrasing. They should also specify what kind of effects this extraction technique has on IPL distributions, specifically on the abundance of phospholipids. The effects of their unconventional extraction method should be taken into account in interpreting the data in the whole manuscript.

Different extraction protocols will lead to different results, no matter which sample is analyzed or what the targeted analyte is; this is common knowledge in organic geochemistry. At the same time, there is probably not THE single extraction protocol that is superior to all others when we consider the great chemical diversity of compounds found in environmental samples, let alone the different sample matrices encountered. This said, most established protocols are reasonably good compromises, and so is soxhlet extraction, as long as the limitations are acknowledged. While in environmental chemistry standardization may be useful and practicable, organic geochemists have for decades used non-standardized protocols that are tailored to the sample matrix as well as to the infrastructure available in the respective laboratories. Likewise, the use of a different mass spectrometers or chromatographic separation techniques will probably have similar or greater effects as the criticized choice of an extraction protocol, as already shown for much simpler compounds in numerous round robin tests (for the paleo sea surface proxies TEX86 and UK37, Rosell-Melé et al., 2001, G3 2:2000GC000141Schouten et al., 2009, G3 10:Q03012; Schouten et al., 2013, G3 14:5263-5285;). This approach may limit quantitative comparisons with datasets produced in different laboratories, but certainly not within a coherent sample set with identical methods, as done in our study. It is imperative among organic geochemists to acknowledge these differences and keep these in mind when interpreting the resulting data.

In fact, we have acknowledged that there may be issues with soxhlet extraction technique – hardly "unconventional" - but stand by our use of soxhlet extractions for the very large filters needed to collect particles at low seawater concentrations in the deep ocean: our "at that time" is entirely appropriate. We have pointed this fact out in the revised manuscript. It is true that the Bligh-Dyer protocol has been around for decades, as we know, but as used by other investigations with much smaller and/or freeze-dried samples, it simply will not work for large, seawater saturated filters. We have published papers over two decades using soxhlet for large filters (originally the filters were even larger) (e.g., Wakeham et al., 1995, DSR-I 42:1749-

1771; DeLong et al., 1998, AEM 64:1133-1138; Wakeham et al., 2004, Chem.Geol. 205:427-442; Wakeham et al., 2007, OGC 38:2070-2097; Schubotz et al., 2009, EM 11:2720-2734; Sáenz et al., 2011, OGC 42:1351-1362; Rush et al., 2012, OGC 53:80-87; Close et al., 2014, DSR-I 85:15-34).  The referenced Macnaughton paper does not actually address the efficiency of soxhlet extractions, certainly not for large filters (Macnaughton used 0.25 cm sq filters vs the 2x147 (158 cm sq) mm filters we had), and actually did not measure IPLs.  At the start of a soxhlet extraction, the solvent mixture is DCM:MeOH:water as in the Bligh-Dyer protocol (chloroform is no longer used for health reasons).  The temperature of the DCM:methanol azeotrope in the soxhlet extraction is 35°C compared to 40°C or 60°C in microwave or ultrasonication protocols, and 80°C and 120°C in the ASE protocol of Macnaughton et al., so elevated temperature is not an issue.  We do not see any reason why phospholipid compositions would be adversely affected.  Further, to our knowledge there has not been a comprehensive comparison of extraction techniques that involved soxhlet vs. other protocols for the large samples and for the wide range of lipids we analyzed (in addition to those reported here), so we are unable to comment on this, except to say that more sensitive analytical techniques coming on-line may reduce to need for such large filter volumes.

I still read in the abstract "Abundant non-phosphorus 44 "substitute" lipids within the OMZ suggest that the indigenous microbes might be phosphorus limited (P 45 starved) at ambient phosphate concentrations of 1 to 3.5 µM, although specific microbial sources for many 46 of these lipids still remain unknown." The authors cannot talk about "abundant" because their extraction method does not allow to say anything about the abundance of one type of IPLs over another type of IPL as their extraction method leads to a bias in the distribution (especially if it relates to phospholipids. If the authors fail to build this concept into their manuscript, it remains flawed.  I welcome the changes that have been made to Figure 4 (representation of minor IPLs) although (as indicated earlier) their extraction technique will have a substantial impact on the relative abundance of the phospholipids.  In all tables and figures the authors present in this manuscript they have to mention in the legend or footnote that the extraction technique used has led to a discrimination of the phospholipids. Otherwise, presenting these data is not scientifically correct.

We reject the use of terms such as "flawed" or "scientifically incorrect". By the same criteria, much of the organic geochemical data and papers produced in the past decades were "flawed and incorrect", because they did not always with "the ideal" analytical protocol, especially when measured by modern standards.

As to this further comment on extraction, we have no evidence - nor is there any in the literature that we know of – that shows biases due to using soxhlet extraction, and in any event, we have noted in the revised manuscript that potential biases cannot be absolutely ruled out.  Therefore, we have no reason not to be able to compare compound abundances within our sample set.  We are confident that our results are not "flawed" or "scientifically incorrect". Nevertheless, we exchange "abundant" with "the presence" in the abstract.

It is also nice that the authors now provide data on the analyses of the mixtures of standards over time (the new. Fig. S1). However, the legend of the figure should explain what is shown in the first panel: is it the response of all IPLs comprising the standard mixture? Furthermore, they should specify the initial composition of the three standard mixtures in much more detail (concentration of the specified IPLs). The authors also report relative abundances of specific IPLs (e.g. GDGTs with a variety of polar head groups) but these IPLs are not represented by any of the IPL standards they use in their Standard Mix. How, are they able to derive mass spectrometric response factors for these IPLs.  An explanation is required.

The plot in new Fig. S1 shows the slope of standards measured in different concentrations. We modified the figure caption to make this clearer. We only show select IPL standards to illustrate how response factors change over time. As stated in the methods section and shown in Suppl. Table 2 we used the commercially available standard 'Main phospholipid *Thermoplasma acidophilum*' as IPL-GDGT standard. These analyses were done at a time where we did not yet have our own standards for 1G-GDGT and 2G-GDGT, which we currently use besides the 'Main phospholipid *Thermoplasma acidophilum*' standard.

**List of relevant changes:**

Line 43 in the abstract: Changed "Abundant" to the "The presence of"

Changed caption of Suppl. Fig. 1: Fluctuations in (A) absolute and (B) relative responses of select commercially available IPL standards over time. The values represent the slope of standards measured in different concentrations (usually 100 pg to 10 ng injected on column). Standard Mix A, B and C represents newly prepared standard mixtures. The standard mix used in this study was from November 2015.

[revised manuscript text omitted]

1G-DAG | PG-DAG | DGTS
2G-DAG | PE-DAG
SQ-DAG | PE+PC-AEG | ∑ Minor compounds
| PC-DAG

**Archaea**
**Glycolipids** | **Phospholipids**
1G-GDGT | HPH-GDGT
2G-GDGT

**Bacteria and Eukarya**
**Glycolipids** | **Phospholipids** | **Aminolipids**
1G-OH-DAG | DPG | OH-DGTS
3G-DAG | PME-DAG | AL-1
SQ-AEG | PDME-DAG | AL-II
1G-CER | PI-DAG | OL
1G-OH-CER

**a)** Station 1

**b)** Station 2

**c)** Station 5

**d)** Station 8

fig05

[Figure]

fig06

[Figure]

**Intact polar lipids in the water column of the Eastern Tropical North Pacific: Abundance and structural variety of non-phosphorus lipids**

Florence Schubotz [1*], Sitan Xie [1,¶], Julius S. Lipp [1], Kai-Uwe Hinrichs [1], Stuart G. Wakeham [2]

[1]MARUM Center for Marine Environmental Sciences and Department of Geosciences, University of Bremen, 28359 Bremen, Germany

[2]Skidaway Institute of Oceanography, Savannah, GA 31411, USA

[¶]Current address: Wai Gao Qiao Free Trade Zone, 200131 Shanghai, China

[*]Corresponding author. MARUM, University of Bremen, Leobener Str. 13, Room 1070, 28359 Bremen, Germany. Tel: +49-421-218-65724. Fax: +49-421-218-65715. E-mail: schubotz@uni-bremen

**Suppl. Table 1.** Relative abundance of detected intact polar lipids (IPL) at all four stations (1, 2, 5 and 8) in the Eastern Tropical North Pacific, t.a. - trace amonts, n.d. - not detected. For IPL abbreviations refer to main text.

| Station | Depth (m) | Water filtered (L) | Total IPL (ng/L) | 1G-DAG | 2G-DAG | SQ-DAG | DGTS | PE+PC-AEG | PE-DAG | PC-DAG | PG-DAG | DPG | 3G-DAG | SQ-DEG | 1G-Cer | AL-I | AL-II | OL | PME-DAG | PDME-DAG | PI-DAG | 1G-OH-DAG | 1G-OH-Cer | OH-DGTS | 1G-GDGT | 2G-GDGT | HPH-GDGT |
|---|---|---|---|---|---|---|---|---|---|---|---|---|---|---|---|---|---|---|---|---|---|---|---|---|---|---|---|
| 1 | 3 | 575 | 882.3 | 17.4 | 9.8 | 51.2 | 1.3 | 1.4 | 2.8 | 3.5 | 9.0 | 0.0 | 0.2 | 1.3 | 0.3 | t.a. | 0.9 | n.d. | n.d. | n.d. | 0.5 | 0.1 | 0.2 | n.d. | n.d. | n.d. | n.d. |
| 1 | 25 | 193 | 1357.8 | 18.4 | 9.6 | 34.3 | 5.3 | 4.9 | 3.2 | 5.0 | 13.5 | 0.0 | 0.2 | 1.7 | 0.6 | 0.1 | 1.9 | n.d. | n.d. | n.d. | 0.3 | 0.3 | 0.8 | n.d. | n.d. | n.d. | n.d. |
| 1 | 35 | 1636 | 361.6 | 15.0 | 13.5 | 16.7 | 2.4 | 29.8 | 3.0 | 8.5 | 4.0 | 0.1 | 0.5 | 0.3 | 0.8 | 0.3 | 2.2 | n.d. | n.d. | n.d. | 1.8 | 0.4 | 0.7 | n.d. | n.d. | n.d. | n.d. |
| 1 | 75 | 1013 | 304.0 | 28.5 | 8.7 | 39.8 | 0.9 | 4.2 | 1.5 | 3.7 | 7.7 | 0.2 | 0.2 | n.d. | 0.4 | 0.2 | 1.9 | n.d. | n.d. | n.d. | 0.4 | 0.3 | 0.5 | n.d. | 0.2 | 0.1 | 0.4 |
| 1 | 120 | 1347 | 22.3 | 22.9 | 4.5 | 8.6 | 5.5 | 11.8 | 4.2 | 10.7 | 14.4 | 1.1 | 0.1 | n.d. | 1.4 | 0.2 | 2.9 | n.d. | 0.1 | 0.4 | 0.3 | n.d. | 0.5 | n.d. | 3.3 | 2.6 | 4.5 |
| 1 | 200 | 1578 | 8.4 | 23.2 | 1.0 | 0.7 | 14.0 | 11.9 | 3.8 | 10.9 | 5.7 | 0.8 | n.d. | n.d. | 1.1 | 0.4 | 2.9 | n.d. | 1.1 | 2.3 | n.d. | n.d. | 0.3 | n.d. | 16.4 | 3.2 | 0.3 |
| 1 | 300 | 1337 | 2.9 | 21.4 | 2.9 | 4.0 | 7.8 | 4.5 | 4.1 | 4.9 | 18.5 | 4.8 | n.d. | n.d. | 1.2 | 1.2 | 3.2 | n.d. | 2.8 | n.d. | n.d. | n.d. | 1.2 | n.d. | 14.9 | 2.6 | n.d. |
| 1 | 400 | 1300 | 56.4 | 3.1 | 2.9 | 11.3 | 2.7 | 0.0 | 8.8 | 8.1 | 50.6 | 3.1 | n.d. | 0.4 | 0.4 | 1.5 | 2.3 | n.d. | 2.0 | 0.6 | 0.1 | n.d. | n.d. | n.d. | 1.4 | 0.8 | n.d. |
| 1 | 600 | 1748 | 11.4 | 5.6 | 0.9 | 2.8 | 3.5 | 3.9 | 15.1 | 8.0 | 33.9 | 3.1 | n.d. | n.d. | n.d. | 1.3 | 0.2 | n.d. | 4.3 | 0.1 | n.d. | 0.1 | n.d. | n.d. | 13.3 | 3.8 | n.d. |
| 1 | 725 | 800 | 9.7 | 9.8 | 1.0 | 1.9 | 9.6 | 8.7 | 6.1 | 13.2 | 16.2 | 1.5 | n.d. | n.d. | 0.3 | 0.3 | 2.1 | n.d. | n.d. | n.d. | 0.2 | n.d. | 0.5 | n.d. | 14.9 | 10.4 | 3.2 |
| 1 | 820 | 1571 | 3.3 | 12.3 | 2.0 | 1.1 | 8.4 | 11.4 | 7.3 | 15.1 | 16.8 | 1.8 | n.d. | n.d. | 0.3 | 0.2 | 3.0 | n.d. | n.d. | n.d. | 0.1 | n.d. | 0.5 | n.d. | 9.3 | 9.4 | 1.2 |
| 1 | 1250 | 1374 | 0.8 | 32.4 | 1.3 | 1.9 | 14.9 | 1.9 | 0.8 | 1.6 | 13.0 | 1.4 | n.d. | n.d. | 0.6 | 0.3 | 1.6 | n.d. | n.d. | n.d. | n.d. | 0.2 | 0.5 | n.d. | 21.0 | 6.3 | 0.3 |
| 2 | 3 | 1071 | 266.2 | 9.3 | 18.9 | 35.7 | 6.8 | 4.8 | 1.4 | 8.6 | 6.2 | 0.1 | 0.2 | 1.4 | 0.8 | 0.2 | 2.5 | t.a. | n.d. | 0.3 | 0.4 | t.a. | 0.5 | 1.7 | 0.0 | n.d. | n.d. |
| 2 | 6 | 1166 | 349.6 | 30.1 | 14.9 | 20.3 | 7.2 | 5.9 | 0.8 | 3.5 | 9.0 | 0.1 | t.a. | 5.0 | 0.1 | 0.0 | 1.0 | n.d. | 0.1 | n.d. | 0.4 | 0.1 | 1.4 | t.a. | 0.0 | n.d. | n.d. |
| 2 | 55 | 1647 | 21.9 | 18.1 | 17.5 | 12.4 | 3.7 | 26.5 | 2.8 | 8.9 | 3.6 | 0.1 | 0.2 | 0.4 | 0.3 | 0.3 | 3.4 | t.a. | t.a. | n.d. | 0.4 | 0.1 | t.a. | n.d. | 0.3 | n.d. | n.d. |
| 2 | 85 | 1435 | 160.0 | 24.3 | 7.5 | 33.5 | 3.5 | 7.3 | 2.7 | 7.1 | 4.7 | 0.9 | t.a. | 0.4 | 0.8 | 0.1 | 4.2 | t.a. | 0.6 | 0.6 | 0.3 | 0.5 | 0.3 | n.d. | 0.6 | 0.2 | 0.0 |
| 2 | 115 | 1517 | 50.0 | 7.0 | 6.6 | 18.2 | 5.0 | 7.4 | 5.8 | 9.1 | 21.9 | 2.0 | 1.6 | 0.9 | 1.8 | 0.3 | 6.2 | 0.1 | 1.5 | 0.3 | 0.3 | n.d. | 1.1 | n.d. | 2.1 | 1.0 | n.d. |
| 2 | 200 | 193 | 6.0 | 80.0 | 0.0 | 0.4 | 10.8 | 0.2 | 0.2 | 0.3 | 0.7 | 0.0 | n.d. | n.d. | n.d. | n.d. | n.d. | n.d. | n.d. | n.d. | n.d. | 0.5 | n.d. | n.d. | 6.8 | 0.0 | n.d. |
| 2 | 400 | 1725 | 12.6 | 4.2 | 2.8 | 4.8 | 5.4 | 4.1 | 13.0 | 17.2 | 31.2 | 4.7 | 0.5 | n.d. | 0.9 | n.d. | 1.6 | 0.5 | 3.9 | 1.9 | 0.1 | n.d. | n.d. | n.d. | 1.7 | 0.7 | 1.0 |
| 2 | 600 | 1476 | 11.8 | 2.4 | 0.0 | 12.5 | 2.1 | 1.9 | 0.1 | 1.9 | 34.1 | 11.3 | n.d. | n.d. | n.d. | n.d. | 0.0 | 0.0 | n.d. | 6.1 | n.d. | n.d. | n.d. | n.d. | 6.5 | 13.8 | 7.5 |
| 2 | 830 | 1397 | 7.2 | 6.2 | 3.5 | 6.8 | 7.7 | 12.5 | 8.7 | 11.5 | 15.0 | 1.6 | 1.0 | 0.5 | 1.2 | 0.2 | 5.1 | 0.0 | 2.9 | 1.7 | 0.6 | n.d. | 1.1 | n.d. | 5.1 | 1.8 | 5.1 |
| 5 | 3 | 223 | 244.4 | 23.1 | 16.1 | 18.4 | 24.6 | 3.8 | 0.8 | 4.1 | 4.6 | 0.1 | n.d. | 1.2 | 1.1 | t.a. | 0.4 | n.d. | n.d. | n.d. | 0.1 | 0.8 | 0.5 | 0.1 | n.d. | n.d. | n.d. |

| | | | | | | | | | | | | | | | | | | | | | | | | | | | |
|---|---|---|---|---|---|---|---|---|---|---|---|---|---|---|---|---|---|---|---|---|---|---|---|---|---|---|---|
| 5 | 25 | 128 | 1187.6 | 66.1 | 1.0 | 4.1 | 22.9 | 0.0 | 0.2 | 0.4 | 4.5 | 0.0 | n.d. | n.d. | 0.0 | 0.2 | 0.2 | n.d. | n.d. | n.d. | 0.1 | 0.3 | n.d. | n.d. | n.d. | n.d. | n.d. |
| 5 | 35 | 1683 | 38.1 | 67.7 | 0.7 | 4.0 | 14.5 | 2.8 | 0.4 | 1.7 | 3.3 | 0.0 | n.d. | n.d. | 0.6 | n.d. | 0.2 | n.d. | t.a. | n.d. | 0.0 | 0.8 | n.d. | 1.0 | 2.2 | n.d. | n.d. |
| 5 | 75 | 995 | 174.8 | 20.5 | 11.0 | 49.3 | 2.9 | 0.2 | 0.3 | 0.9 | 8.6 | 1.8 | t.a. | 0.2 | 0.2 | t.a. | 0.3 | t.a. | 0.1 | 0.2 | 0.2 | 0.4 | n.d. | t.a. | 1.1 | 0.8 | 0.8 |
| 5 | 125 | 1289 | 2.3 | 3.9 | 6.7 | 35.3 | 5.6 | 9.3 | 3.6 | 2.8 | 3.6 | 0.8 | 0.2 | 0.2 | 1.8 | 0.2 | 5.6 | n.d. | 11.9 | 0.4 | 0.1 | 0.1 | n.d. | t.a. | 7.1 | n.d. | 0.8 |
| 5 | 250 | 1362 | 1.1 | 41.7 | 0.0 | 0.0 | 0.0 | 0.0 | 0.1 | 0.0 | 0.0 | 0.0 | n.d. | n.d. | n.d. | n.d. | n.d. | n.d. | n.d. | n.d. | n.d. | n.d. | n.d. | n.d. | 58.3 | n.d. | n.d. |
| 5 | 400 | 1365 | 11.2 | 5.4 | 2.8 | 5.0 | 7.7 | 3.4 | 11.8 | 11.3 | 33.0 | 3.4 | 0.4 | n.d. | 0.4 | 0.1 | 2.1 | 0.5 | 5.1 | 1.3 | 0.1 | n.d. | 0.3 | n.d. | 4.0 | 0.8 | 1.1 |
| 5 | 600 | 968 | 21.4 | 3.9 | 4.2 | 6.3 | 5.8 | 2.1 | 17.1 | 15.1 | 22.0 | 5.5 | 1.3 | 0.4 | 2.2 | 0.3 | 4.1 | 0.7 | 5.4 | 2.8 | 0.4 | n.d. | 0.7 | n.d. | 0.0 | n.d. | 0.0 |
| 5 | 830 | 1595 | 12.4 | 5.7 | 3.6 | 7.2 | 7.7 | 12.8 | 8.6 | 11.6 | 15.4 | 1.6 | 1.0 | 0.5 | 1.3 | 0.2 | 5.3 | 0.1 | 3.1 | 2.0 | 0.7 | n.d. | 1.2 | n.d. | 4.1 | 1.3 | 5.1 |
| 8 | 3 | 387 | 955.4 | 11.2 | 16.2 | 14.9 | 9.0 | 5.8 | 4.0 | 13.6 | 8.6 | 0.5 | t.a. | 2.4 | 1.1 | t.a. | 3.5 | 0.3 | 0.5 | 0.2 | 0.7 | 0.2 | 0.2 | 6.9 | 0.0 | n.d. | 0.0 |
| 8 | 10 | 309 | 1458.8 | 8.5 | 18.5 | 19.2 | 6.0 | 12.1 | 3.2 | 11.6 | 8.3 | 0.2 | 0.1 | 2.5 | 1.4 | 0.1 | 3.8 | 0.1 | t.a. | 0.1 | 1.1 | n.d. | 0.2 | 3.0 | 0.0 | n.d. | n.d. |
| 8 | 25 | 1648 | 348.7 | 13.2 | 22.2 | 11.6 | 6.8 | 12.9 | 3.9 | 9.5 | 8.2 | 0.0 | 0.1 | 1.4 | 1.0 | 0.6 | 3.7 | 0.1 | 0.1 | n.d. | t.a. | 0.1 | 1.3 | 3.2 | 0.0 | n.d. | n.d. |
| 8 | 50 | 887 | 474.4 | 18.0 | 18.3 | 18.4 | 5.7 | 10.2 | 5.3 | 11.3 | 3.1 | 0.2 | 0.1 | 0.3 | 1.1 | 0.6 | 3.2 | n.d. | n.d. | n.d. | 0.3 | 0.5 | 0.7 | 0.9 | 0.8 | 0.2 | 0.7 |
| 8 | 125 | 1231 | 54.2 | 9.8 | 1.9 | 1.7 | 4.4 | 14.1 | 8.5 | 15.8 | 6.5 | 0.9 | t.a. | 0.1 | 1.3 | 0.4 | 3.5 | t.a. | 1.9 | 0.8 | 0.2 | n.d. | 0.4 | 1.0 | 9.1 | 6.7 | 11.0 |
| 8 | 200 | 1698 | 3.8 | 41.1 | 0.2 | 0.3 | 14.5 | n.d. | t.a. | 0.3 | 1.8 | 1.0 | n.d. | n.d. | 0.1 | 0.3 | 0.7 | n.d. | n.d. | n.d. | 0.5 | n.d. | 0.3 | 1.1 | 22.3 | 0.4 | 15.0 |
| 8 | 350 | 1633 | 19.5 | 6.7 | 1.3 | 1.1 | 10.5 | 8.5 | 7.5 | 12.6 | 11.3 | 5.6 | 0.1 | n.d. | 2.3 | 0.5 | 7.2 | 0.2 | 6.0 | 4.3 | 0.1 | n.d. | 1.0 | 1.8 | 6.5 | 1.0 | 3.6 |
| 8 | 450 | 1440 | 1.5 | 24.2 | 0.0 | n.d. | 22.6 | 2.1 | 1.2 | 2.4 | 1.5 | 0.5 | n.d. | n.d. | 0.5 | 0.8 | 1.1 | n.d. | 1.1 | 3.2 | n.d. | n.d. | 0.3 | 2.4 | 22.8 | 0.9 | 12.7 |
| 8 | 550 | 1251 | 17.2 | 9.4 | 1.4 | 3.5 | 8.7 | 9.4 | 11.4 | 15.5 | 15.1 | 1.9 | 0.2 | n.d. | n.d. | 0.4 | 5.8 | 0.1 | 5.3 | 2.9 | 0.2 | n.d. | 0.7 | 0.6 | 4.0 | 1.4 | 2.1 |
| 8 | 650 | 1633 | 5.1 | 32.4 | n.d. | n.d. | 30.4 | 1.2 | 0.1 | 0.6 | 1.2 | 0.0 | n.d. | n.d. | n.d. | 1.3 | 1.0 | n.d. | n.d. | n.d. | n.d. | n.d. | n.d. | 1.6 | 19.5 | 0.1 | 10.5 |
| 8 | 750 | 1926 | 0.1 | 14.5 | n.d. | n.d. | 0.4 | n.d. | 0.1 | 0.1 | n.d. | n.d. | n.d. | n.d. | n.d. | n.d. | t.a. | n.d. | n.d. | n.d. | n.d. | n.d. | n.d. | n.d. | 61.2 | n.d. | 23.8 |
| 8 | 1000 | 1633 | 0.1 | 37.9 | n.d. | n.d. | 1.6 | n.d. | 0.1 | 0.1 | n.d. | n.d. | n.d. | n.d. | n.d. | 0.1 | n.d. | n.d. | n.d. | n.d. | n.d. | n.d. | n.d. | 40.7 | n.d. | 19.8 |
| 8 | 1250 | 1417 | 0.1 | 47.3 | n.d. | n.d. | 0.6 | n.d. | 0.1 | 0.1 | n.d. | n.d. | n.d. | n.d. | n.d. | n.d. | t.a. | n.d. | n.d. | n.d. | n.d. | 0.1 | n.d. | 0.8 | 36.3 | 0.0 | 14.9 |

**Suppl. Table 2.** List of commercially available standards used to determine response factors of intact polar lipids (IPL) in this study. For absolute quantification by HPLC-ion trap-MS the response factor was evaluated relative to the injection standard C19:0 PC-DAG. For determining relative abundances of IPLs via HPLC-QTOF-MS (see methods in the main text), the absolute responses of the individual standards were used.    For IPL abbreviations refer to main text.

| Short ID | Full Name | Fatty acid distribution | Company | Used for IPL class (HPLC-ion trap-MS) | Used for IPL class (HPLC-QTOF-MS) |
|---|---|---|---|---|---|
| 16:0 PE-DAG | 1,2-dipalmitoyl-sn-glycero-3-phosphoethanolamine | 16:0/16:0 | Avanti Polar Lipids, USA | PE-DAG, PME-DAG, PDME-DAG | PE-DAG |
| 16:0 PC-DAG | 1,2-dipalmitoyl-sn-glycero-3-phosphocholine | 16:0/16:0 | Avanti Polar Lipids, USA | PC-DAG | PC-DAG |
| 19:0 PC-DAG | 1,2-dinonadecanoyl-sn-glycero-3-phosphocholine | 19:0/19:0 | Avanti Polar Lipids, USA | | internal standard for all IPLs |
| 16:0 PME-DAG | 1,2-dipalmitoyl-sn-glycero-3-phosphoethanolamine-N-methyl | 16:0/16:0 | Avanti Polar Lipids, USA | | PME-DAG |
| 16:0 PDME-DAG | 1,2-dipalmitoyl-sn-glycero-3-phosphoethanolamine-N,N-dimethyl | 16:0/16:0 | Avanti Polar Lipids, USA | | PDME-DAG |
| 16:0 PG-DAG | 1,2-dipalmitoyl-sn-glycero-3-phospho-(1'-rac-glycerol) | 16:0/16:0 | Avanti Polar Lipids, USA | PG-DAG, SQ-DAG, DPG | PG-DAG, SQ-DAG |
| 16:1 DPG | 1',3'-bis(1,2-dipalmitoleoyl-sn-glycero-3-phospho)-glycerol | 16:1/16:1/16:1/16:1 | Avanti Polar Lipids, USA | | DPG |
| 1Glc-DAG | 1,2-diacyl-3-O-(a-D-glucopyranosyl)-sn-glycerol (E.coli) | 18:1/16:0, 18:1/16:1, 16:1/16:0, 18:1/18:1 | Avanti Polar Lipids, USA | 1G-DAG | 1G-DAG |
| 2G-DAG | Digalactosyldiacylglcyerol (plant, hydrogenated) | 18:0/18:0, 18:0/16:0 | Avanti Polar Lipids, USA | 2G-DAG | 2G-DAG |
| DGTS | 1,2-dipalmitoyl-sn-glycero-3-O-4'-[N,N,N-trimethyl]-homoserine | 16:0/16:0 | Avanti Polar Lipids, USA | DGTS | DGTS |
| C18 1G-CER | D-galactosyl-b-1,1'-N-stearoyl-D-erythro-sphingosine | d18:1/18:0 | Avanti Polar Lipids, USA | 1G-CER | 1G-CER |
| 1G-GDGT-PG | Main phospholipid of Thermoplasma acidophilum (>95% pure) | | Matreya LLC, USA | HPH-GDGT, 1G-GDGT, 2G-GDGT | HPH-GDGT, 1G-GDGT, 2G-GDGT |

**Suppl. Table 3.** Examples of HPLC-MS fragmentation patterns (MS2) in positive ionization mode for select major ions (MS1) of intact polar lipids (IPLs) detected in this study.

| IPL | MS1 (pos ion mode) | MS2 (pos ion mode) | MS2 (pos ion mode) |
|---|---|---|---|
| | Select major ions (m/z) | Neutral loss (Da) | Select diagnostic ions (m/z) |
| *Glycolipids* | | | |
| 1G-DAG | 766.546, 718.546, 716.531 [M+NH$_4$]$^+$ | Hexose plus NH$_3$ (179.079) | 335.258, 305.211, 313.277, 285.245 |
| 1G-OH-DAG | 732.526, 730.510 [M+NH$_4$]$^+$ | Hexose plus NH$_3$ (179.079) | 313.277, 285.245 |
| 2G-DAG | 936.662, 934.646, 882.615, 880.599 [M+NH$_4$]$^+$ | Two hexoses plus NH$_3$ (341.132) | 339.291, 337.274, 313.277, 285.245 |
| SQ-DAG | 812.555, 784.524, 782.508, 756.493 [M+H]$^+$ | Sulfoquinovosyl (261.052) | 313.277, 313.277, 311.258, 285.245 |
| SQ-AEG | 824.592, 798.576 [M+H]$^+$ | Sulfoquinovosyl (261.052) | 339.291 |
| 1G-CER | 748.572, 734.557 [M+H]$^+$ | Hexose plus H$_2$O (180.063) | 294.279 (LCB) |
| 1G-OH-CER | 780.598, 766.583, 752.567 [M+H]$^+$ | Hexose plus H$_2$O (180.063) | 294.279 (LCB) |
| 1G-GDGT | 1481.402, 1471.324 [M+NH$_4$]$^+$ | Hexose plus NH$_3$ (179.079) | 1302.323, 1292.44 |
| 2G-GDGT | 1639.424 [M+NH$_4$]$^+$ | Two hexoses plus NH$_3$ (341.132) | 1298.291 |
| HPH-GDGT | 1723.421, 1713.343 [M+NH$_4$]$^+$ | Hexose plus NH$_3$ (179.079), hexose (162.053) | 1544.342, 1382.289, 1534.264, 1372.211 |
| | | | |
| *Phospholipids* | | | |
| PE-DAG | 730.538, 718.538, 704.522, 678.507 [M+H]$^+$ | Phosphoethanolamine (141.019) | DAG fragments |
| PE-AEG | 706.575, 702.543 [M+H]$^+$ | Phosphoethanolamine (141.019) | |
| PC-DAG | 806.569, 780.554, 746.569, 706.538 [M+H]$^+$ | - | 184.073, DAG fragments |
| PC-AEG | 746.606 742.575, 716.559 [M+H]$^+$ | - | 184.073 |
| PG-DAG | 766.559, 764.544, 762.528 [M+NH$_4$]$^+$ | Phosphoglycerol plus NH$_3$ (189.040) | DAG fragments |
| DPG | 1404.990, 1380.990, 1352.959, 1338.943 [M+NH$_4$]$^+$ | - | DAG fragments |
| PME-DAG | 732.554, 718.538, 692.522 [M+H]$^+$ | Phosphomethylethanolamine (155.035) | DAG fragments |
| PDME-DAG | 732.554, 730.538, 704.522 [M+H]$^+$ | Phosphodimethylethanolamine (169.050) | DAG fragments |
| PI-DAG | 902.575, 900.560 [M+NH$_4$]$^+$ | Phosphoinositol plus NH$_3$ (277.056) | DAG fragments |

*Aminolipids*

[revised manuscript text omitted]

**Suppl. Table 5.** Fatty acyl combinations (number of carbon atoms and double bond equivalents, DBE, in the alkyl side chains) of the major groups of intact polar lipids and their relative abundance at different depths within the water column of the Eastern Tropical North Pacific.

| IPL | m/z (pos mode) | Carbon atoms | DBE | FA combination | Rel. Abundance (%) Oxic Mean | Oxic SD | Upper OMZ Mean | Upper OMZ SD | Core OMZ Mean | Core OMZ SD | Deep Oxycline Mean | Deep Oxycline SD |
|---|---|---|---|---|---|---|---|---|---|---|---|---|
| 1G-DAG | 788.531 | 36 | 8 | 18:4/18:4 | 1.86 | 1.03 | 3.55 | 4.15 | 0.74 | 0.50 | 1.10 | 0.24 |
| | 786.515 | 36 | 9 | 18:4/18:5 | 6.05 | 7.89 | 1.72 | 1.67 | 0.84 | 0.87 | 1.52 | 0.31 |
| | 774.609 | 34 | 1 | | 2.69 | 1.99 | 1.48 | 1.36 | 2.59 | 1.93 | 2.96 | 1.06 |
| | 766.546 | 34 | 5 | | 1.17 | 0.82 | 1.74 | 1.31 | 1.11 | 0.47 | 3.43 | 3.63 |
| | 762.515 | 34 | 7 | 18:3/16:4 | 5.07 | 3.61 | 1.59 | 1.00 | 2.20 | 1.52 | 2.70 | 0.61 |
| | 746.578 | 32 | 1 | | 3.82 | 5.24 | 3.59 | 2.09 | 4.53 | 1.78 | 4.54 | 0.71 |
| | 732.562 | 31 | 1 | | 2.09 | 2.55 | 2.02 | 1.51 | 2.96 | 1.93 | 2.43 | 1.72 |
| | 720.562 | 30 | 0 | 16:0/14:0 | 6.98 | 10.09 | 5.76 | 2.92 | 7.15 | 3.49 | 6.99 | 0.86 |
| | 718.546 | 30 | 1 | 16:1/14:0 | 27.91 | 10.70 | 40.79 | 11.46 | 49.59 | 5.41 | 39.07 | 3.42 |
| | 716.531 | 30 | 2 | 16:2/14:0 | 10.59 | 7.40 | 11.90 | 5.88 | 7.93 | 2.21 | 7.74 | 2.53 |
| | 692.531 | 28 | 0 | | 4.79 | 2.82 | 2.97 | 1.30 | 4.48 | 1.18 | 3.71 | 1.20 |
| | 690.515 | 28 | 1 | | 3.83 | 2.61 | 4.54 | 2.48 | 3.18 | 1.18 | 2.45 | 1.35 |
| 2G-DAG | 948.568 | 36 | 9 | | 2.11 | 1.62 | 1.18 | 1.35 | 0.41 | 0.50 | 4.10 | 6.60 |
| | 936.662 | 34 | 1 | 18:1/16:0 | 5.27 | 2.43 | 4.50 | 4.54 | 13.94 | 1.86 | 11.00 | 5.92 |
| | 934.646 | 34 | 2 | 18:2/16:0 | 6.68 | 2.86 | 2.87 | 2.77 | 7.43 | 3.02 | 9.11 | 4.42 |
| | 932.631 | 34 | 3 | 18:3/16:0 | 4.94 | 1.71 | 2.45 | 1.91 | 2.95 | 1.75 | 4.93 | 1.89 |
| | 930.615 | 34 | 4 | | 3.15 | 1.65 | 1.66 | 1.62 | 1.54 | 0.89 | 3.37 | 1.14 |
| | 928.599 | 34 | 5 | 18:3/16:2 | 4.11 | 1.86 | 1.64 | 1.22 | 1.43 | 0.93 | 6.93 | 9.29 |
| | 926.584 | 34 | 6 | | 3.12 | 1.66 | 1.61 | 1.56 | 1.36 | 0.91 | 2.28 | 1.57 |
| | 924.568 | 34 | 7 | 18:3/16:4 | 4.90 | 2.40 | 1.73 | 1.78 | 1.20 | 0.80 | 2.75 | 0.39 |
| | 908.631 | 32 | 1 | 18:1/14:0 ; 16:1/16:0 | 5.03 | 2.49 | 13.67 | 12.27 | 23.54 | 5.07 | 15.29 | 4.20 |
| | 906.615 | 32 | 2 | | 2.20 | 0.92 | 3.52 | 4.42 | 2.89 | 1.71 | 3.42 | 0.84 |
| | 902.584 | 32 | 4 | 18:4/14:0 | 5.45 | 2.91 | 1.14 | 2.07 | 0.78 | 0.68 | 0.76 | 0.94 |
| | 900.568 | 32 | 5 | 18:5/14:0 | 4.76 | 2.10 | 4.25 | 13.38 | 0.17 | 0.49 | 0.63 | 1.27 |
| | 898.552 | 32 | 6 | | 4.19 | 7.62 | 0.27 | 0.42 | 0.00 | 0.00 | 0.00 | 0.00 |
| | 882.615 | 30 | 0 | 16:0/14:0 | 13.49 | 6.98 | 17.20 | 7.58 | 15.68 | 1.76 | 9.10 | 7.37 |
| | 880.599 | 30 | 1 | 16:1/14:0 | 9.95 | 3.90 | 24.46 | 11.88 | 16.88 | 2.52 | 9.50 | 6.58 |
| | 878.584 | 30 | 2 | 16:2/14:0 | 2.71 | 2.54 | 5.21 | 4.96 | 0.09 | 0.26 | 0.00 | 0.00 |
| | 854.584 | 28 | 0 | | 3.82 | 2.91 | 3.17 | 3.47 | 2.73 | 4.45 | 1.72 | 3.44 |
| SQ-DAG | 838.571 | 34 | 1 | 18:1/16:0 | 1.97 | 1.08 | 2.83 | 3.36 | 4.98 | 4.57 | 8.52 | 7.97 |
| | 836.555 | 34 | 2 | | 4.02 | 2.38 | 1.70 | 1.63 | 1.02 | 1.23 | 1.64 | 1.90 |

| | m/z | C | DB | Composition | | | | | | | | |
|---|---|---|---|---|---|---|---|---|---|---|---|---|
| | 834.540 | 34 | 3 | | 3.22 | 1.71 | 0.56 | 0.72 | 0.44 | 0.87 | 1.24 | 1.43 |
| | 812.555 | 32 | 0 | 16:0/16:0 ; 15:0/14:0 | 9.15 | 3.10 | 4.46 | 3.78 | 9.29 | 4.07 | 12.60 | 3.12 |
| | 810.540 | 32 | 1 | 16:1/16:0 | 7.29 | 2.63 | 11.61 | 6.09 | 12.53 | 2.50 | 10.36 | 4.74 |
| | 808.524 | 32 | 2 | 16:1/16:1 | 5.23 | 3.07 | 11.49 | 7.09 | 6.69 | 3.79 | 5.43 | 4.34 |
| | 784.524 | 30 | 0 | 16:0/14:0 | 26.72 | 5.24 | 24.93 | 16.07 | 30.45 | 3.33 | 30.89 | 3.33 |
| | 782.508 | 30 | 1 | 16:1/14:0 | 15.90 | 6.46 | 31.16 | 11.93 | 13.75 | 4.21 | 12.69 | 4.69 |
| | 780.493 | 30 | 2 | 16:2/14:0 | 4.95 | 3.47 | 2.81 | 4.28 | 7.29 | 11.51 | 0.47 | 0.54 |
| | 756.493 | 28 | 0 | 14:0/14:0 | 16.24 | 6.46 | 6.84 | 5.26 | 13.57 | 11.15 | 13.66 | 9.59 |
| PG-DAG | 806.591 | 37 | 2 | | 1.59 | 2.88 | 1.88 | 2.51 | 3.07 | 1.23 | 2.63 | 1.39 |
| | 792.575 | 36 | 2 | 18:1/18:1 | 9.81 | 5.61 | 7.12 | 3.44 | 9.78 | 3.67 | 7.69 | 0.25 |
| | 780.575 | 35 | 1 | | 1.55 | 2.32 | 3.48 | 2.67 | 4.22 | 1.57 | 3.26 | 0.62 |
| | 778.559 | 35 | 2 | 18:1/17:1 ; 19:1/16:1 | 3.26 | 2.46 | 10.69 | 6.95 | 13.26 | 2.11 | 11.05 | 0.62 |
| | 766.559 | 34 | 1 | 18:1/16:0 | 12.32 | 2.78 | 8.63 | 8.75 | 7.21 | 2.63 | 7.00 | 0.45 |
| | 764.544 | 34 | 2 | 18:2/16:0 | 21.87 | 10.62 | 18.84 | 8.87 | 19.50 | 4.17 | 25.57 | 4.49 |
| | 762.528 | 34 | 3 | 18:3/16:0 | 15.03 | 9.19 | 2.67 | 3.22 | 3.78 | 1.34 | 3.78 | 0.32 |
| | 752.544 | 33 | 1 | 17:1/15:0 | 7.52 | 4.63 | 13.03 | 5.80 | 16.60 | 2.27 | 14.65 | 3.77 |
| | 738.528 | 32 | 1 | 16:1/16:0 | 11.41 | 4.41 | 11.43 | 5.08 | 10.84 | 3.29 | 10.11 | 1.45 |
| | 736.512 | 32 | 2 | 16:1/16:1 | 13.19 | 3.79 | 19.08 | 12.58 | 9.77 | 2.20 | 10.42 | 0.80 |
| | 710.497 | 30 | 1 | | 2.31 | 1.41 | 3.09 | 1.76 | 1.76 | 1.22 | 3.84 | 0.17 |
| PE-DAG | 730.538 | 32 | 2 | 18:1/17:1 | 12.02 | 9.37 | 19.53 | 13.16 | 23.32 | 7.97 | 19.51 | 11.03 |
| | 718.538 | 34 | 1 | 18:1/16:0 | 7.28 | 6.14 | 6.44 | 7.72 | 11.29 | 5.49 | 9.85 | 6.57 |
| | 716.522 | 34 | 2 | 18:1/16:1 ; 17:1/17:1 | 3.78 | 3.64 | 2.91 | 2.90 | 9.97 | 8.36 | 3.83 | 2.93 |
| | 704.522 | 33 | 1 | 17:1/16:0 | 22.14 | 14.88 | 16.16 | 8.54 | 13.55 | 2.63 | 10.32 | 6.89 |
| | 702.507 | 33 | 2 | | 1.32 | 1.33 | 1.79 | 2.25 | 2.05 | 1.35 | 4.12 | 5.50 |
| | 690.507 | 32 | 1 | 16:1/16:0 | 6.93 | 10.80 | 5.53 | 7.05 | 7.11 | 4.76 | 8.29 | 10.09 |
| | 688.491 | 32 | 2 | | 2.26 | 4.02 | 3.14 | 10.46 | 3.16 | 3.63 | 0.48 | 0.97 |
| | 678.507 | 31 | 0 | 16:0/15:0 | 33.48 | 20.83 | 23.89 | 15.41 | 10.07 | 4.24 | 15.60 | 11.12 |
| | 676.491 | 31 | 1 | | 3.82 | 3.31 | 2.83 | 2.43 | 3.07 | 1.10 | 1.66 | 1.85 |
| | 674.476 | 31 | 2 | | 0.09 | 0.30 | 0.02 | 0.07 | 0.04 | 0.11 | 0.00 | 0.00 |
| | 664.491 | 30 | 0 | 16:0/14:0 | 1.54 | 2.69 | 8.02 | 8.30 | 6.76 | 2.37 | 11.74 | 4.36 |
| | 662.476 | 30 | 1 | | 1.57 | 1.44 | 2.74 | 5.83 | 1.92 | 1.77 | 2.72 | 4.35 |
| | 650.476 | 29 | 0 | 15:0/14:0 | 2.61 | 2.09 | 6.66 | 5.43 | 6.50 | 3.55 | 10.94 | 8.75 |
| | 648.460 | 29 | 1 | | 0.08 | 0.19 | 0.14 | 0.34 | 0.04 | 0.11 | 0.00 | 0.00 |
| | 636.460 | 28 | 0 | | 1.08 | 1.54 | 0.21 | 0.49 | 1.17 | 0.88 | 0.95 | 1.44 |
| PC-DAG | 878.575 | 44 | 12 | 22:6/22:6 | 3.87 | 3.27 | 6.12 | 11.60 | 3.83 | 7.70 | 2.29 | 3.39 |
| | 852.560 | 42 | 11 | 22:6/20:5 | 1.87 | 1.60 | 2.90 | 3.11 | 3.55 | 6.04 | 4.79 | 5.74 |
| | 822.601 | 39 | 5 | | 1.59 | 1.12 | 1.50 | 1.49 | 0.67 | 0.93 | 1.47 | 1.35 |
| | 806.569 | 38 | 6 | 22:6/16:0 | 20.28 | 7.02 | 19.69 | 16.84 | 13.97 | 11.29 | 36.10 | 36.79 |
| | 788.616 | 36 | 1 | | 1.74 | 1.20 | 1.76 | 1.63 | 2.28 | 4.20 | 0.74 | 0.68 |

| | m/z | C | DB | ID | | | | | | | | |
|---|---|---|---|---|---|---|---|---|---|---|---|---|
| | 780.554 | 36 | 5 | 20:5/16:0 | 12.14 | 7.79 | 9.03 | 6.74 | 4.58 | 3.79 | 8.41 | 6.72 |
| | 776.616 | 35 | 0 | | 0.49 | 0.66 | 1.09 | 1.23 | 0.59 | 0.58 | 0.71 | 0.77 |
| | 774.601 | 35 | 1 | | 1.69 | 0.98 | 1.68 | 1.58 | 1.53 | 1.07 | 2.12 | 3.00 |
| | 760.585 | 34 | 1 | 18:1/16:0 | 6.68 | 4.39 | 5.27 | 3.65 | 8.01 | 8.70 | 2.70 | 3.23 |
| | 754.538 | 34 | 4 | | 2.73 | 1.57 | 1.86 | 2.02 | 0.76 | 0.66 | 1.15 | 1.10 |
| | 748.585 | 33 | 0 | | 1.21 | 1.18 | 1.93 | 1.89 | 1.59 | 1.36 | 1.24 | 1.20 |
| | 746.569 | 33 | 1 | | 1.85 | 3.95 | 3.18 | 3.63 | 3.88 | 2.12 | 2.66 | 2.43 |
| | 744.554 | 33 | 2 | 17:1/16:1 | 0.91 | 1.03 | 1.97 | 1.68 | 5.96 | 6.54 | 2.33 | 1.39 |
| | 734.569 | 32 | 0 | 16:0/16:0 ; 17:0/15:0 | 6.20 | 3.84 | 4.16 | 3.36 | 4.78 | 4.00 | 3.09 | 2.34 |
| | 732.554 | 32 | 1 | 16:0/16:1 | 4.83 | 2.98 | 5.96 | 3.69 | 7.15 | 3.92 | 6.15 | 3.68 |
| | 730.538 | 32 | 2 | 16:1/16:1 | 2.19 | 1.51 | 4.96 | 4.91 | 7.57 | 4.32 | 4.47 | 3.08 |
| | 720.554 | 31 | 0 | | 1.86 | 1.32 | 2.86 | 3.45 | 2.09 | 1.43 | 1.61 | 1.59 |
| | 718.538 | 31 | 1 | | 1.65 | 1.61 | 1.91 | 3.22 | 3.86 | 2.53 | 2.63 | 2.51 |
| | 716.522 | 31 | 2 | | 0.91 | 1.01 | 0.87 | 1.10 | 4.55 | 5.76 | 0.97 | 0.89 |
| | 706.538 | 30 | 0 | 16:0/14:0 | 9.35 | 2.79 | 5.68 | 4.00 | 3.29 | 1.87 | 2.53 | 2.65 |
| | 704.522 | 30 | 1 | 16:1/14:0 | 4.89 | 4.78 | 3.71 | 2.41 | 4.62 | 2.79 | 2.75 | 2.62 |
| | 702.507 | 30 | 2 | | 0.25 | 0.27 | 0.48 | 0.66 | 0.87 | 0.90 | 1.19 | 1.98 |
| | 692.522 | 29 | 0 | | 2.13 | 1.68 | 3.21 | 2.35 | 2.26 | 1.52 | 1.67 | 1.99 |
| | 690.507 | 29 | 1 | | 2.01 | 3.26 | 1.57 | 1.99 | 3.08 | 3.45 | 2.38 | 3.68 |
| | 688.491 | 29 | 2 | | 0.45 | 0.64 | 0.76 | 2.63 | 1.57 | 2.59 | 0.10 | 0.21 |
| | 678.507 | 28 | 0 | 14:0/14:0 | 6.22 | 3.39 | 5.89 | 4.62 | 3.09 | 1.68 | 3.76 | 3.82 |
| DGTS | 764.640 | 36 | 2 | 18:1/18:1 | 6.72 | 3.20 | 8.70 | 4.81 | 14.23 | 4.45 | 7.86 | 2.68 |
| | 762.624 | 36 | 3 | | 3.66 | 1.56 | 2.53 | 0.76 | 1.43 | 0.64 | 3.14 | 0.91 |
| | 760.609 | 36 | 4 | 18:2/18:2 ; 20:0/16:4 | 7.36 | 2.99 | 4.41 | 2.02 | 1.57 | 0.80 | 5.41 | 2.37 |
| | 758.593 | 36 | 5 | 18:2/18:3 | 5.12 | 2.44 | 2.89 | 1.46 | 1.45 | 0.80 | 3.96 | 1.22 |
| | 740.640 | 34 | 0 | | 1.11 | 0.39 | 2.65 | 1.48 | 4.93 | 1.81 | 1.98 | 1.10 |
| | 738.624 | 34 | 1 | 18:1/16:0 | 3.90 | 1.70 | 14.87 | 13.30 | 32.49 | 12.73 | 14.69 | 2.83 |
| | 736.609 | 34 | 2 | 18:2/16:0 | 7.44 | 3.95 | 6.57 | 1.85 | 8.60 | 0.91 | 8.33 | 2.26 |
| | 734.593 | 34 | 3 | 18:3/16:0 | 5.04 | 2.11 | 2.84 | 1.05 | 1.14 | 0.59 | 3.33 | 0.85 |
| | 732.577 | 34 | 4 | | 4.34 | 1.23 | 2.15 | 1.12 | 0.58 | 0.48 | 1.30 | 1.08 |
| | 710.593 | 32 | 1 | 18:1/14:0 | 3.94 | 1.25 | 6.92 | 2.50 | 9.60 | 4.41 | 10.54 | 2.56 |
| | 708.577 | 32 | 2 | 18:2/14:0 | 3.85 | 1.31 | 3.80 | 1.46 | 2.89 | 2.26 | 5.61 | 1.86 |
| | 706.562 | 32 | 3 | | 2.65 | 0.83 | 2.07 | 1.25 | 0.74 | 0.41 | 3.05 | 2.00 |
| | 698.593 | 31 | 0 | | 1.40 | 0.47 | 2.90 | 2.18 | 2.26 | 2.04 | 3.24 | 1.44 |
| | 684.577 | 30 | 0 | 16:0/14:0 | 3.90 | 1.45 | 3.69 | 1.69 | 2.18 | 1.75 | 3.45 | 0.77 |
| | 682.562 | 30 | 1 | 16:1/14:0 | 3.58 | 3.14 | 8.92 | 4.80 | 4.13 | 4.52 | 7.75 | 0.96 |
| | 670.562 | 29 | 0 | | 3.60 | 1.39 | 3.47 | 1.93 | 2.14 | 1.62 | 2.34 | 0.76 |
| | 656.546 | 28 | 0 | 14:0/14:0 | 9.56 | 4.74 | 6.54 | 3.47 | 4.44 | 2.18 | 4.81 | 0.46 |

**Suppl. Table 6.** Goodness of fit statistics for the NMDS analyses of normalized intact polar lipid (IPL) composition and quantitative microbial data (FISH), number of double bond equivalents (DBE) and carbon atoms in the alkyl side chains of IPLs.

| FISH Probe | IPL (relative abundance) | | | | Environmental parameter | Number of DBE | | | | Environmental parameter | Number of carbon atoms | | | |
|---|---|---|---|---|---|---|---|---|---|---|---|---|---|---|
| | NMDS1 | NMDS2 | r2 | p | | NMDS1 | NMDS2 | r2 | p | | NMDS1 | NMDS2 | r2 | p |
| Alphaproteobacteria | -0.56 | -0.83 | 0.02 | 0.613 | Depth | -0.80 | 0.59 | 0.12 | 0.081 | Depth | 0.48 | 0.88 | 0.00 | 0.944 |
| Betaproteobacteria | -0.60 | -0.80 | 0.10 | 0.133 | POC | 0.93 | -0.35 | 0.19 | 0.019 | POC | -0.14 | 0.99 | 0.05 | 0.340 |
| Gammaproteobacteria | -0.66 | -0.76 | 0.24 | 0.004 | TN | 0.95 | -0.30 | 0.20 | 0.012 | TN | -0.10 | 1.00 | 0.56 | 0.324 |
| SRB | -0.42 | -0.91 | 0.33 | 0.002 | Phosphate | -0.90 | 0.44 | 0.12 | 0.740 | Phosphate | 0.06 | -1.00 | 0.03 | 0.531 |
| Epsilonproteobacteria | -0.88 | -0.47 | 0.18 | 0.022 | Nitrate | -0.51 | 0.86 | 0.07 | 0.250 | Nitrate | 0.65 | -0.76 | 0.34 | 0.472 |
| Nso | -0.94 | -0.33 | 0.18 | 0.023 | Nitrite | -0.57 | -0.82 | 0.14 | 0.043 | Nitrite | -0.54 | -0.84 | 0.15 | 0.052 |
| Anammox | -0.04 | -1.00 | 0.25 | 0.006 | Ammonium | 0.67 | -0.75 | 0.06 | 0.311 | Ammonium | -0.15 | 0.99 | 0.05 | 0.339 |
| Planctomycetes | -0.87 | -0.50 | 0.13 | 0.730 | Salinity | 0.96 | 0.29 | 0.00 | 0.909 | Salinity | 0.94 | 1.00 | 0.01 | 0.713 |
| | | | | | Temperature | 0.66 | -0.76 | 0.02 | 0.686 | Temperature | -0.23 | -0.97 | 0.01 | 0.882 |
| | | | | | Fluorescence | 0.97 | -0.26 | 0.27 | 0.003 | Fluorescence | 0.04 | 1.00 | 0.05 | 0.374 |
| | | | | | Oxygen | 0.83 | -0.55 | 0.26 | 0.002 | Oxygen | -0.12 | 0.99 | 0.09 | 0.138 |
| | | | | | Chl-a | 0.96 | -0.27 | 0.28 | 0.002 | Chl-a | -0.01 | 1.00 | 0.16 | 0.042 |

**Suppl. Figure 1.** Fluctuations in (A) absolute and (B) relative responses of select commercially available IPL standards over time. The values represent the slope of standards measured in different concentrations (usually 100 pg to 10 ng injected on column). Standard Mix A, B and C represents newly prepared standard mixtures. The standard mix used in this study was from November 2015.

**Suppl. Figure 2.** Depth profiles of (a) total particulate nitrogen and (b) phaeophytin concentrations, at the investigated four stations in the ETNP.

**Suppl. Figure 3.** Structures of (a) bacterial/eukaryotic and (b) archaeal intact polar lipids (IPLs) observed in the ETNP. The position of the double bonds or rings and the OH-, epoxy- and keto-groups of the R' and R'' side chains were not determined.

**Suppl. Figure 4.** Extracted ion chromatograms of intact polar and core GDGTs showing the ring distribution within each individual compound class. Analyses were performed by HPLC-QTOF-MS with reversed phase chromatography as described in Zhu et al., (2016).   Extracted ions for HPH-GDGTs were: $m/z$ 1723.421, 1721.406, 1719.390, 1717.374, 1715.359, 1713.343; for 2G-GDGT: $m/z$ 1643.455, 1641.439, 1639.424, 1637.408, 1635.392, 1633.377; for 1G-GDGT: $m/z$ 1481.402, 1479.386, 1477.371, 1475.355, 1473.339, 1471.324 and for core GDGTs: $m/z$ 1302.323, 1300.307, 1298.291, 1296.276, 1294.2, 1292.244. The numbers denote number of rings: 0 – GDGT-0, 1 – GDGT-1, 2 – GDGT-2, 5 – crenarchaeol.

**Suppl. Figure 5.** Identification of hydroxylated aminolipids and sphingolipids in water column samples of the ETNP. (a) HPLC-MS density map, full scan (MS[1]), in the mass range from m/z 600 to 900 and retention time range from 7 to 9 minutes. Representative high-resolution accurate mass MS[2] mass spectrum showing fragmentation patterns of (b) DGTS, (c) 1OH-DGTS, (d) 3OH-DGTS and (e) 1G-2OH-CER in positive ionization mode. Typical fragments for DGTS include monoacylglycerol side chains with the head group still attached. Similar fragmentation patters are observed between DGTS, 1OH-DGTS and 3OH-DGTS with exact masses pointing to additional hydroxyl-groups attached to the fatty acyl side chains. Note, that it's possible that the dihydroxylated fatty acid, could also be an epoxy-hydroxy or keto-hydroxy acid as only one loss of water was observed in the MS2 (from fragment *m/z* 466.281 to *m/z* 448.270). Multiple fatty acid side chain combinations are possible. Fragments of 1G-2OH-CER include the glycosidic head group loss of 180 Da and two hydroxyl-group losses as well as the long chain base (LCB), *m/z* 294.279.

**Suppl. Figure 6.** Identification of aminolipids AL-I and AL-II and ester/ether-sulfoquinovosyl (SQ-AEG) in water column samples of the ETNP. (a) HPLC-MS density map, full scan (MS[1]), in the mass range from *m/z* 600 to 900 and retention time range from 6.5 to 11.5 minutes. Representative high-resolution accurate mass MS[2] mass spectrum showing fragmentation patterns of (b) AL-I and (c) AL-II in positive ionization mode. Fragmentation patterns of AL-I and AL-II are very similar to DGTS (Suppl. Fig. 3) showing monoacylglycerol fragments with the amino-head group still attached. The sum formula of the AL-I headgroup matches the head group of DGCC with an extra methyl group. However, since no head group fragments were observed no further structural inference could be made. The sum formula of AL-II matches exactly the head group of DGCC, however, the DGCC-characteristic head group ion fragment

*m/z* 252.144 was not observed and no structural inference from the detected head group fragments *m/z* 132.102 and 104.107 could be made. Representative high-resolution accurate mass MS$^2$ mass spectrum showing fragmentation patterns of (d) SQ-DAG and (e) SQ-AEG in positive and negative ionization mode. Both compound classes exhibit the sulfoquinovosyl-diagnostic head group loss of 261.05 Da. However, SQ-AEG only has one fatty acyl side chain fragment, whereas SQ-DAG has two fatty acyl fragments in positive and negative ion mode. Furthermore, the exact mass of the parent ion and the fragments indicate that SQ-AEG has one oxygen less than SQ-DAG, indicating the replacement of one of the ester bonds with an ether bond.

suppl.fig01

suppl.fig02

a) TN (µg L⁻¹)
b) Phaeophytin (µg L⁻¹)

a) Station 1

b) Station 2

c) Station 5

d) Station 8

suppl.fig03

[Figure]

**(a)**

Monoglycosyl (1G), n=1
Diglycosyl (2G), n=2
Triglycosyl (3G), n=3

Sulfoquinovosyl
(SQ)

Diphosphatidylglycerol
(DPG)

Phosphatidylglycerol
(PG)

Phosphatidyl-
ethanolamine
(PE)

Phosphatidyl-(*N*)
methylethanolamine
(PME)

Phosphatidyl-(*N,N*)-
dimethylethanolamine
(PDME)

Phosphatidylcholine
(PC)

Betaine lipid
Diacylglyceryl trimethylhomoserine
(DGTS)

Ornithine lipid
(OL)

Phosphatidylinositol
(PI)

Diacylglycerol
(DAG)

Diether-glycerol
(DEG)

Acyl/ether-glycerol
(AEG)

Ceramide
(CER)

R',R'' - variations:

saturated

Mono-hydroxy

Epoxy-hydroxy polyunsaturated

Di-hydroxy

Keto-hydroxy

**(b)**

1G-GDGT, n=1
2G-GDGT, n=2

HPH-GDGT

X,Y - variations:

[Figure]

suppl.fig05

suppl.fig06

[Figure]

**(a)**

**(b)** AL-1 (C40:6) ESI⁺
*m/z* 842.650

**(c)** AL-II (C34:2) ESI⁺
*m/z* 800.604

**(d)** SQ-DAG (C30:0) ESI⁺
*m/z* 784.524

SQ-DAG (C30:0) ESI⁻, *m/z* 765.483

**(e)** SQ-AEG (C34:1) ESI⁺
*m/z* 824.592

SQ-AEG (C34:1) ESI⁻, *m/z* 805.551